# CONVERGENCE RATE OF PRIMAL-DUAL APPROACH TO CONSTRAINED REINFORCEMENT LEARNING WITH SOFTMAX POLICY

## ABSTRACT

In this paper, we consider primal-dual approach to solve constrained reinforcement learning (RL) problems, where we formulate constrained reinforcement learning under constrained Markov decision process (CMDP). We propose the primal-dual policy gradient (PD-PG) algorithm with softmax policy. Although the constrained RL involves a non-concave maximization problem over the policy parameter space, we show that for both exact policy gradient and model-free learning, the proposed PD-PG needs iteration complexity of $\mathcal{O}\left(\epsilon^{-2}\right)$ to achieve its optimal policy for both constraint and reward performance. Such an iteration complexity outperforms or matches most constrained RL algorithms. For the learning with exact policy gradient, the main challenge is to show the positivity of deterministic optimal policy (at the optimal action) is independent on both state space and iteration times. For the model-free learning, since we consider the discounted infinite-horizon setting, and the simulator can not rollout with an infinite-horizon sequence; thus one of the main challenges lies in how to design unbiased value function estimators with finite-horizon trajectories. We consider the unbiased estimators with finite-horizon trajectories that involve geometric distribution horizons, which is the key technique for us to obtain the theoretical results for model-free learning.

## 1 INTRODUCTION

Reinforcement learning (RL) has achieved significant success in many fields (e.g., (Silver et al., 2017; Vinyals et al., 2019; OpenAI, 2019)). However, most RL algorithms improve the performance under the assumption that an agent is free to explore any behaviors (that may be detrimental). For example, a robot agent should avoid playing actions that irrevocably harm its hardware (Deisenroth et al., 2013).Thus, it is important to consider *safe exploration* that is known as constrained RL (or safe RL), which is usually formulated as constrained Markov decision processes (CMDP) (Altman, 1999).

The primal-dual approach (Altman, 1999; Bertsekas, 2014) is a fundamental way to solve CMDP problems. Recently, the primal-dual method has also been extended to policy gradient (e.g.,(Tessler et al., 2019; Petsagkourakis et al., 2020; Xu et al., 2021)). However, most previous work only focus on natural policy gradient (NPG) (Kakade, 2002) to solve constrained RL (e.g., (Ding et al., 2020; Xu et al., 2021; Zeng et al., 2021)), little is known about the vanilla policy gradient (Sutton et al., 2000) with primal-dual approach to constrained RL, which involves the following foundational theoretical issues: (i) how to employ the primal-dual vanilla policy gradient method to constrained RL with exact information and model-free learning? (ii) how fast does primal-dual vanilla policy gradient converge to the optimal policy? (iii) what is the sample complexity of the primal-dual policy gradient? These questions are the focus of this paper, and we mainly consider softmax policy for the discounted infinite-horizon CMDP with finite action space and state space.

### 1.1 MAIN CONTRIBUTIONS

**Constrained RL with Exact Policy Gradient**. In Section 3, we propose a primal-dual policy gradient (PD-PG) algorithm, which improves reward performance via gradient ascent on the primal policy parameter space and plays safe explorations via projecting gradient descent on the dual space.

Theorem 2 shows that PD-PG with exact policy gradient needs the iteration complexity of

$$\mathcal{O}\left(\left\|\frac{d_{\pi_\star}^{\rho_0}}{\rho_0}\right\|_\infty^2 \frac{|\mathcal{S}|\log|\mathcal{A}|}{c_\star(1-\gamma)^4\epsilon^2}\right) \tag{1}$$

to obtain the $\mathcal{O}(\epsilon)$-optimality, where $c_\star$ is the infimum of the probability of the optimal action from softmax policy, $c_\star$ is a positive scalar independent on the time-step $t$ and independent on the state space $\mathcal{S}$. One of the main challenges to obtain the complexity (1) is to show that $c_\star$ is bounded away from 0, see Proposition 2. From Table 1, we know the proposed PD-PG is with the iteration complexity of $\widetilde{\mathcal{O}}(\epsilon^{-2})$, which is comparable to extensive constrained RL algorithms.

**Model-Free Constrained RL**. In Section 4, we propose a sample-based PD-PG that only uses empirical data to learn a safe policy. The sample-based PD-PG needs a complexity of

$$\mathcal{O}\left(\left\|\frac{d_{\pi_\star}^{\rho_0}}{\rho_0}\right\|_\infty^2 \frac{|\mathcal{S}|\left(|\mathcal{S}||\mathcal{A}|+m\right)\log|\mathcal{A}|}{c_\star(1-\gamma)^4\epsilon^2}\right) \tag{2}$$

to obtain the $\mathcal{O}(\epsilon)$-optimality, where $m$ is the number of constraints. The iteration complexity (2) outperforms or matches extensive existing state-of-the-art constrained RL algorithms, see Table 1. Since this work considers discounted *infinite-horizon* CMDP, and the simulator can not rollout with an infinite-horizon sequence; thus the main challenge lies in designing unbiased value function estimators with finite-horizon trajectories. In Section 4.2, according to Paternain (2018, Chapter 6), we introduce unbiased estimators with finite-horizon trajectories that involve geometric distribution horizons, which plays a critical role for us to obtain the iteration complexity of sample-based PD-PG. Finally, in Section 4.6, we also illustrate an iteration complexity trade-off between PD-PG and NPD-PG (Ding et al., 2020), where we analyze it from the trade-off between the distribution mismatch coefficient $\left\|\frac{d_{\pi_\star}^{\rho_0}}{\rho_0}\right\|_\infty$ (contained in the proposed PD-PG) and the Moore-Penrose pseudo inverse Fisher information matrix $\mathbf{F}^\dagger(\boldsymbol{\theta})$ (contained in NPD-PG (Ding et al., 2020)).

## 1.2 RELATED WORK

**Constrained RL with Exact Policy Gradient**. The proposed PD-PG (Algorithm 1) is Lagrangian-based CMDP algorithm (Borkar, 2005; Bhatnagar & Lakshmanan, 2012; Liang et al., 2018; Tessler et al., 2019; Yu et al., 2019; Chow et al., 2017; Koppel et al., 2019; Miryoosefi et al., 2019; Paternain et al., 2019a;b). However, those work only focus on the asymptotic convergence results. Primal-dual method is extended with policy gradient (e.g.,(Borkar, 2005; Bhatnagar & Lakshmanan, 2012; Tessler et al., 2019; Petsagkourakis et al., 2020; Wachi et al., 2021)), but those work focus on natural policy gradient (NPG) with Fisher information (Kakade, 2002) or regularized policy iteration to solve constrained RL problems (e.g., (Bharadhwaj et al., 2021)). It is still known litter about vanilla policy gradient (Williams, 1992; Sutton et al., 2000) with primal-dual approach (i.e., the proposed PD-PG) to constrained RL. This work studies the finite-sample performance of the vanilla PD-PG. From Table 1 we know expect UCBVI-$\gamma$ (He et al., 2021) outperforms PD-PG by a factor $\frac{1}{1-\gamma}$, PD-PG is comparable to extensive existing state-of-the-art CMDP algorithms.

**Model-Free Constrained RL**. Model-free constrained RL algorithms, including CPO (Achiam et al., 2017), IPO (Liu et al., 2020), Lyapunov-Based Safe RL (Chow et al., 2018), SAILR (Wagener et al., 2021), SPRL (Sohn et al., 2021), SNO-MDP (Wachi & Sui, 2020), A-CRL (Calvo-Fullana et al., 2021) and DCRL (Qin et al., 2021) all lack convergence rate analysis.

Recently, Ding et al. (2020) propose the natural policy gradient primal-dual (NPD-PG) method for solving discounted infinite-horizon CMDP. Even though the underlying maximization involves a non-concave objective function and a nonconvex constraint setting under the softmax policy parametrization, Ding et al. (2020) show NPD-PG converges at sublinear rates regarding both the optimality gap and the constraint violation, which shares a similar iteration complexity as the proposed PD-PG. Later, Zeng et al. (2021) extend the critical idea of NPD-PG, propose an online version of NPD-PG, and show their algorithm needs the sample complexity of $\mathcal{O}(\epsilon^{-6})$. Xu et al. (2021) propose a primal-type algorithmic framework to solve SRL problems, and Xu et al. (2021) show it needs $\mathcal{O}(\epsilon^{-4})$ sample complexity to obtain $\mathcal{O}(\epsilon)$-optimality [7]. Finally, from Table 1 we know PD-PG achieves the best sample complexity among the policy-based safe RL algorithms.

| Exact Information | Algorithm | Iteration Complexity | Model-Free Learning | Algorithm | Iteration Complexity |
|---|---|---|---|---|---|
| Value-Based | OptPrimalDual-CMDP (Efroni et al., 2020) | $\mathcal{O}\left(\frac{|\mathcal{S}|^2|\mathcal{A}|}{(1-\gamma)^3\epsilon^2}\right)$ | Value-Based | ConRL (Brantley et al., 2020, Remark 3.5) | $\mathcal{O}\left(\frac{|\mathcal{S}|^2|\mathcal{A}|}{(1-\gamma)^5\epsilon^2}\right)$ |
| Value-Based | OptDual-CMDP (Efroni et al., 2020) | $\mathcal{O}\left(\frac{|\mathcal{S}|^2|\mathcal{A}|}{(1-\gamma)^3\epsilon^2}\right)$ | Value-Based | CSPDA[4] (Bai et al., 2021) | $\mathcal{O}\left(\frac{|\mathcal{S}|^2|\mathcal{A}|^2}{(1-\gamma)^7\epsilon^4}\right)$ |
| Value-Based | UC-CFH[1] (Kalagarla et al., 2021, Theorem 1) | $\mathcal{O}\left(\frac{|\mathcal{S}|^3|\mathcal{A}|}{(1-\gamma)^3\epsilon^2}\right)$ | Value-Based | Triple-Q[5] (Wei et al., 2021) | $\mathcal{O}\left(\frac{|\mathcal{S}|^{2.5}|\mathcal{A}|^{2.5}}{(1-\gamma)^{18.5}\epsilon^5}\right)$ |
| Value-Based | OptPess-PrimalDual (Liu et al., 2021) | $\tilde{\mathcal{O}}\left(\frac{|\mathcal{S}|^3|\mathcal{A}|}{(1-\gamma)^4\epsilon^2}\right)$ | Value-Based | Reward-Free CRL[6] (Miryoosefi & Jin, 2021, Theorem 7) | $\mathcal{O}\left(\frac{|\mathcal{S}||\mathcal{A}|}{(1-\gamma)^4\epsilon^2}\right)$ |
| Value-Based | OPDOP (Ding et al., 2021, Theorem 1) | $\tilde{\mathcal{O}}\left(\frac{|\mathcal{S}|^2|\mathcal{A}|}{(1-\gamma)^4\epsilon^2}\right)$ | Policy-Based | CRPO[7] (Xu et al., 2021, Theorem 1) | $\mathcal{O}\left(\frac{|\mathcal{S}||\mathcal{A}|}{(1-\gamma)^7\epsilon^4}\right)$ |
| Value-Based | UCBVI-$\gamma$[2] (He et al., 2021, Theorem 4.3) | $\mathcal{O}\left(\frac{|\mathcal{S}||\mathcal{A}|}{(1-\gamma)^3\epsilon^2}\right)$ | Policy-Based | On-Line NPD-PG[8] (Zeng et al., 2021, Theorem 1) | $\tilde{\mathcal{O}}\left(\frac{|\mathcal{S}|^6|\mathcal{A}|^6}{(1-\gamma)^{12}\epsilon^6}\right)$ |
| Policy-Based | NPD-PG[3] (Ding et al., 2020, Theorem 1) | $\mathcal{O}\left(\frac{1}{(1-\gamma)^4\epsilon^2}\right)$ | Policy-Based | NPD-PG[3] (Ding et al., 2020, Theorem 4) | $\tilde{\mathcal{O}}\left(\frac{|\mathcal{S}|^2|\mathcal{A}|^2}{(1-\gamma)^4\epsilon^2}\right)$ |
| Policy-Based | **PD-PG (This Work, Algorithm 1)** | $\tilde{\mathcal{O}}\left(\frac{|\mathcal{S}|}{(1-\gamma)^4\epsilon^2}\right)$ | Policy-Based | **PD-PG (This Work, Algorithm 3)** | $\tilde{\mathcal{O}}\left(\frac{|\mathcal{S}|^2|\mathcal{A}|}{(1-\gamma)^4\epsilon^2}\right)$ |

Table 1: Typical exact gradient and model-free state-of-the-art algorithms for constrained RL.

## 2 PRELIMINARIES

**Constrained Reinforcement Learning.** Constrained RL is often formulated as a constrained Markov decision process (CMDP), which is the standard Markov decision process (MDP) $\mathcal{M}$ augmented with an additional constraint set $\mathcal{C}$. A MDP is a tuple $\mathcal{M} = (\mathcal{S}, \mathcal{A}, \mathbb{P}, r, \rho_0, \gamma)$. Here $\mathcal{S}$ is state space, $\mathcal{A}$ is action space. $\mathbb{P}(s'|s, a)$ is probability of state transition from $s$ to $s'$ after playing $a$. $r(s'|s, a)$ denotes the reward that the agent observes when state transition from $s$ to $s'$ after it plays $a$, and it is bounded as $|r(\cdot)| \leq 1$. $\rho_0(\cdot) : \mathcal{S} \to [0, 1]$ is the initial state distribution and $\gamma \in (0, 1)$. The policy $\pi(a|s)$ denotes the probability of playing $a$ in state $s$, and $\Pi_{\mathsf{S}}$ denotes the set of all stationary policies. $\mathbb{P}_\pi(s'|s)$ denotes one-step state transformation probability from $s$ to $s'$ by executing $\pi$. Let $\mathcal{T} = \{s_t, a_t, r_{t+1}\}_{t \geq 0} \sim \pi$ be a trajectory generated by $\pi$, where $s_0 \sim \rho_0(\cdot)$, $a_t \sim \pi(\cdot|s_t)$, $s_{t+1} \sim \mathbb{P}(\cdot|s_t, a_t)$, and $r_{t+1} = r(s_{t+1}|s_t, a_t)$. Let $d_\pi^{s_0}(s) = (1 - \gamma)\sum_{t=0}^\infty \gamma^t \mathbb{P}_\pi(s_t = s|s_0)$ be the state distribution of the Markov chain (starting at $s_0$) induced by policy $\pi$, and $d_\pi^{\rho_0}(s) = \mathbb{E}_{s_0 \sim \rho_0(\cdot)}[d_\pi^{s_0}(s)]$. Let $V_\pi(s) = \mathbb{E}_\pi[\sum_{t=0}^\infty \gamma^t r_{t+1}|s_0 = s]$ be the *state value function*. The *state-action value function* is $Q_\pi(s, a) = \mathbb{E}_\pi[\sum_{t=0}^\infty \gamma^t r_{t+1}|s_0 = s, a_0 = a]$, and advantage function is $A_\pi(s, a) = Q_\pi(s, a) - V_\pi(s)$. Finally, we define the objective function $J(\pi) = \mathbb{E}_{s \sim \rho_0(\cdot)}[V_\pi(s)]$.

CMDP extends MDP with an additional constraint set $\mathcal{C} = \{(c_i, b_i)\}_{i=1}^m$, where $c_i : \mathcal{S} \times \mathcal{A} \to \mathbb{R}$ is cost function, each $b_i$ is cost limit, and $|c_i(\cdot)| \leq 1$. We define value functions $V_\pi^{c_i}$, action-value functions $Q_\pi^{c_i}$, and advantage functions $A_\pi^{c_i}$ for costs in analogy to $V_\pi, Q_\pi$, and $A_\pi$, with $c_i$ replacing $r$ respectively, $i \in \{1, 2 \cdots, m\}$, e.g., $V_\pi^{c_i}(s) = \mathbb{E}_\pi[\sum_{t=0}^\infty \gamma^t c_i(s_t, a_t)|s_0 = s]$. Furthermore, we define the expected cost $C_i(\pi) = \mathbb{E}_{s \sim \rho_0(\cdot)}[V_\pi^{c_i}(s)]$. The feasible policy set $\Pi_\mathcal{C}$ is defined as follows, $\Pi_\mathcal{C} = \cap_{i=1}^m \{\pi \in \Pi_{\mathsf{S}} \text{ and } C_i(\pi) \leq b_i\}$. The goal of safe RL is to search a policy $\pi$ satisfies

$$\max_{\pi \in \Pi_{\mathsf{S}}} J(\pi), \text{ such that } \mathbf{c}(\pi) \preceq \mathbf{b}, \tag{3}$$

where the vector $\mathbf{c}(\pi) = (C_1(\pi), C_2(\pi), \cdots, C_m(\pi))^\top$, and $\mathbf{b} = (b_1, b_2, \cdots, b_m)^\top$. If the constrained policy optimization problem (3) exists a solution, we denote it as:

$$\pi_\star = \arg \max_{\pi \in \Pi_\mathcal{C}} J(\pi). \tag{4}$$

---

[1]According to Bai et al. (2021), (Kalagarla et al., 2021, Theorem 1) involves a constant $C$ bounded by $|\mathcal{S}|$.

[2]UCBVI-$\gamma$ matches the lower bound $\tilde{\Omega}\left(\frac{|\mathcal{S}||\mathcal{A}|}{(1-\gamma)^3\epsilon^2}\right)$ for MDP (Lattimore & Hutter, 2012; Azar et al., 2013).

[3]Theorem 1 of Ding et al. (2020) shows a convergence rate independent on $\mathcal{S}$ and $\mathcal{A}$. Notice that in Theorem 4 of Ding et al. (2020), $|\mathcal{S}|^2|\mathcal{A}|^2$ samples are necessary for the two outer loops.

[4]Bai et al. (2021) claim CSPDA needs $\mathcal{O}(\frac{|\mathcal{S}||\mathcal{A}|}{(1-\gamma)^4\epsilon^2})$, but the inner loop of their Algorithm 1 needs an additional generative model that needs $\frac{1}{(1-\gamma)^3} \frac{|\mathcal{S}||\mathcal{A}|\log(|\mathcal{S}||\mathcal{A}|)}{\epsilon^2}$ samples (Agarwal et al., 2020a, Chapter 2).

[5]We show this iteration complexity according to a recent work (Bai et al., 2021). Since Wei et al. (2021) study the finite-horizon CMDP, we believe their Triple-Q plays at least $\mathcal{O}(\frac{|\mathcal{S}|^2|\mathcal{A}|^2}{\epsilon^5})$.

[6]The worst-case of constraint violation shown in (Miryoosefi & Jin, 2021) reaches $\mathcal{O}\left(\frac{|\mathcal{S}|^2|\mathcal{A}|}{(1-\gamma)^4\epsilon^2}\right)$ if the number of constraint function is large than $|\mathcal{S}|$.

[7]Notice that the inner loop with $K_{\text{in}} = \mathcal{O}(\frac{T}{(1-\gamma)|\mathcal{S}||\mathcal{A}|})$ iteration is needed (Xu et al., 2021, Theorem 3).

[8]We show the iteration complexity after some simple algebra according to (Zeng et al., 2021, Lemma 8-9).

**Strong Duality.** Let $\boldsymbol{\lambda} \in \mathbb{R}^m$, and $\boldsymbol{\lambda} \succeq \mathbf{0}$, the Lagrange multiplier function $\mathcal{L}(\pi, \boldsymbol{\lambda})$ is defined as:

$$\mathcal{L}(\pi, \boldsymbol{\lambda}) = J(\pi) + \boldsymbol{\lambda}^\top \left( \mathbf{b} - \mathbf{c}(\pi) \right). \tag{5}$$

Its associated dual function is defined as: $\mathcal{L}_{\mathrm{D}}(\boldsymbol{\lambda}) = \max_{\pi \in \Pi_{\mathrm{S}}} \mathcal{L}(\pi, \boldsymbol{\lambda})$, its optimal dual parameter is:

$$\boldsymbol{\lambda}_\star = \arg \min_{\boldsymbol{\lambda} \succeq \mathbf{0}} \mathcal{L}_{\mathrm{D}}(\boldsymbol{\lambda}), \quad \text{i.e.,} \quad \mathcal{L}_{\mathrm{D}}(\boldsymbol{\lambda}_\star) = \min_{\boldsymbol{\lambda} \succeq \mathbf{0}} \max_{\pi \in \Pi_{\mathrm{S}}} \mathcal{L}(\pi, \boldsymbol{\lambda}). \tag{6}$$

**Assumption 1** (Slater Condition). *There exists a vector $\boldsymbol{\xi} \prec \mathbf{0}$, and a policy $\bar{\pi} \in \Pi_{\mathrm{S}}$ such that*

$$\mathbf{c}(\bar{\pi}) - \mathbf{b} \preceq \boldsymbol{\xi}. \tag{7}$$

The Slater condition (Slater, 1950) is mild in practice (otherwise, we can simply increase the constraint vector $\mathbf{b}$ by a tiny amount), and Slater condition is a standard assumption for CMDP appears in the previous work (Chow et al., 2018; Paternain et al., 2019a;b; Le et al., 2019; Ding et al., 2020; Ying et al., 2021). Slater's condition and convexity of policy class $\Pi_{\mathrm{S}}$ ensure that strong duality holds. We formulate the problem (3) as the following strong duality version.

**Theorem 1** (Strong Duality (Altman, 1999)). *Let the stationary policy space $\Pi_{\mathrm{S}}$ be a convex set. Under Assumption 1, the CMDP problem (3) shares the same optimal solution as the following min-max and max-min problems*

$$J(\pi_\star) = \min_{\boldsymbol{\lambda} \succeq \mathbf{0}} \max_{\pi \in \Pi_{\mathrm{S}}} \mathcal{L}(\pi, \boldsymbol{\lambda}) = \max_{\pi \in \Pi_{\mathrm{S}}} \min_{\boldsymbol{\lambda} \succeq \mathbf{0}} \mathcal{L}(\pi, \boldsymbol{\lambda}). \tag{8}$$

**Policy Gradient with Softmax Policy.** In this paper, we mainly consider softmax policy:

$$\pi_{\boldsymbol{\theta}}(a|s) = \frac{\exp\{\theta_{s,a}\}}{\sum_{\tilde{a} \in \mathcal{A}} \exp\{\theta_{s,\tilde{a}}\}}, \ \forall\, (s, a) \in \mathcal{S} \times \mathcal{A}, \tag{9}$$

where $\boldsymbol{\theta} \in \mathbb{R}^{|\mathcal{S}| \times |\mathcal{A}|}$, and each $\boldsymbol{\theta}[s, a] := \theta_{s,a}$. Finally, we define two additional notations:

$$\mathbf{a}_{\pi_{\boldsymbol{\theta}}}^c(s, a) = \left( A_{\pi_{\boldsymbol{\theta}}}^{c_1}(s, a), A_{\pi_{\boldsymbol{\theta}}}^{c_2}(s, a), \cdots, A_{\pi_{\boldsymbol{\theta}}}^{c_m}(s, a) \right)^\top, \quad A_{\pi_{\boldsymbol{\theta}}}(s, a, \boldsymbol{\lambda}) = A_{\pi_{\boldsymbol{\theta}}}(s, a) - \boldsymbol{\lambda}^\top \mathbf{a}_{\pi_{\boldsymbol{\theta}}}^c(s, a).$$

**Proposition 1.** *Let $\pi_{\boldsymbol{\theta}}$ be softmax policy (9), the gradient of $J(\pi_{\boldsymbol{\theta}})$ and $C_i(\pi_{\boldsymbol{\theta}})$ with respect to $\boldsymbol{\theta}$ is:*

$$\frac{\partial J(\pi_{\boldsymbol{\theta}})}{\partial \theta_{s,a}} = \frac{1}{1 - \gamma} d_{\pi_{\boldsymbol{\theta}}}^{\rho_0}(s) \pi_{\boldsymbol{\theta}}(a|s) A_{\pi_{\boldsymbol{\theta}}}(s, a), \quad \frac{\partial C_i(\pi_{\boldsymbol{\theta}})}{\partial \theta_{s,a}} = \frac{1}{1 - \gamma} d_{\pi_{\boldsymbol{\theta}}}^{\rho_0}(s) \pi_{\boldsymbol{\theta}}(a|s) A_{\pi_{\boldsymbol{\theta}}}^{c_i}(s, a). \tag{10}$$

*Then the gradients of $\mathcal{L}(\pi_{\boldsymbol{\theta}}, \boldsymbol{\lambda})$ with respect to $\boldsymbol{\theta}, \boldsymbol{\lambda}$ are:*

$$\frac{\partial \mathcal{L}(\pi_{\boldsymbol{\theta}}, \boldsymbol{\lambda})}{\partial \theta_{s,a}} = \frac{1}{1 - \gamma} d_{\pi_{\boldsymbol{\theta}}}^{\rho_0}(s) \pi_{\boldsymbol{\theta}}(a|s) A_{\pi_{\boldsymbol{\theta}}}(s, a, \boldsymbol{\lambda}), \quad \frac{\partial \mathcal{L}(\pi_{\boldsymbol{\theta}}, \boldsymbol{\lambda})}{\partial \boldsymbol{\lambda}} = \mathbf{b} - \mathbf{c}(\pi_{\boldsymbol{\theta}}). \tag{11}$$

## 3 PRIMAL-DUAL POLICY GRADIENT METHOD

According to strong duality shown in Theorem 1, to solve the constrained RL problem (3), we only need to solve the equivalent unconstrained problem (8). We define primal-dual approach as follows,

$$\boldsymbol{\lambda}_{t+1} \leftarrow \left\{ \boldsymbol{\lambda}_t - \eta \nabla_{\boldsymbol{\lambda}} \mathcal{L}(\pi_{\boldsymbol{\theta}_t}, \boldsymbol{\lambda}_t) \right\}_+, \quad \boldsymbol{\theta}_{t+1} \leftarrow \boldsymbol{\theta}_t + \eta \nabla_{\boldsymbol{\theta}} \mathcal{L}(\pi_{\boldsymbol{\theta}_t}, \boldsymbol{\lambda}_t), \tag{12}$$

where the elements of $\nabla_{\boldsymbol{\theta}} \mathcal{L}(\pi_{\boldsymbol{\theta}_t}, \boldsymbol{\lambda}_t)$ and $\nabla_{\boldsymbol{\lambda}} \mathcal{L}(\pi_{\boldsymbol{\theta}_t}, \boldsymbol{\lambda}_t)$ are shown in Proposition 1, $\{\cdot\}_+$ denotes the positive part operator, i.e., if $x \le 0$, $\{x\}_+ = 0$, else $\{x\}_+ = x$, and $\eta > 0$ is step-size. The complete primal-dual approach has been shown in Algorithm 1, where we introduce a notation $\mathbf{G}(\boldsymbol{\theta}, \boldsymbol{\lambda}) \in \mathbb{R}^{|\mathcal{S}| \times |\mathcal{A}|}$ that is the matrix version of $\frac{\partial \mathcal{L}(\pi_{\boldsymbol{\theta}}, \boldsymbol{\lambda})}{\partial \theta_{s,a}}$, i.e., each $\mathbf{G}(\boldsymbol{\theta}, \boldsymbol{\lambda})[s, a]$ is defined as:

$$\mathbf{G}(\boldsymbol{\theta}, \boldsymbol{\lambda})[s, a] = \frac{1}{1 - \gamma} d_{\pi_{\boldsymbol{\theta}}}^{\rho_0}(s) \pi_{\boldsymbol{\theta}}(a|s) A_{\pi_{\boldsymbol{\theta}}}(s, a, \boldsymbol{\lambda}).$$

Before we show the convergence rate of Algorithm 1, we assume that the initial state distribution $\rho_0(\cdot)$ used in the gradient updates is bounded away from zero.

**Assumption 2** (Sufficient Exploration). *The initial state distribution $\rho_0(\cdot)$ satisfies*

$$\rho_{\min} := \min_{s \in \mathcal{S}} \left\{ \rho_0(s) \right\} > 0. \tag{13}$$

---

**Algorithm 1** Primal-Dual Policy Gradient (PD-PG)

---

**Initialization**: step-size $\eta$, $\boldsymbol{\theta}_0 = \mathbf{0}$, $\boldsymbol{\lambda}_0 = \mathbf{0}$, policy gradient $\mathbf{G}(\boldsymbol{\theta}, \boldsymbol{\lambda})$;
**for** $t = 0, 1, \cdots, T - 1$ **do**
$\quad \mathbf{G}(\boldsymbol{\theta}_t, \boldsymbol{\lambda}_t)[s, a] = \frac{1}{1-\gamma} d_{\pi_{\boldsymbol{\theta}_t}}^{\rho_0}(s) \pi_{\boldsymbol{\theta}_t}(a|s) A_{\pi_{\boldsymbol{\theta}_t}}(s, a, \boldsymbol{\lambda}_t)$;
$\quad \boldsymbol{\lambda}_{t+1} \leftarrow \left\{ \boldsymbol{\lambda}_t - \eta (\mathbf{b} - \mathbf{c}(\pi_{\boldsymbol{\theta}_t})) \right\}_+$;
$\quad \boldsymbol{\theta}_{t+1} \leftarrow \boldsymbol{\theta}_t + \eta \mathbf{G}(\boldsymbol{\theta}_t, \boldsymbol{\lambda}_t)$;
**end for**

---

Assumption 2 has been adapted by Agarwal et al. (2020b); Mei et al. (2020); Ying et al. (2021), which requires the initial distribution $\rho_0(\cdot)$ lies in the interior of the probability simplex $\Delta^{\circ}(\mathcal{S}) := \{p_s | p_s > 0, \sum_{s \in \mathcal{S}} p_s = 1\}$. The condition (13) ensures "*sufficient exploration*", which means that for any policy $\pi \in \Pi_{\mathbf{s}}$, the distribution $d_{\pi}^{\rho_0}(s)$ keeps positive over the state space $\mathcal{S}$. Additionally, Assumption 2 is necessary for global optimality of policy gradient methods. Concretely, Mei et al. (2020) have shown that there exists an MDP with the condition $\min_{s \in \mathcal{S}} \rho_0(s) = 0$, and there exists a parameter $\boldsymbol{\theta}_\star$ such that this $\boldsymbol{\theta}_\star$ is a stationary policy of $J(\pi_{\boldsymbol{\theta}})$, but $\pi_{\boldsymbol{\theta}_\star}$ is not an optimal policy.

To short the expression, we define some additional notations. Recall $\boldsymbol{\xi} := (\xi_1, \xi_2, \cdots, \xi_m)^\top \prec \mathbf{0}$ defined in Assumption 1, i.e., $\xi_i < 0, i = 1, 2, \cdots, m$, let $\iota := \min_{1 \le i \le m} \{-\xi_i\}$, $\varrho := 1 + \frac{2m}{(1-\gamma)^2 \iota^2}$. The distribution mismatch coefficient is defined as: $\left\| \frac{d_{\pi_\star}^{\rho_0}}{\rho_0} \right\|_\infty := \max_{s \in \mathcal{S}} \left\{ \frac{d_{\pi_\star}^{\rho_0}(s)}{\rho_0(s)} \right\}$. Finally, let

$$\pi_t(a|s) := \pi_{\boldsymbol{\theta}_t}(a|s), \text{ and } \pi_t := \pi_{\boldsymbol{\theta}_t}. \tag{14}$$

From stating this and the remaining results, we fix a *deterministic optimal policy* $\pi_\star(\cdot|s)$, denote it as $a_\star(s)$, i.e., $\pi_\star(a_\star(s)|s) = 1$; if $a \neq a_\star(s)$, $\pi_\star(a|s) = 0$.

**Proposition 2.** *Under Assumption 2, updating $\pi_t$ according to Algorithm 1, we obtain*

$$c_\star =: \inf_{s \in \mathcal{S}, t \ge 1} \{\pi_t(a_\star(s)|s)\} > 0. \tag{15}$$

We provide the proof in Appendix D.1 (see Lemma 15). According to (15), $c_\star$ is a positive scalar independent on the time-step $t$ and independent on the state space $\mathcal{S}$.

**Theorem 2.** *Under Assumption 1-2, $\pi_{\boldsymbol{\theta}}$ is the softmax policy defined in (9). Let time-step $T$ satisfy*

$$T \ge \max \left\{ \frac{1}{(1-\gamma)^2}, \frac{1}{((1-\gamma)^2 + 2m/\iota^2)^2} \right\} \cdot \frac{D_2^2}{|\mathcal{S}| \log |\mathcal{A}| \rho_{\min}^2 D_1}, \tag{16}$$

*where $D_1$ and $D_2$ are positive scalars will be special later. The sequence $\{\boldsymbol{\lambda}_t, \boldsymbol{\theta}_t\}_{t=0}^{T-1}$ is generated by Algorithm 1. Let $\eta = \sqrt{\left\| \frac{d_{\pi_\star}^{\rho_0}}{\rho_0} \right\|_\infty \frac{|\mathcal{S}| \log |\mathcal{A}|}{C(1-\gamma)T}}$, $\beta = \sqrt{\left\| \frac{d_{\pi_\star}^{\rho_0}}{\rho_0} \right\|_\infty \frac{4|\mathcal{S}| \log |\mathcal{A}|}{(1-\gamma)^3 \iota^2 c_\star}}$, where $C$ is a positive scalar will be special later. Then for all $i \in \{1, 2, \cdots, m\}$, $\pi_t := \pi_{\boldsymbol{\theta}_t}$ satisfies*

$$\min_{t < T} \{J(\pi_\star) - J(\pi_t)\} \le 4 \left\| \frac{d_{\pi_\star}^{\rho_0}}{\rho_0} \right\|_\infty \sqrt{\frac{|\mathcal{S}| \log |\mathcal{A}|}{c_\star (1-\gamma)^4 T}}, \tag{17}$$

$$\min_{t < T} \{C_i(\pi_t) - b_i\}_+ \le \frac{4\varrho}{\beta - \|\boldsymbol{\lambda}_\star\|_\infty} \left\| \frac{d_{\pi_\star}^{\rho_0}}{\rho_0} \right\|_\infty \sqrt{\frac{|\mathcal{S}| \log |\mathcal{A}|}{c_\star (1-\gamma)^4 T}}. \tag{18}$$

**Remark 1.** *Lemma 3 (see Appendix B.3) has shown the boundedness of $\boldsymbol{\lambda}_\star$ as follows: $\|\boldsymbol{\lambda}_\star\|_\infty \le \frac{2}{(1-\gamma)\iota}$. Furthermore, according to the discussions in Remark 3 (see Appendix D.2), the inequality $\beta > \frac{2}{(1-\gamma)\iota}$ always holds. Thus $\beta > \|\boldsymbol{\lambda}_\star\|_\infty$, which implies the bounds (17) and (18) are well-defined.*

**Remark 2.** *Theorem 2 implies Algorithm 1 needs the iteration complexity of*

$$\mathcal{O} \left( \left\| \frac{d_{\pi_\star}^{\rho_0}}{\rho_0} \right\|_\infty^2 \frac{|\mathcal{S}| \log |\mathcal{A}|}{c_\star (1-\gamma)^4 \epsilon^2} \right) \tag{19}$$

*to obtain $\mathcal{O}(\epsilon)$-optimality. The iteration complexity of (19) is a function with respect to the toleration $\epsilon$, which matches the best known policy gradient methods from (Ding et al., 2020) for CMDP, where both NPD-PG (Ding et al., 2020) and the proposed PD-PG share a complexity of $\mathcal{O}(\epsilon^{-2})$.*

## 4 PRIMAL-DUAL METHOD TO SOLVE MODEL-FREE CONSTRAINED RL

The main difficulty to implementing a model-free algorithm lies in designing an efficient policy gradient estimator for the discounted infinite-horizon MDP, which is intractable for sampling-based policy optimization since it requires a trajectory with an infinite horizon, which is impossible for practical simulation. In Section 4.2-4.3, we present an unbiased value function and policy gradient estimators with finite horizon trajectory, which is the benchmark for us to propose sample-based algorithms. The proposed algorithm and convergence analysis lie in Section 4.4-4.5.

### 4.1 DILEMMA IN MONTE-CARLO ROLLOUT

Recall Proposition 1, to obtain unbiased estimators of $\frac{\partial J(\pi_{\boldsymbol{\theta}})}{\partial \theta_{s,a}}$ and $\frac{\partial C_i(\pi_{\boldsymbol{\theta}})}{\partial \theta_{s,a}}$, it is necessary to satisfy the following two conditions:

- **(C1):** draw the state-action pair $(s, a)$ according to: $(s, a) \sim \left(d_{\pi_{\boldsymbol{\theta}}}^{\rho_0}(\cdot), \pi_{\boldsymbol{\theta}}(\cdot|s)\right)$;
- **(C2):** obtain the unbiassed estimator of the advantage functions $A_{\pi_{\boldsymbol{\theta}}}(s, a)$ and $A_{\pi_{\boldsymbol{\theta}}}^{c_i}(s, a)$.

However, we can not obtain exact $d_{\pi_{\boldsymbol{\theta}}}^{s_0}(s) = \mathbb{E}_{s_0 \sim \rho_0(\cdot)}[(1 - \gamma) \sum_{t=0}^{\infty} \gamma^t \mathbb{P}_{\pi_{\boldsymbol{\theta}}}(s_t = s|s_0)]$ for model-free RL, since the transformation probability $\mathbb{P}_{\pi_{\boldsymbol{\theta}}}(s_t = s|s_0)$ is unknown. Additionally, Monte-Carlo rollout is a theoretically possible but intractable sampling-based approach to obtain an unbiased estimator of $A_{\pi_{\boldsymbol{\theta}}}(s, a)$, which requires us to run infinite-horizon trajectories to estimate the value functions. For example, let $\{(s_t, a_t, r(s_t, a_t))\}_{t \geq 0}^{\infty} \sim \pi_{\boldsymbol{\theta}}$ start from $(s_0, a_0) = (s, a)$,

$$\widehat{Q}(s, a) = \sum_{t=0}^{\infty} \gamma^t r(s_t, a_t) \tag{20}$$

is an unbiased estimators of $Q_{\pi_{\boldsymbol{\theta}}}(s, a)$. Despite the unbiasedness of $\widehat{Q}(s, a)$, Monte-Carlo rollout (20) requires infinite number of horizons, which is impossible in practice.

### 4.2 UNBIASED VALUE FUNCTION ESTIMATOR WITH FINITE HORIZON TRAJECTORY

Both of the conditions **(C1)** and **(C2)** can be implemented via a geometric random variable horizon during the simulated process (Paternain, 2018, Chapter 6). Now, we present the insights behind this process, which requires us to master geometric distribution Geo$(\cdot)$, see Appendix E.1.

Recall the Monte-Carlo rollout (20), if $\gamma \approx 0$, then infinite series $\widehat{Q}(s, a)$ (20) prioritizes the present reward information. In that sense, when $\gamma$ is very small, we do not need to require the agent to evolve to collect the future reward for a long time. On the contrary, if $\gamma \approx 1$, we need to require the agent to look far away into the future reward information. The geometric distribution provides us a way to formulate this idea (Paternain, 2018). Concretely, let $\tau \sim$ Geo$(1 - \gamma)$, and rollout a finite horizon trajectory as $\mathcal{D}_{\tau} = \{s_t, a_t, r(s_t, a_t)\}_{t=0:\tau} \sim \pi$, where the initial state-action $(s_0, a_0) = (s, a)$. Then, we define an estimator of $Q_{\pi}(s, a)$ according to the sum of reward along the trajectory $\mathcal{D}_{\tau}$:

$$\widehat{Q}_{\pi}(s, a) = \sum_{t=0}^{\tau} r(s_t, a_t). \tag{21}$$

This $\widehat{Q}_{\pi}(s, a)$ unbiasedly estimates $Q_{\pi}(s, a)$ for each $(s, a)$. Such a programming (21) can be extended to cost function if we use $c_i(\cdot)$ to replace $r(\cdot)$ respectively, and can be extended to $V_{\pi}(s)$ and $V_{\pi}^{c_i}(s)$ if $\mathcal{D}_{\tau}$ starts from $s$.

Due to the limitation of space, we have provided the details of implementation to estimate $Q_{\pi}$ and $V_{\pi}$ in Algorithm 4 (denoted as $\text{EstQ}(\pi, g, s, a)$) and Algorithm 5 (denoted as $\text{EstV}(\pi, g, s, a)$), see Appendix E.2, where we denote it as $\text{EstQ}(\pi, g, s, a)$, where $g = r(\cdot)$ or $g = c(\cdot)$. The next Proposition 3-4 show that Algorithm 4 and Algorithm 5 output unbiased estimators for value functions. We have provided the proof in Appendix E.3.

**Proposition 3** (Unbiasedness of Algorithm 4). *Let* $\widehat{Q}_{\pi}(s, a) = \text{EstQ}(\pi, r, s, a), \widehat{Q}_{\pi}^{c_i}(s, a) = \text{EstQ}(\pi, c_i, s, a)$, *then the following holds:* $\mathbb{E}[\widehat{Q}_{\pi}(s, a)] = Q_{\pi}(s, a), \mathbb{E}[\widehat{Q}_{\pi}^{c_i}(s, a)] = Q_{\pi}^{c_i}(s, a)$.

**Proposition 4** (Unbiasedness of Algorithm 5). *Let* $\widehat{V}_{\pi}(s) = \text{EstV}(\pi, r, s), \widehat{V}_{\pi}^{c_i}(s) = \text{EstV}(\pi, c_i, s)$, *then the following holds:* $\mathbb{E}[\widehat{V}_{\pi}(s)] = V_{\pi}(s), \mathbb{E}[\widehat{V}_{\pi}^{c_i}(s, a)] = V_{\pi}^{c_i}(s)$.

---

**Algorithm 2** $\mathrm{PG}(\pi, g, s, a)$: Estimate Policy Gradient

1: **Input**: Policy $\pi_{\boldsymbol{\theta}}$; Reward function or Cost function $g(\cdot, \cdot)$; State-action pair $(s, a)$;
2: **First Rollout**: $\widehat{Q}(s, a) = \mathrm{EstQ}(\pi_{\boldsymbol{\theta}}, g, s, a)$, let $\tau \sim \mathrm{Geo}(1 - \gamma)$ denote the terminal time;
3: **Second Rollout**: $\widehat{Q}(s_\tau, a_\tau) = \mathrm{EstQ}(\pi_{\boldsymbol{\theta}}, g, s_\tau, a_\tau)$, let $\tau' \sim \mathrm{Geo}(1 - \gamma)$ denote the terminal time; collect the trajectory $\{(s'_j, a'_j, g(s'_j, a'_j))\}_{j=0:\tau'}$, where the initial $(s'_0, a'_0) = (s_\tau, a_\tau)$;
4: **Output**: $\widehat{G}(s, a) = \dfrac{1}{1 - \gamma} \widehat{Q}(s_\tau, a_\tau) \dfrac{\partial \log \pi_{\boldsymbol{\theta}}(a_\tau | s_\tau)}{\partial \theta_{s,a}}$.

---

### 4.3 Unbiased Policy Gradient Estimator

Now, we introduce an unbiased policy gradient estimator, which involves two rollouts.

**First Rollout**: we play a rollout with respect to $\pi_{\boldsymbol{\theta}}$ according to Algorithm 4,

$$\widehat{Q}_{\pi_{\boldsymbol{\theta}}}(s, a) = \mathrm{EstQ}(\pi_{\boldsymbol{\theta}}, r, s, a), \tag{22}$$

we use $\tau \sim \mathrm{Geo}(1 - \gamma)$ to denote the finite terminal time of the horizon of the rollout (22).

**Second Rollout**: we play a rollout for the last state-action pair $(s_\tau, a_\tau)$ according to Algorithm 4,

$$\widehat{Q}_{\pi_{\boldsymbol{\theta}}}(s_\tau, a_\tau) = \mathrm{EstQ}(\pi_{\boldsymbol{\theta}}, r, s_\tau, a_\tau). \tag{23}$$

Let $\tau' \sim \mathrm{Geo}(1 - \gamma)$ be the terminal time of the rollout (23), and we denote it as $\mathcal{D}' = \left\{ s'_j, a'_j, r(s'_j, a'_j) \right\}_{j=0:\tau'}$, where the initial state-action pair $(s'_0, a'_0) = (s_\tau, a_\tau)$.

**Output**: let $\widehat{G}_{\pi_{\boldsymbol{\theta}}}(s, a)$ be an estimator defined as follows,

$$\widehat{G}_{\pi_{\boldsymbol{\theta}}}(s, a) = \frac{1}{1 - \gamma} \widehat{Q}_{\pi_{\boldsymbol{\theta}}}(s_\tau, a_\tau) \frac{\partial \log \pi_{\boldsymbol{\theta}}(a_\tau | s_\tau)}{\partial \theta_{s,a}}. \tag{24}$$

Since return objective $J(\pi_{\boldsymbol{\theta}})$ and cost function $C_i(\pi_{\boldsymbol{\theta}})$ share a similar structure, all the estimators (22)-(24) can be extended to $C_i(\pi_{\boldsymbol{\theta}})$ if we use $c_i$ to replace $r$ respectively. We have provided such policy gradient estimator with finite horizon trajectory in Algorithm 2, and denote it as $\mathrm{PG}(\pi, g, s, a)$.

**Theorem 3.** *Let $\pi_{\boldsymbol{\theta}}$ be the softmax policy (9), and $\widehat{G}_{\pi_{\boldsymbol{\theta}}}(s, a) = \mathrm{PG}(\pi_{\boldsymbol{\theta}}, r, s, a), \widehat{G}_{\pi_{\boldsymbol{\theta}}}^{c_i}(s, a) = \mathrm{PG}(\pi_{\boldsymbol{\theta}}, c_i, s, a)$, Then $\widehat{G}_{\pi_{\boldsymbol{\theta}}}(s, a)$ and $\widehat{G}_{\pi_{\boldsymbol{\theta}}}^{c_i}(s, a)$ satisfy*

$$\mathbb{E}[\widehat{G}_{\pi_{\boldsymbol{\theta}}}(s, a)] = \frac{\partial J(\pi_{\boldsymbol{\theta}})}{\partial \theta_{s,a}}, \quad \mathbb{E}[\widehat{G}_{\pi_{\boldsymbol{\theta}}}^2(s, a)] \leq \frac{4}{(1 - \gamma)^3};$$

$$\mathbb{E}[\widehat{G}_{\pi_{\boldsymbol{\theta}}}^{c_i}(s, a)] = \frac{\partial C_i(\pi_{\boldsymbol{\theta}})}{\partial \theta_{s,a}}, \quad \mathbb{E}[(\widehat{G}_{\pi_{\boldsymbol{\theta}}}^{c_i}(s, a))^2] \leq \frac{4}{(1 - \gamma)^3}.$$

Theorem 3 has guaranteed the unbiasedness and boundedness of the estimator $\mathrm{PG}(\pi, g, s, a)$, which is the benchmark for us to show theoretical results. We provide its proof in Appendix E.4.

### 4.4 Model-Free Algorithm Derivation

We have shown model-free PD-PG in Algorithm 3. To show a stochastic primal-dual implementation, the iteration (12) implies that we need to estimate $\frac{\partial \mathcal{L}(\pi_{\boldsymbol{\theta}}, \boldsymbol{\lambda})}{\partial \boldsymbol{\lambda}}$ and $\frac{\partial \mathcal{L}(\pi_{\boldsymbol{\theta}}, \boldsymbol{\lambda})}{\partial \boldsymbol{\theta}}$.

**Estimator for** $\frac{\partial \mathcal{L}(\pi_{\boldsymbol{\theta}}, \boldsymbol{\lambda})}{\partial \boldsymbol{\lambda}}$. We obtain the estimators of the cost value function: $\widehat{V}_{\pi_{\boldsymbol{\theta}}}^{c_i}(s) = \mathrm{EstV}(\pi_{\boldsymbol{\theta}}, c_i, s)$. Furthermore, let $\widehat{C}_i(\pi_{\boldsymbol{\theta}}) = \sum_{s \in \mathcal{S}} \rho_0(s) \widehat{V}_{\pi_{\boldsymbol{\theta}}}^{c_i}(s)$, and

$$\hat{\mathbf{c}}(\pi_{\boldsymbol{\theta}}) = (\widehat{C}_1(\pi_{\boldsymbol{\theta}}), \widehat{C}_2(\pi_{\boldsymbol{\theta}}), \cdots, \widehat{C}_m(\pi_{\boldsymbol{\theta}}))^\top, \tag{25}$$

then according to Proposition 4, $\mathbf{b} - \hat{\mathbf{c}}(\pi_{\boldsymbol{\theta}})$ is an unbiased estimator of $\frac{\partial \mathcal{L}(\pi_{\boldsymbol{\theta}}, \boldsymbol{\lambda})}{\partial \boldsymbol{\lambda}}$, i.e., for any given policy parameter $\boldsymbol{\theta}$, the following holds

$$\mathbb{E}[\mathbf{b} - \hat{\mathbf{c}}(\pi_{\boldsymbol{\theta}})] = \frac{\partial \mathcal{L}(\pi_{\boldsymbol{\theta}}, \boldsymbol{\lambda})}{\partial \boldsymbol{\lambda}}.$$

---

**Algorithm 3** Primal-Dual Approach to Model-Free Safe RL

---

1: **Initialization**: step-size $\eta$, $\boldsymbol{\theta}_0 = \mathbf{0}$, and Lagrange multiplier $\boldsymbol{\lambda}_0 = \mathbf{0}$;
2: **for** $t = 0, 1, 2, \cdots, T-1$ **do**
3:     # Estimate $\frac{\partial \mathcal{L}(\pi_{\boldsymbol{\theta}_t}, \boldsymbol{\lambda}_t)}{\partial \boldsymbol{\lambda}_t}$. Obtain cost value estimator $\hat{\mathbf{c}}(\pi_{\boldsymbol{\theta}_t})$ according to (25); Let
4:
$$\widehat{\nabla_{\boldsymbol{\lambda}_t}} \mathcal{L}(\pi_{\boldsymbol{\theta}_t}, \boldsymbol{\lambda}_t) = \mathbf{b} - \hat{\mathbf{c}}(\pi_{\boldsymbol{\theta}_t});$$

5:     # Estimate $\frac{\partial \mathcal{L}(\pi_{\boldsymbol{\theta}_t}, \boldsymbol{\lambda}_t)}{\partial \boldsymbol{\theta}_t}$. Let $\widehat{\mathbf{G}}(\pi_{\boldsymbol{\theta}_t}, \boldsymbol{\lambda}_t)[s,a] = \widehat{G}_{\pi_{\boldsymbol{\theta}_t}}(s,a) - \boldsymbol{\lambda}_t^\top \mathbf{g}_{\pi_{\boldsymbol{\theta}_t}}^c(s,a)$; calculate
$$\widehat{\nabla_{\boldsymbol{\theta}_t}} \mathcal{L}(\pi_{\boldsymbol{\theta}_t}, \boldsymbol{\lambda}_t) = \widehat{\mathbf{G}}(\pi_{\boldsymbol{\theta}_t}, \boldsymbol{\lambda}_t);$$

6:     #Primal-Dual Update for Parameters.
$$\boldsymbol{\lambda}_{t+1} = \{\boldsymbol{\lambda}_t - \eta \widehat{\nabla_{\boldsymbol{\lambda}_t}} \mathcal{L}(\pi_{\boldsymbol{\theta}_t}, \boldsymbol{\lambda}_t)\}_+; \boldsymbol{\theta}_{t+1} = \boldsymbol{\theta}_t + \eta \widehat{\nabla_{\boldsymbol{\theta}_t}} \mathcal{L}(\pi_{\boldsymbol{\theta}_t}, \boldsymbol{\lambda}_t);$$

7: **end for**

---

**Estimator for $\frac{\partial \mathcal{L}(\pi_{\boldsymbol{\theta}}, \boldsymbol{\lambda})}{\partial \boldsymbol{\theta}}$.** According to Algorithm 2, we obtain the policy gradient estimators with respect to $\frac{\partial J(\pi_{\boldsymbol{\theta}})}{\partial \theta_{s,a}}$ and $\frac{\partial C_i(\pi_{\boldsymbol{\theta}})}{\partial \theta_{s,a}}$ as follows,

$$\widehat{G}_{\pi_{\boldsymbol{\theta}}}(s,a) = \mathrm{PG}(\pi_{\boldsymbol{\theta}}, r, s, a), \quad \widehat{G}_{\pi_{\boldsymbol{\theta}}}^{c_i}(s,a) = \mathrm{PG}(\pi_{\boldsymbol{\theta}}, c_i, s, a). \tag{26}$$

Let $\mathbf{g}_{\pi_{\boldsymbol{\theta}}}^c(s,a) = (\widehat{G}_{\pi_{\boldsymbol{\theta}}}^{c_1}(s,a), \widehat{G}_{\pi_{\boldsymbol{\theta}}}^{c_2}(s,a), \cdots, \widehat{G}_{\pi_{\boldsymbol{\theta}}}^{c_m}(s,a))^\top$ collect all the policy gradient estimators of cost value function. Let the matrix $\widehat{\mathbf{G}}(\pi_{\boldsymbol{\theta}}, \boldsymbol{\lambda}) \in \mathbb{R}^{|\mathcal{S}| \times |\mathcal{A}|}$, and each element $\widehat{\mathbf{G}}(\pi_{\boldsymbol{\theta}}, \boldsymbol{\lambda})[s,a]$ is:

$$\widehat{\mathbf{G}}(\pi_{\boldsymbol{\theta}}, \boldsymbol{\lambda})[s,a] = \widehat{G}_{\pi_{\boldsymbol{\theta}}}(s,a) - \boldsymbol{\lambda}^\top \mathbf{g}_{\pi_{\boldsymbol{\theta}}}^c(s,a). \tag{27}$$

Then according to Theorem 3, $\widehat{\mathbf{G}}(\pi_{\boldsymbol{\theta}}, \boldsymbol{\lambda})$ is an unbiased estimator of $\frac{\partial \mathcal{L}(\pi_{\boldsymbol{\theta}}, \boldsymbol{\lambda})}{\partial \boldsymbol{\theta}}$.

**Stochastic Primal-Dual Iteration.** We rewrite the iteration (12) as the following stochastic version,

$$\boldsymbol{\lambda}_t \leftarrow \{\boldsymbol{\lambda}_{t-1} - \eta(\mathbf{b} - \hat{\mathbf{c}}(\pi_{\boldsymbol{\theta}_{t-1}}))\}_+, \boldsymbol{\theta}_t \leftarrow \boldsymbol{\theta}_{t-1} + \eta \widehat{\mathbf{G}}(\pi_{\boldsymbol{\theta}_{t-1}}, \boldsymbol{\lambda}_{t-1}), \tag{28}$$

where we calculate $\hat{\mathbf{c}}(\pi_{\boldsymbol{\theta}_{t-1}})$ and $\widehat{\mathbf{G}}(\pi_{\boldsymbol{\theta}_{t-1}}, \boldsymbol{\lambda}_{t-1})$ according to estimator (25) and estimator (27).

## 4.5 CONVERGENCE RATE

For each time $t$, we notice the estimator $\hat{\mathbf{c}}(\pi_t)$ in the inner loop (see Line 3 of Algorithm 3) involves $m$ trajectories, and estimator $\widehat{\mathbf{G}}(\pi_{\boldsymbol{\theta}_t}, \boldsymbol{\lambda}_t)$ (see Line 5 of Algorithm 3) involves $(2|\mathcal{S}||\mathcal{A}|+m)$ trajectories. We use $\mathcal{D}_t$ to collect all those $(2|\mathcal{S}||\mathcal{A}| + 2m)$ trajectories,

$$\mathcal{D}_t = \{\mathcal{T}_{t,i}\}_{i=1}^{2|\mathcal{S}||\mathcal{A}|+2m}. \tag{29}$$

According to rollout rule in Algorithm 4, Algorithm 5, and Algorithm 2, those $(2|\mathcal{S}||\mathcal{A}| + 2m)$ trajectories among $\mathcal{D}_t$ are independent with each other.

**Theorem 4.** *Under Assumption 1-2, $\pi_{\boldsymbol{\theta}}$ is the softmax policy (9). The time-step $T$ shares a fixed low bound similar to (16). The initial $\boldsymbol{\lambda}_0 = \mathbf{0}$, $\boldsymbol{\theta}_0 = \mathbf{0}$, the parameter sequence $\{\boldsymbol{\lambda}_t, \boldsymbol{\theta}_t\}_{t=0}^{T-1}$ is generated according to Algorithm 3. Let $\eta$, $\beta$ satisfy $\eta = \sqrt{\left\|\frac{d_{\pi_\star}^{\rho_0}}{\rho_0}\right\|_\infty \frac{|\mathcal{S}| \log |\mathcal{A}|}{C'(1-\gamma)T}}$, $\beta = \sqrt{\left\|\frac{d_{\pi_\star}^{\rho_0}}{\rho_0}\right\|_\infty \frac{4|\mathcal{S}| \log |\mathcal{A}|}{(1-\gamma)^3 \iota^2 c_\star}}$, where $C'$ is a positive scalar will be special later. Then for all $i \in \{1, 2, \cdots, m\}$, $\pi_t := \pi_{\boldsymbol{\theta}_t}$ satisfies*

$$\mathbb{E}\left[\min_{t<T}\{J(\pi_\star) - J(\pi_t)\}\right] \leq 4\left\|\frac{d_{\pi_\star}^{\rho_0}}{\rho_0}\right\|_\infty \sqrt{\frac{|\mathcal{S}| \log |\mathcal{A}|}{c_\star(1-\gamma)^4 T}},$$

$$\mathbb{E}\left[\min_{t<T}\{C_i(\pi_t) - b_i\}_+\right] \leq 4\left\|\frac{d_{\pi_\star}^{\rho_0}}{\rho_0}\right\|_\infty \frac{\varrho}{\beta - \|\boldsymbol{\lambda}_\star\|_\infty} \sqrt{\frac{|\mathcal{S}| \log |\mathcal{A}|}{c_\star(1-\gamma)^4 T}},$$

*where the notation $\mathbb{E}[\cdot]$ is short for $\mathbb{E}_{\mathcal{D}_0:\mathcal{D}_{T-1}}[\cdot]$ that denotes the expectation with respect to the randomness over the trajectories $\{\mathcal{D}_t\}_{t=0}^{T-1}$.*

The details of the proof is in Appendix F.2, and the unbiased policy gradient estimator and the independent samples among $\mathcal{D}_t$ play a critical role for us to obtain the Theorem 4. According to (29), we need $(2|\mathcal{S}||\mathcal{A}| + 2m)$ trajectories to rollout the policy gradient estimator, Theorem 4 implies Algorithm 3 needs a total iteration complexity of

$$
\mathcal{O}\left( \left\| \frac{d_{\pi_\star}^{\rho_0}}{\rho_0} \right\|_\infty^2 \frac{|\mathcal{S}|\left(|\mathcal{S}||\mathcal{A}| + m\right)\log|\mathcal{A}|}{c_\star(1-\gamma)^4\epsilon^2} \right)
\tag{30}
$$

to obtain $\mathcal{O}(\epsilon)$-optimality. The iteration complexity shown in (30) matches the best known algorithm NPD-PG Ding et al. (2020). NPD-PG (Ding et al., 2020) and the proposed PD-PG share a complexity of $\mathcal{O}(\epsilon^{-2})$, which is better than CRPO (Xu et al., 2021) that is with $\mathcal{O}(\epsilon^{-4})$ and on-line NPD-PG (Zeng et al., 2021) that is with $\mathcal{O}(\epsilon^{-6})$.

### 4.6 COMMENT ON (DING ET AL., 2020): A TRADE-OFF BETWEEN PD-PG AND NPD-PG

The initial $\boldsymbol{\theta}_0 = \mathbf{0}$ implies the distribution of initial policy is uniform, which implies $\pi_0(a(s)|s) = |\mathcal{A}|^{-1}$. Then $c_\star$ is upper bounded as follows, $c_\star = \inf_{s\in\mathcal{S},t\geq 1}\{\pi_t(a_\star(s)|s)\} \leq |\mathcal{A}|^{-1}$, which implies the iteration complexity of the proposed PD-PG (30) is lower bounded as follows,

$$
\widetilde{\mathcal{O}}\left( \left\| \frac{d_{\pi_\star}^{\rho_0}}{\rho_0} \right\|_\infty^2 \frac{|\mathcal{S}|^2|\mathcal{A}|^2}{(1-\gamma)^4\epsilon^2} \right)
\tag{31}
$$

to obtain $\mathcal{O}(\epsilon)$-optimality, where we omit the constant $m$ (that is the number of constraints), and $\widetilde{\mathcal{O}}(\cdot)$ to hide polylogarithmic factors in the input parameters. According to (Ding et al., 2020, Theorem 4), the complexity of NPD-PG is upper bounded by

$$
\widetilde{\mathcal{O}}\left( \frac{|\mathcal{S}|^2|\mathcal{A}|^2}{(1-\gamma)^4\epsilon^2} \right).
\tag{32}
$$

Although the proposed PD-PG shares the same state-action *independent* iteration complexity of $\mathcal{O}(\epsilon^{-2})$ with NPD-PG, the state-action *dependent* iteration complexity (31) and (32) implies PD-PG is difficult than NPD-PG (Ding et al., 2020) due to the following two aspects. Firstly, the lower bound (31) w.r.t PD-PG and the upper bound (32) w.r.t NPD-PG implies NPD-PG plays never worse than PD-PG. Additionally, the bound w.r.t PD-PG (31) involves the distribution mismatch coefficient $\left\| \frac{d_{\pi_\star}^{\rho_0}}{\rho_0} \right\|_\infty$, which heavily depends on the initial distribution $\rho_0(\cdot)$. Concretely, if $\rho_0(\cdot)$ is near 0 at some state, then distribution mismatch coefficient can be very large, which is indeed detrimental for PD-PG to search a safe policy. In this sense, it also demonstrates the necessity of Assumption 2.

However, although the upper bound (32) w.r.t NPD-PG does not contain the distribution mismatch coefficient, the NPD-PG (Ding et al., 2020) requires additional computation w.r.t the Moore-Penrose pseudo inverse $\mathbf{F}^\dagger(\boldsymbol{\theta})$, where $\mathbf{F}(\boldsymbol{\theta})$ is the Fisher information matrix:

$$
\mathbf{F}(\boldsymbol{\theta}) = \mathbb{E}_{s\sim d_{\pi_{\boldsymbol{\theta}}}^{\rho_0}(\cdot),a\sim\pi_{\boldsymbol{\theta}}(\cdot|s)}\left[ \nabla\log\pi_{\boldsymbol{\theta}}(a|s)(\nabla\log\pi_{\boldsymbol{\theta}}(a|s))^\top \right].
$$

Thus, there exists a hidden trade-off between PD-PG and NPD-PG. Finally, we should emphasize that trade-off is hidden due to the notation $\widetilde{\mathcal{O}}(\cdot)$ covers some information w.r.t MDP or policy space.

## 5 CONCLUSION

This work proposes PD-PG algorithm to solve constrained reinforcement learning problem, which is a Lagrangian-based algorithm with policy gradient. Although the maximization objective is non-concave and the minimization is non-convex over the parameter space, we show that for both exact policy gradient and model-free learning, PD-PG converges to the optimal solution at a sublinear rate for both reward objective and safety constraint. Since we consider discounted infinite-horizon CMDP, we consider unbiased estimators with finite-horizon trajectories, which plays a critical role for us to obtain the iteration complexity of sample-based PD-PG. Additionally, we investigate that PD-PG needs a complexity of $\mathcal{O}\left(\epsilon^{-2}\right)$ to obtain a $\mathcal{O}(\epsilon)$-optimality, which is comparable to state-of the-art algorithms available in the literature in constrained RL.

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

# A   NOTATIONS

For any positive integer $m$, $[m] := \{1, 2, \cdots, m\}$. We use a bold capital letter to denote matrix, e.g., $\mathbf{A} = (a_{i,j}) \in \mathbb{R}^{m \times n}$, and its $(i, j)$-th element denoted as $\mathbf{A}[i, j] := a_{i,j}$, where $i \in [m], j \in [n]$. A bold lowercase letter denotes a vector, e.g., $\mathbf{a} = (a_1, a_2, \cdots, a_n) \in \mathbb{R}^n$, and $\mathbf{a}[i] := a_i$.

## A.1   VECTOR AND MATRIX

$$
\begin{array}{rcl}
\mathbf{1}_m & : & \mathbf{1}_m \in \mathbb{R}^m, \text{ and each element of vector } \mathbf{1}_m \text{ is 1, i.e., } \mathbf{1}_m = (1, 1, \cdots, 1)^\top. \\
\mathbf{0}_m & : & \mathbf{0}_m \in \mathbb{R}^m, \text{ and each element of vector } \mathbf{0}_m \text{ is 1, i.e., } \mathbf{0}_m = (0, 0, \cdots, 0)^\top. \\
\mathbf{a} \preceq \mathbf{b} & : & \text{It denotes component-wise order, i.e., } \mathbf{a}[j] \leq \mathbf{b}[j], j \in [m]. \\
\mathbf{a} \prec \mathbf{b} & : & \text{It denotes component-wise order, i.e., } \mathbf{a}[j] < \mathbf{b}[j], j \in [m].
\end{array}
$$

## A.2   MARKOV DECISION PROCESS

$$
\begin{array}{rcl}
\mathcal{S} & : & \text{The set of states.} \\
\Delta^\circ(\mathcal{S}) & : & \text{The interior of the probability simplex over the state space } \mathcal{S},
\end{array}
$$

$$
\Delta^\circ(\mathcal{S}) := \left\{ p_s \middle| p_s > 0, \sum_{s \in \mathcal{S}} p_s = 1 \right\}.
$$

$$
\begin{array}{rcl}
\mathcal{A} & : & \text{The set of actions.} \\
\mathbb{P}(s'|s, a) & : & \text{The probability of state transition from } s \text{ to } s' \text{ under playing the action } a. \\
r(\cdot) & : & \text{The reward function } r(\cdot) : \mathcal{S} \times \mathcal{S} \times \mathcal{A} \to \mathbb{R}, \text{ and it is bounded by } |r(\cdot)| \leq 1. \\
\rho_0 & : & \rho_0(\cdot) : \mathcal{S} \to [0, 1] \text{ is the initial state distribution.} \\
\gamma & : & \text{The discount factor, and } \gamma \in (0, 1).
\end{array}
$$

## A.3   STATE DISTRIBUTION

$$
\begin{array}{rcl}
\mathbb{P}_\pi(s'|s_0) & : & \text{The probability of single step state transformation probability from } s \text{ to } s' \text{ by} \\
& & \text{executing } \pi. \\
\mathbb{P}_\pi(s_t|s_0) & : & \text{The probability of visiting the state } s_t \text{ after } t \text{ time steps from the initial state } s_0 \\
& & \text{by executing } \pi_\theta. \\
\mathbf{P}_\pi & : & \text{The state transition probability matrix, and its } (s, s')\text{-th component is}
\end{array}
$$

$$
\mathbf{P}_\pi[s, s'] = \sum_{a \in \mathcal{A}} \pi(a|s) \mathbb{P}(s'|s, a) := \mathbb{P}_\pi(s'|s).
$$

$$
\begin{array}{rcl}
d_{s_0}^{\pi_\theta}(s), d_{\rho_0}^{\pi_\theta}(s) & : & \text{The discounted stationary state distribution of the Markov chain (starting at } s_0) \\
& & \text{induced by } \pi,
\end{array}
$$

$$
d_{s_0}^{\pi_\theta}(s) = \sum_{t=0}^\infty \gamma^t \mathbb{P}_\pi(s_t = s|s_0), \quad d_{\rho_0}^{\pi_\theta}(s) = \mathbb{E}_{s_0 \sim \rho_0(\cdot)}[d_{s_0}^\pi(s)].
$$

$$
\left\| \frac{d_{\pi_\star}^{\rho_0}}{\rho_0} \right\|_\infty \quad : \quad \text{The distribution mismatch coefficient, i.e.,}
$$

$$
\left\| \frac{d_{\pi_\star}^{\rho_0}}{\rho_0} \right\|_\infty := \max_{s \in \mathcal{S}} \left\{ \frac{d_{\pi_\star}^{\rho_0}(s)}{\rho_0(s)} \right\}.
$$

Due to the Assumption 2, the term $\dfrac{d_{\pi_\star}^{\rho_0}(s)}{\rho_0(s)}$ is well-defined.

## A.4 STATE, STATE-ACTION AND COST VALUE FUNCTION

| | | |
|---|---|---|
| $V_\pi(s)$ | : | State value function $V_\pi(s) = \mathbb{E}_\pi[\sum_{t=0}^{\infty} \gamma^t r_{t+1}\|s_0 = s]$. |
| $Q_\pi(s,a)$ | : | State-action value function $Q_\pi(s,a) = \mathbb{E}_\pi[\sum_{t=0}^{\infty} \gamma^t r_{t+1}\|s_0 = s, a_0 = a]$. |
| $A_\pi(s,a)$ | : | Advantage function $A_\pi(s,a) = Q_\pi(s,a) - V_\pi(s)$. |
| $J(\pi)$ | : | The objective function. |
| $\mathcal{C}$ | : | The constraint set $\mathcal{C} = \{(c_i, b_i)\}_{i=1}^m$, where $b_i$ are limited values, and cost function $c_i$: $c_i : \mathcal{S} \times \mathcal{A} \to \mathbb{R}$, and it is bounded by $\|c_i(\cdot)\| \leq 1$. |
| $\mathbf{b}$ | : | The vector stores all the limited values: $\mathbf{b} = (b_1, b_2, \cdots, b_m)^\top$. |
| $V_\pi^{c_i}(s)$ | : | State cost value function $V_\pi^{c_i}(s) = \mathbb{E}_\pi[\sum_{t=0}^{\infty} \gamma^t c_i(s_t, a_t)\|s_0 = s]$. |
| $Q_\pi^{c_i}(s,a)$ | : | State-action cost function $Q_\pi^{c_i}(s) = \mathbb{E}_\pi[\sum_{t=0}^{\infty} \gamma^t c_i(s_t, a_t)\|s_0 = s, a_0 = a]$. |
| $A_\pi^{c_i}(s,a)$ | : | Advantage function $A_\pi^{c_i}(s,a) = Q_\pi^{c_i}(s,a) - V_\pi^{c_i}(s)$. |
| $C_i(\pi)$ | : | The expected cost value function $C_i(\pi) = \mathbb{E}_{s \sim \rho_0(\cdot)}[V_\pi^{c_i}(s)]$. |
| $\mathbf{c}(\pi)$ | : | The vector stores all the expected cost values: |

$$\mathbf{c}(\pi) = (C_1(\pi), C_2(\pi), \cdots, C_m(\pi))^\top$$

.

| | | |
|---|---|---|
| $\Pi_{\mathcal{C}}$ | : | The feasible policy set $\Pi_{\mathcal{C}}$ is defined as follows, |

$$\Pi_{\mathcal{C}} = \bigcap_{i=1}^m \{\pi \in \Pi_S \text{ and } C_i(\pi) \leq b_i\}.$$

## A.5 PARAMETER

| | | |
|---|---|---|
| $\boldsymbol{\lambda}$ | : | Lagrange multiplier parameter, $\boldsymbol{\lambda} \in \mathbb{R}^m$. |
| $\boldsymbol{\lambda}_\star$ | : | Optimal dual parameter $\boldsymbol{\lambda}_\star = \arg\min_{\lambda \succeq \mathbf{0}} \max_{\pi \in \Pi_s} \left(J(\pi) + \boldsymbol{\lambda}^\top(\mathbf{b} - \mathbf{c}(\pi))\right)$. |

## A.6 CONSTANT

| | | |
|---|---|---|
| $\rho_{\min}$ | : | Sufficient exploration condition $\rho_{\min} = \min_{s \in \mathcal{S}}\{\rho_0(s)\}$. |
| $m$ | : | The dimension of the vector $\mathbf{b}$, i.e, the number of the constrained conditions (3). |
| $\boldsymbol{\xi}$ | : | A component-wise negitive vector defined in Assumption 1: $\boldsymbol{\xi} := (\xi_1, \xi_2, \cdots, \xi_m)^\top \prec \mathbf{0}$, and each $\xi_i < 0$, $i = 1, 2, \cdots, m$. |
| $\iota$ | : | A positive constant $\iota := \min_{1 \leq i \leq m}\{-\xi_i\}$. |
| $\varrho$ | : | A positive constant $\varrho := 1 + \dfrac{2m}{(1-\gamma)^2 \iota^2}$. |
| $D_1, D_2$ | : | Two positive constants that are defined in Eq.(173). |

# B PRELIMINARIES AND AUXILIARY LEMMA

## B.1 STATE DISTRIBUTION

We use $\mathbf{P}_\pi \in \mathbb{R}^{|\mathcal{S}| \times |\mathcal{S}|}$ to denote the state transition matrix by executing $\pi$, and their components are:

$$\mathbf{P}_\pi[s, s'] = \sum_{a \in \mathcal{A}} \pi(a|s) \mathbb{P}(s'|s, a) := \mathbb{P}_\pi(s'|s), \ \ s, s' \in \mathcal{S},$$

which denotes one-step state transformation probability from $s$ to $s'$.

We use $\mathbb{P}_\pi(s_t = s|s_0)$ to denote the probability of visiting $s$ after $t$ time steps from the initial state $s_0$ by executing $\pi$. Particularly, we notice if $t = 0$, $s_t \neq s_0$, then $\mathbb{P}_\pi(s_t = s|s_0) = 0$, i.e.,

$$\mathbb{P}_\pi(s_t = s|s_0) = 0, \ \ t = 0 \text{ and } s \neq s_0. \tag{33}$$

Then for any initial state $s_0 \sim \rho(\cdot)$, the following holds,

$$\mathbb{P}_\pi(s_t = s|s_0) = \sum_{s' \in \mathcal{S}} \mathbb{P}_\pi(s_t = s|s_{t-1} = s') \mathbb{P}_\pi(s_{t-1} = s'|s_0). \tag{34}$$

In this paper, we also use $\mathbb{P}_\pi^{(t)}$ to denote $\mathbb{P}_\pi(s_t = s'|s)$, i.e.,

$$\mathbb{P}_\pi^{(t)}(s'|s) = \mathbb{P}_\pi(s_t = s'|s_0 = s).$$

Recall $d_\pi^{s_0}(s)$ denotes the normalized discounted distribution of the future state $s$ encountered starting at $s_0$ by executing $\pi$,

$$d_\pi^{s_0}(s) = (1 - \gamma) \sum_{t=0}^{\infty} \gamma^t \mathbb{P}_\pi(s_t = s|s_0). \tag{35}$$

Furthermore, since $s_0 \sim \rho_0(\cdot)$, we define

$$d_\pi^{\rho_0}(s) = \mathbb{E}_{s_0 \sim \rho_0(\cdot)}[d_\pi^{s_0}(s)] = \int_{s_0 \in \mathcal{S}} \rho_0(s_0) d_\pi^{s_0}(s) \mathrm{d}s_0$$

as the discounted state visitation distribution over the initial distribution $\rho_0(\cdot)$

## B.2 PERFORMANCE DIFFERENCE LEMMA

**Lemma 1** (Performance Difference (Kakade & Langford, 2002)). *For any policies $\pi$ and $\pi'$, $s_0 \sim \rho_0(\cdot)$, and for each $i = 1, 2, \cdots, m$, the following performance (or cost) difference holds*

$$J(\pi) - J(\pi') = \frac{1}{1 - \gamma} \mathbb{E}_{s \sim d_\pi^{\rho_0}(\cdot), a \sim \pi(\cdot|s)} \left[ A_{\pi'}(s, a) \right], \tag{36}$$

$$C_i(\pi) - C_i(\pi') = \frac{1}{1 - \gamma} \mathbb{E}_{s \sim d_\pi^{\rho_0}(\cdot), a \sim \pi(\cdot|s)} \left[ A_{\pi'}^{c_i}(s, a) \right]. \tag{37}$$

## B.3 BASIC FACTS

In this section, we present some basic facts will be use later, those results are adaptive to Ding et al. (2020), and we extend them to the versions of vectors.

Recall $\mathcal{L}_D(\boldsymbol{\lambda}) = \max_{\pi \in \Pi_s} \mathcal{L}(\pi, \boldsymbol{\lambda})$, for a give scalar $c \in \mathbb{R}$, we define a notation

$$\boldsymbol{\Gamma}(c) := \{\boldsymbol{\lambda} \succeq \mathbf{0} : \mathcal{L}_D(\boldsymbol{\lambda}) \leq c\}.$$

We notice the following inequality alway holds

$$\max_{\pi \in \Pi_s} \min_{\boldsymbol{\lambda} \succeq \mathbf{0}} \mathcal{L}(\pi, \boldsymbol{\lambda}) \leq \min_{\boldsymbol{\lambda} \succeq \mathbf{0}} \max_{\pi \in \Pi_s} \mathcal{L}(\pi, \boldsymbol{\lambda}) = \min_{\boldsymbol{\lambda} \succeq \mathbf{0}} \mathcal{L}_D(\boldsymbol{\lambda}), \tag{38}$$

if $c < \max_{\pi \in \Pi_s} \min_{\boldsymbol{\lambda} \succeq \mathbf{0}} \mathcal{L}(\pi, \boldsymbol{\lambda}) \overset{(8)}{=} J(\pi_\star)$, then $\boldsymbol{\Gamma}(c) = \emptyset$. We assume $c \geq J(\pi_\star)$ throughout this paper.

**Lemma 2.** *Recall the optimal dual variable* $\boldsymbol{\lambda}_\star = \arg\min_{\boldsymbol{\lambda} \succeq \mathbf{0}} \mathcal{L}_D(\boldsymbol{\lambda})$, *let* $c = J(\pi_\star)$, *the following holds*

$$\boldsymbol{\lambda}_\star \in \boldsymbol{\Gamma}(c). \tag{39}$$

*Proof.* If we choose $c = J(\pi_\star)$, and let $\tilde{\boldsymbol{\lambda}} \in \boldsymbol{\Gamma}(c)$, then by the definition of $\boldsymbol{\Gamma}(c)$, we achieve

$$\mathcal{L}_D(\tilde{\boldsymbol{\lambda}}) = \max_{\pi \in \Pi_s} \mathcal{L}(\pi, \tilde{\boldsymbol{\lambda}}) \le J(\pi_\star) \overset{(8)}{=} \max_{\pi \in \Pi_s} \min_{\boldsymbol{\lambda} \succeq \mathbf{0}} \mathcal{L}(\pi, \boldsymbol{\lambda}). \tag{40}$$

Let's maximize the left-hand of Eq.(40) overset the space $\left\{\tilde{\boldsymbol{\lambda}} \succeq \mathbf{0}\right\}$, combining the result (38), then the vector $\tilde{\boldsymbol{\lambda}}$ satisfies

$$\max_{\pi \in \Pi_s} \min_{\tilde{\boldsymbol{\lambda}} \succeq \mathbf{0}} \mathcal{L}(\pi, \boldsymbol{\lambda}) = \min_{\tilde{\boldsymbol{\lambda}} \succeq \mathbf{0}} \max_{\pi \in \Pi_s} \mathcal{L}(\pi, \boldsymbol{\lambda}), \tag{41}$$

which implies $\tilde{\boldsymbol{\lambda}} = \boldsymbol{\lambda}_\star$, further it implies

$$\boldsymbol{\lambda}_\star \in \boldsymbol{\Gamma}(J(\pi_\star)). \tag{42}$$

$\square$

With the result of Lemma 2, we denote the set of all optimal dual variables as $\boldsymbol{\Gamma}_\star$, i.e.,

$$\boldsymbol{\Gamma}_\star := \left\{\boldsymbol{\lambda} : \arg\min_{\boldsymbol{\lambda} \succeq \mathbf{0}} \mathcal{L}_D(\boldsymbol{\lambda})\right\} = \boldsymbol{\Gamma}(J(\pi_\star)).$$

**Lemma 3.** *Consider the policy* $\bar{\pi}$ *satisfies Assumption 1, let* $\boldsymbol{\lambda} \in \boldsymbol{\Gamma}(c)$, *then the following holds*

$$c - J(\bar{\pi}) \ge -\boldsymbol{\lambda}^\top \boldsymbol{\xi}.$$

*Furthermore, the optimal dual variable* $\boldsymbol{\lambda}_\star$ *is bounded as follows,*

$$\|\boldsymbol{\lambda}_\star\|_\infty \le \frac{1}{\iota}(J(\pi_\star) - J(\bar{\pi})) \le \frac{2}{(1-\gamma)\iota}.$$

*Proof.* Let $\boldsymbol{\lambda} \in \boldsymbol{\Gamma}(c)$, and recall Assumption 1 and $\mathcal{L}_D(\boldsymbol{\lambda}) = \max_{\pi \in \Pi_s} \mathcal{L}(\pi, \boldsymbol{\lambda})$, then

$$c \ge \mathcal{L}_D(\boldsymbol{\lambda}) \ge J(\bar{\pi}) + \boldsymbol{\lambda}^\top(\mathbf{b} - \mathbf{c}(\bar{\pi})) \overset{(7)}{\ge} J(\bar{\pi}) - \boldsymbol{\lambda}^\top \boldsymbol{\xi},$$

which implies

$$c - J(\bar{\pi}) \ge -\boldsymbol{\lambda}^\top \boldsymbol{\xi}. \tag{43}$$

Furthermore, according to Lemma 2, if $c = J(\pi_\star)$, then for each $\boldsymbol{\lambda}_\star \in \boldsymbol{\Gamma}(J(\pi_\star))$, Eq.(43) implies

$$J(\pi_\star) - J(\bar{\pi}) \ge -\boldsymbol{\lambda}_\star^\top \boldsymbol{\xi} = \boldsymbol{\lambda}_\star^\top(-\boldsymbol{\xi}). \tag{44}$$

Recall Assumption 1, we know $\boldsymbol{\xi} \prec \mathbf{0}$, i.e., $\boldsymbol{\xi} := (\xi_1, \xi_2, \cdots, \xi_m)^\top \prec \mathbf{0}$, which implies each $\xi_i < 0$, $i = 1, 2, \cdots, m$. Let

$$\iota := \min_{1 \le i \le m} \{-\xi_i\},$$

then $\iota$ is a positive scalar, and $-\boldsymbol{\xi} \succ \iota \mathbf{1}_m$. Let $\boldsymbol{\lambda}_\star = (\lambda_1^*, \lambda_2^*, \cdots, \lambda_m^*)^\top$, since $\boldsymbol{\lambda}_\star \succ \mathbf{0}$, then each $\lambda_i^* \ge 0$. Furthermore, $\|\boldsymbol{\lambda}_\star\|_\infty = \max\{\lambda_1^*, \lambda_2^*, \cdots, \lambda_m^*\}$, according to Eq.(44), we achieve

$$J(\pi_\star) - J(\bar{\pi}) \ge \iota \boldsymbol{\lambda}_\star^\top \mathbf{1}_m = \iota \sum_{i=1}^{m} \lambda_i^* \ge \iota \|\boldsymbol{\lambda}_\star\|_\infty, \tag{45}$$

which implies

$$\|\boldsymbol{\lambda}_\star\|_\infty \le \frac{1}{\iota}(J(\pi_\star) - J(\bar{\pi})) \le \frac{2}{(1-\gamma)\iota}. \tag{46}$$

$\square$

**Lemma 4.** *Let $\varphi > \|\boldsymbol{\lambda}_\star\|_\infty$, and for any policy $\tilde{\pi}$ such that*

$$J(\pi_\star) - J(\tilde{\pi}) + \varphi \mathbf{1}_m^\top \{\mathbf{c}(\tilde{\pi}) - \mathbf{b}\}_+ \leq \delta, \tag{47}$$

*then*

$$\mathbf{1}_m^\top \{\mathbf{c}(\tilde{\pi}) - \mathbf{b}\}_+ < \frac{\delta}{\varphi - \|\boldsymbol{\lambda}_\star\|_\infty}. \tag{48}$$

*Proof.* Let

$$v(\boldsymbol{\omega}) = \max_{\pi \in \Pi_S} \{J(\pi), \text{ and } \mathbf{b} - \mathbf{c}(\pi) \succeq \boldsymbol{\omega}\},$$

according to Paternain et al. (2019b); Ding et al. (2020), $v(\boldsymbol{\omega})$ is concave. We notice $J(\pi_\star) = v(\mathbf{0})$. Recall $\boldsymbol{\lambda}_\star = \arg \min_{\boldsymbol{\lambda} \succeq \mathbf{0}} \mathcal{L}_D(\boldsymbol{\lambda})$, then according to Theorem 1, we achieve

$$\mathcal{L}(\pi, \boldsymbol{\lambda}_\star) \leq \max_{\pi \in \Pi_S} \mathcal{L}(\pi, \boldsymbol{\lambda}_\star) = \mathcal{L}_D(\boldsymbol{\lambda}_\star) = J(\pi_\star) = v(\mathbf{0}), \quad \forall \pi \in \Pi_S. \tag{49}$$

Then, for each $\pi$ such that

$$\pi \in \{\pi \in \Pi_S : \mathbf{b} - \mathbf{c}(\pi) \succeq \boldsymbol{\omega}\},$$

the following holds

$$v(\mathbf{0}) - \boldsymbol{\lambda}_\star^\top \boldsymbol{\omega} \overset{(49)}{\geq} \mathcal{L}(\pi, \boldsymbol{\lambda}_\star) - \boldsymbol{\lambda}_\star^\top \boldsymbol{\omega} = J(\pi) + \boldsymbol{\lambda}_\star^\top (\mathbf{b} - \mathbf{c}(\pi)) - \boldsymbol{\lambda}_\star^\top \boldsymbol{\omega} \geq J(\pi), \tag{50}$$

where the last Eq.(50) holds: since $\mathbf{b} - \mathbf{c}(\pi) \succeq \boldsymbol{\omega}$, then $\boldsymbol{\lambda}_\star^\top (\mathbf{b} - \mathbf{c}(\pi)) - \boldsymbol{\lambda}_\star^\top \boldsymbol{\omega} \geq 0$.

Let's maximize the right-hand of Eq.(50) with respect to $\pi$ over the space $\{\pi \in \Pi_S : \mathbf{b} - \mathbf{c}(\pi) \succeq \boldsymbol{\omega}\}$, then we achieve

$$v(\mathbf{0}) - \boldsymbol{\lambda}_\star^\top \boldsymbol{\omega} \geq v(\boldsymbol{\omega}). \tag{51}$$

Furthermore, if we choose

$$\tilde{\boldsymbol{\omega}} := -\{\mathbf{c}(\tilde{\pi}) - \mathbf{b}\}_+, \tag{52}$$

then

$$J(\tilde{\pi}) \leq J(\pi_\star) = v(\mathbf{0}) \leq v(\tilde{\boldsymbol{\omega}}), \tag{53}$$

where the last inequality holds since

$$\{\pi : \mathbf{b} - \mathbf{c}(\pi) \succeq \mathbf{0}\} \subset \{\pi : \mathbf{b} - \mathbf{c}(\pi) \succeq \tilde{\boldsymbol{\omega}}\}.$$

Finally, considering the results from (51) to (53), we have

$$J(\tilde{\pi}) - J(\pi_\star) \overset{(53)}{\leq} v(\tilde{\boldsymbol{\omega}}) - J(\pi_\star) = v(\tilde{\boldsymbol{\omega}}) - v(\mathbf{0}) \overset{(51)}{\leq} -\boldsymbol{\lambda}_\star^\top \tilde{\boldsymbol{\omega}}. \tag{54}$$

Consider the condition (47),

$$\begin{aligned}
\delta &\overset{(47)}{\geq} J(\pi_\star) - J(\tilde{\pi}) + \varphi \mathbf{1}_m^\top \{\mathbf{c}(\tilde{\pi}) - \mathbf{b}\}_+ \\
&\overset{(54)}{\geq} \boldsymbol{\lambda}_\star^\top \tilde{\boldsymbol{\omega}} + \varphi \mathbf{1}_m^\top \{\mathbf{c}(\tilde{\pi}) - \mathbf{b}\}_+ \\
&\overset{(52)}{\geq} (\varphi \mathbf{1}_m - \boldsymbol{\lambda}_\star)^\top \{\mathbf{c}(\tilde{\pi}) - \mathbf{b}\}_+ > (\varphi - \|\boldsymbol{\lambda}_\star\|_\infty) \mathbf{1}_m^\top \{\mathbf{c}(\tilde{\pi}) - \mathbf{b}\}_+,
\end{aligned} \tag{55}$$

where the last inequality holds: since $\varphi > \|\boldsymbol{\lambda}_\star\|_\infty$, then the following equation always holds

$$\varphi \mathbf{1}_m - \boldsymbol{\lambda}_\star \succeq (\varphi - \|\boldsymbol{\lambda}_\star\|_\infty) \mathbf{1}_m.$$

Eq.(55) implies

$$\mathbf{1}_m^\top \{\mathbf{c}(\tilde{\pi}) - \mathbf{b}\}_+ < \frac{\delta}{\varphi - \|\boldsymbol{\lambda}_\star\|_\infty}. \tag{56}$$

$\square$

## C  POLICY GRADIENT W.R.T. OBJECTIVE AND COST VALUE FUNCTION

Although some similar results with respect to Proposition 1 have appeared in Agarwal et al. (2020b); Mei et al. (2020); Lan (2021), we also need to provide the details since we will use some key details later. Before we show Proposition 1, we need to calculate $\dfrac{\partial \pi_{\boldsymbol{\theta}}(a^{'}|s^{'})}{\partial \theta_{s,a}}$, which plays a critical role to proof Proposition 1.

**Lemma 5.** *Let $\pi_{\boldsymbol{\theta}}$ be the softmax policy defined in (9), then*

$$
\begin{aligned}
\frac{\partial \pi_{\boldsymbol{\theta}}(a^{'}|s^{'})}{\partial \theta_{s,a}} &= \frac{\partial}{\partial \theta_{s,a}} \left( \frac{\exp\{\theta_{s^{'},a^{'}}\}}{\sum_{\tilde{a}\in\mathcal{A}} \exp\{\theta_{s^{'},\tilde{a}}\}} \right) \\
&= \frac{\frac{\partial}{\partial \theta_{s,a}}\exp\{\theta_{s^{'},a^{'}}\}(\sum_{\tilde{a}\in\mathcal{A}}\exp\{\theta_{s^{'},\tilde{a}}\}) - \exp\{\theta_{s^{'},a^{'}}\}\frac{\partial}{\partial \theta_{s,a}}(\sum_{\tilde{a}\in\mathcal{A}}\exp\{\theta_{s^{'},\tilde{a}}\})}{(\sum_{\tilde{a}\in\mathcal{A}}\exp\{\theta_{s^{'},\tilde{a}}\})^2} \\
&= \begin{cases} \dfrac{\exp\{\theta_{s,a}\}(\sum_{\tilde{a}\in\mathcal{A}}\exp\{\theta_{s,\tilde{a}}\}) - (\exp\{\theta_{s,a}\})^2}{(\sum_{\tilde{a}\in\mathcal{A}}\exp\{\theta_{s,\tilde{a}}\})^2}, & \text{if } s^{'}=s \text{ and } a^{'}=a; \\[4mm] -\dfrac{\exp\{\theta_{s,a^{'}}\}\exp\{\theta_{s,a}\}}{(\sum_{\tilde{a}\in\mathcal{A}}\exp\{\theta_{s,\tilde{a}}\})^2}, & \text{if } s^{'}=s \text{ and } a^{'}\neq a; \\[4mm] 0, & \text{if } s^{'}\neq s \text{ or } a^{'}\neq a; \end{cases} \\
&= \begin{cases} \pi_{\boldsymbol{\theta}}(a|s) - (\pi_{\boldsymbol{\theta}}(a|s))^2, & \text{if } s^{'}=s \text{ and } a^{'}=a; \\[4mm] -\pi_{\boldsymbol{\theta}}(a^{'}|s)\pi_{\boldsymbol{\theta}}(a|s) & \text{if } s^{'}=s \text{ and } a^{'}\neq a; \qquad (57) \\[4mm] 0. & \text{if } s^{'}\neq s \text{ or } a^{'}\neq a. \end{cases}
\end{aligned}
$$

**Proposition 1**. *Under the softmax policy parameterization (9), the gradient of the objective $J(\pi_{\boldsymbol{\theta}})$ and cost $C_i(\pi_{\boldsymbol{\theta}})$ with respect to $\boldsymbol{\theta}$ is*

$$
\begin{aligned}
\frac{\partial J(\pi_{\boldsymbol{\theta}})}{\partial \theta_{s,a}} &= \frac{1}{1-\gamma} d_{\pi_{\boldsymbol{\theta}}}^{\rho_0}(s)\pi_{\boldsymbol{\theta}}(a|s)A_{\pi_{\boldsymbol{\theta}}}(s,a), \\
\frac{\partial C_i(\pi_{\boldsymbol{\theta}})}{\partial \theta_{s,a}} &= \frac{1}{1-\gamma} d_{\pi_{\boldsymbol{\theta}}}^{\rho_0}(s)\pi_{\boldsymbol{\theta}}(a|s)A_{\pi_{\boldsymbol{\theta}}}^{c_i}(s,a).
\end{aligned}
$$

*Furthermore, let*

$$
\begin{aligned}
\mathbf{a}_{\pi_{\boldsymbol{\theta}}}^c(s,a) &= \left( A_{\pi_{\boldsymbol{\theta}}}^{c_1}(s,a), A_{\pi_{\boldsymbol{\theta}}}^{c_2}(s,a), \cdots, A_{\pi_{\boldsymbol{\theta}}}^{c_m}(s,a) \right)^{\top}, \\
A_{\pi_{\boldsymbol{\theta}}}(s,a,\boldsymbol{\lambda}) &= A_{\pi_{\boldsymbol{\theta}}}(s,a) - \boldsymbol{\lambda}^{\top}\mathbf{a}_{\pi_{\boldsymbol{\theta}}}^c(s,a),
\end{aligned}
$$

*then the gradients of $\mathcal{L}(\pi_{\boldsymbol{\theta}},\boldsymbol{\lambda})$ with respect to $\boldsymbol{\theta},\boldsymbol{\lambda}$ are:*

$$
\frac{\partial \mathcal{L}(\pi_{\boldsymbol{\theta}},\boldsymbol{\lambda})}{\partial \theta_{s,a}} = \frac{1}{1-\gamma} d_{\pi_{\boldsymbol{\theta}}}^{\rho_0}(s)\pi_{\boldsymbol{\theta}}(a|s)A_{\pi_{\boldsymbol{\theta}}}(s,a,\boldsymbol{\lambda}), \quad \frac{\partial \mathcal{L}(\pi_{\boldsymbol{\theta}},\boldsymbol{\lambda})}{\partial \boldsymbol{\lambda}} = \mathbf{b} - \mathbf{c}(\pi_{\boldsymbol{\theta}}). \qquad (58)
$$

*Proof.* Since $J(\pi_{\boldsymbol{\theta}}) = \mathbb{E}_{s\sim d_{\pi_{\boldsymbol{\theta}}}^{\rho_0}(\cdot)}[V_{\pi_{\boldsymbol{\theta}}}(s)]$, to derive the gradient $\dfrac{\partial J(\pi_{\boldsymbol{\theta}})}{\partial \theta_{s,a}}$, we only need to show $\dfrac{\partial V_{\pi_{\boldsymbol{\theta}}}(s_0)}{\partial \theta_{s,a}}$. According to the relationship between $V_{\pi_{\boldsymbol{\theta}}}(s)$ and $Q_{\pi_{\boldsymbol{\theta}}}(s,a)$:

$$
V_{\pi_{\boldsymbol{\theta}}}(s_0) = \sum_{a^{'}\in\mathcal{A}} \pi_{\boldsymbol{\theta}}(a^{'}|s_0)Q_{\pi_{\boldsymbol{\theta}}}(s_0,a^{'}),
$$

then we have

$$\frac{\partial V_{\pi_\theta}(s_0)}{\partial \theta_{s,a}} = \sum_{a' \in \mathcal{A}} \left( \frac{\partial \pi_\theta(a'|s_0)}{\partial \theta_{s,a}} Q_{\pi_\theta}(s_0, a') + \pi_\theta(a'|s_0) \frac{\partial Q_{\pi_\theta}(s_0, a')}{\partial \theta_{s,a}} \right). \tag{59}$$

Due to the equation

$$Q_{\pi_\theta}(s, a) = \sum_{s' \in \mathcal{S}} \mathbb{P}(s'|s, a) r(s'|s, a) + \gamma \sum_{s' \in \mathcal{S}} \mathbb{P}(s'|s, a) V_{\pi_\theta}(s'),$$

we achieve the gradient of $Q_{\pi_\theta}$ with respect to $\boldsymbol{\theta}$ as follows,

$$\frac{\partial Q_{\pi_\theta}(s_0, a')}{\partial \theta_{s,a}} = \gamma \sum_{s' \in \mathcal{S}} \mathbb{P}(s'|s_0, a') \frac{\partial V_{\pi_\theta}(s')}{\partial \theta_{s,a}}. \tag{60}$$

Taking Eq.(60) to Eq.(59), we have

$$\frac{\partial V_{\pi_\theta}(s_0)}{\partial \theta_{s,a}} = \sum_{a' \in \mathcal{A}} \left( \frac{\partial \pi_\theta(a'|s_0)}{\partial \theta_{s,a}} Q_{\pi_\theta}(s_0, a') + \gamma \pi_\theta(a'|s_0) \sum_{s' \in \mathcal{S}} \mathbb{P}(s'|s_0, a') \frac{\partial V_{\pi_\theta}(s')}{\partial \theta_{s,a}} \right)$$

$$= \sum_{a' \in \mathcal{A}} \frac{\partial \pi_\theta(a'|s_0)}{\partial \theta_{s,a}} Q_{\pi_\theta}(s_0, a') + \gamma \sum_{s' \in \mathcal{S}} \underbrace{\left( \sum_{a' \in \mathcal{A}} \pi_\theta(a'|s_0) \mathbb{P}(s'|s_0, a') \right)}_{=\mathbb{P}_{\pi_\theta}(s_1 = s'|s_0)} \frac{\partial V_{\pi_\theta}(s')}{\partial \theta_{s,a}}$$

$$= \sum_{a' \in \mathcal{A}} \frac{\partial \pi_\theta(a'|s_0)}{\partial \theta_{s,a}} Q_{\pi_\theta}(s_0, a') + \gamma \sum_{s' \in \mathcal{S}} \mathbb{P}_{\pi_\theta}(s_1 = s'|s_0) \frac{\partial V_{\pi_\theta}(s')}{\partial \theta_{s,a}}, \tag{61}$$

which implies for each $t \in \mathbb{N}^+$, the following equation holds:

$$\frac{\partial V_{\pi_\theta}(s')}{\partial \theta_{s,a}} = \sum_{a' \in \mathcal{A}} \frac{\partial \pi_\theta(a'|s')}{\partial \theta_{s,a}} Q_{\pi_\theta}(s', a') + \gamma \sum_{s'' \in \mathcal{S}} \mathbb{P}_{\pi_\theta}(s_{t+1} = s''|s_t = s') \frac{\partial V_{\pi_\theta}(s'')}{\partial \theta_{s,a}}. \tag{62}$$

Considering Eq.(62) with the case $t = 1$, we write Eq.(61) as follows,

$$\frac{\partial V_{\pi_\theta}(s_0)}{\partial \theta_{s,a}} = \sum_{a' \in \mathcal{A}} \frac{\partial \pi_\theta(a'|s_0)}{\partial \theta_{s,a}} Q_{\pi_\theta}(s_0, a')$$

$$+ \gamma \sum_{s' \in \mathcal{S}} \mathbb{P}_{\pi_\theta}(s_1 = s'|s_0) \left( \sum_{a' \in \mathcal{A}} \frac{\partial \pi_\theta(a'|s')}{\partial \theta_{s,a}} Q_{\pi_\theta}(s', a') + \gamma \sum_{s'' \in \mathcal{S}} \mathbb{P}_{\pi_\theta}(s_2 = s''|s_1 = s') \frac{\partial V_{\pi_\theta}(s'')}{\partial \theta_{s,a}} \right). \tag{63}$$

According to Eq.(34), the following equation holds

$$\sum_{s' \in \mathcal{S}} \mathbb{P}_{\pi_\theta}(s_1 = s'|s_0) \mathbb{P}_{\pi_\theta}(s_2 = s''|s_1 = s') = \mathbb{P}_{\pi_\theta}(s_2 = s''|s_0),$$

and taking it to Eq.(63), we achieve the gradient $\dfrac{\partial V_{\pi_\theta}(s_0)}{\partial \theta_{s,a}}$ as follows,

$$\frac{\partial V_{\pi_\theta}(s_0)}{\partial \theta_{s,a}} = \sum_{a' \in \mathcal{A}} \frac{\partial \pi_\theta(a'|s_0)}{\partial \theta_{s,a}} Q_{\pi_\theta}(s_0, a') + \gamma \sum_{s' \in \mathcal{S}} \mathbb{P}_{\pi_\theta}(s_1 = s'|s_0) \sum_{a' \in \mathcal{A}} \frac{\partial \pi_\theta(a'|s')}{\partial \theta_{s,a}} Q_{\pi_\theta}(s', a')$$

$$+ \gamma^2 \sum_{s'' \in \mathcal{S}} \sum_{s' \in \mathcal{S}} \mathbb{P}_{\pi_\theta}(s_1 = s'|s_0) \mathbb{P}_{\pi_\theta}(s_2 = s''|s_1 = s') \frac{\partial V_{\pi_\theta}(s'')}{\partial \theta_{s,a}}$$

$$= \sum_{a' \in \mathcal{A}} \frac{\partial \pi_\theta(a'|s_0)}{\partial \theta_{s,a}} Q_{\pi_\theta}(s_0, a') + \gamma \sum_{a' \in \mathcal{A}} \sum_{s' \in \mathcal{S}} \mathbb{P}_{\pi_\theta}(s_1 = s'|s_0) \frac{\partial \pi_\theta(a'|s')}{\partial \theta_{s,a}} Q_{\pi_\theta}(s', a')$$

$$+ \gamma^2 \sum_{s'' \in \mathcal{S}} \mathbb{P}_{\pi_\theta}(s_2 = s''|s_0) \frac{\partial V_{\pi_\theta}(s'')}{\partial \theta_{s,a}}. \tag{64}$$

Furthermore, according to (64), we analyze $\dfrac{\partial V_{\pi_\theta}(s_0)}{\partial \theta_{s,a}}$ as follows,

$$
\frac{\partial V_{\pi_\theta}(s_0)}{\partial \theta_{s,a}} = \sum_{a' \in \mathcal{A}} \frac{\partial \pi_\theta(a'|s_0)}{\partial \theta_{s,a}} Q_{\pi_\theta}(s_0,a') + \gamma \sum_{a' \in \mathcal{A}} \sum_{s' \in \mathcal{S}} \mathbb{P}_{\pi_\theta}(s_1 = s'|s_0) \frac{\partial \pi_\theta(a'|s')}{\partial \theta_{s,a}} Q_{\pi_\theta}(s',a')
$$

$$
+ \gamma^2 \sum_{s'' \in \mathcal{S}} \mathbb{P}_{\pi_\theta}(s_2 = s''|s_0) \left( \sum_{a' \in \mathcal{A}} \frac{\partial \pi_\theta(a'|s'')}{\partial \theta_{s,a}} Q_{\pi_\theta}(s'',a') + \gamma \sum_{s''' \in \mathcal{S}} \mathbb{P}_{\pi_\theta}(s_3 = s'''|s_2 = s'') \frac{\partial V_{\pi_\theta}(s''')}{\partial \theta_{s,a}} \right)
$$

$$
= \sum_{a' \in \mathcal{A}} \frac{\partial \pi_\theta(a'|s_0)}{\partial \theta_{s,a}} Q_{\pi_\theta}(s_0,a') + \sum_{a' \in \mathcal{A}} \sum_{s' \in \mathcal{S}} \left( \gamma \mathbb{P}_{\pi_\theta}(s_1 = s'|s_0) + \gamma^2 \mathbb{P}_{\pi_\theta}(s_2 = s'|s_0) \right) \frac{\partial \pi_\theta(a'|s')}{\partial \theta_{s,a}} Q_{\pi_\theta}(s',a')
$$

$$
+ \gamma^3 \sum_{s' \in \mathcal{S}} \mathbb{P}_{\pi_\theta}(s_3 = s'|s_0) \frac{\partial}{\partial \theta_{s,a}} V_{\pi_\theta}(s')
$$

$$
= \cdots\cdots
$$

$$
= \sum_{a' \in \mathcal{A}} \sum_{s' \in \mathcal{S}} \sum_{t=0}^{\infty} \gamma^t \mathbb{P}_{\pi_\theta}(s_t = s'|s_0) \frac{\partial \pi_\theta(a'|s')}{\partial \theta_{s,a}} Q_{\pi_\theta}(s',a') \tag{65}
$$

$$
= \frac{1}{1-\gamma} \sum_{a' \in \mathcal{A}} \sum_{s' \in \mathcal{S}} d_{\pi_\theta}^{s_0}(s') \frac{\partial \pi_\theta(a'|s')}{\partial \theta_{s,a}} Q_{\pi_\theta}(s',a') = \frac{1}{1-\gamma} \sum_{a' \in \mathcal{A}} d_{\pi_\theta}^{s_0}(s) \frac{\partial \pi_\theta(a'|s)}{\partial \theta_{s,a}} Q_{\pi_\theta}(s,a') \tag{66}
$$

$$
= \frac{1}{1-\gamma} d_{\pi_\theta}^{s_0}(s) \pi_\theta(a|s) \left( Q_{\pi_\theta}(s,a) - \sum_{a' \in \mathcal{A}} \pi_\theta(a'|s) Q_{\pi_\theta}(s,a') \right) \tag{67}
$$

$$
= \frac{1}{1-\gamma} d_{\pi_\theta}^{s_0}(s) \pi_\theta(a|s) \left( Q_{\pi_\theta}(s,a) - V_{\pi_\theta}(s) \right) = \frac{1}{1-\gamma} d_{\pi_\theta}^{s_0}(s) \pi_\theta(a|s) A_{\pi_\theta}(s,a), \tag{68}
$$

where Eq.(66) holds since if $s' \neq s$, then

$$
\frac{\partial \pi_\theta(a'|s')}{\partial \theta_{s,a}} = 0;
$$

Eq.(67) holds due to Eq.(57).

Finally, since $J(\pi_\theta) = \mathbb{E}_{s_0 \sim d_{\pi_\theta}^{\rho_0}(\cdot)}[V_{\pi_\theta}(s_0)]$, then

$$
\frac{\partial J(\pi_\theta)}{\partial \theta_{s,a}} = \mathbb{E}_{s_0 \sim \rho_0(\cdot)} \left[ \frac{1}{1-\gamma} d_{\pi_\theta}^{s_0}(s) \pi_\theta(a|s) A_{\pi_\theta}(s,a) \right] = \frac{1}{1-\gamma} d_{\pi_\theta}^{\rho_0}(s) \pi_\theta(a|s) A_{\pi_\theta}(s,a).
$$

Similarly,

$$
\frac{\partial C_i(\pi_\theta)}{\partial \theta_{s,a}} = \frac{1}{1-\gamma} d_{\pi_\theta}^{\rho_0}(s) \pi_\theta(a|s) A_{\pi_\theta}^{c_i}(s,a).
$$

This concludes the proof of Proposition 1. $\qquad\square$

Let

$$
\mathbf{a}_{\pi_\theta}^c(s,a) = \left( A_{\pi_\theta}^{c_1}(s,a), A_{\pi_\theta}^{c_2}(s,a), \cdots, A_{\pi_\theta}^{c_m}(s,a) \right)^\top,
$$

then

$$
\frac{\partial \mathbf{c}(\pi_\theta)}{\partial \theta_{s,a}} = \left( \frac{\partial C_1(\pi_\theta)}{\partial \theta_{s,a}}, \frac{\partial C_2(\pi_\theta)}{\partial \theta_{s,a}}, \cdots, \frac{\partial C_m(\pi_\theta)}{\partial \theta_{s,a}} \right)^\top
$$

$$
= \frac{1}{1-\gamma} d_{\pi_\theta}^{\rho_0}(s) \pi_\theta(a|s) \left( A_{\pi_\theta}^{c_1}(s,a), A_{\pi_\theta}^{c_2}(s,a), \cdots, A_{\pi_\theta}^{c_m}(s,a) \right)^\top
$$

$$
= \frac{1}{1-\gamma} d_{\pi_\theta}^{\rho_0}(s) \pi_\theta(a|s) \mathbf{a}_{\pi_\theta}^c(s,a).
$$

With the result in Proposition 1, it is easy to calculate the gradient of $\mathcal{L}(\pi_\theta, \boldsymbol{\lambda})$ (5) with respect to $\boldsymbol{\theta}, \boldsymbol{\lambda}$, formally, we present it in the following Proposition 5.

**Proposition 5.** *Consider the Lagrange multiplier function $\mathcal{L}(\pi_{\boldsymbol{\theta}}, \boldsymbol{\lambda})$ (5), its gradient with respect to $\boldsymbol{\theta}, \boldsymbol{\lambda}$ is:*

$$\frac{\partial}{\partial \theta_{s,a}} \mathcal{L}(\pi_{\boldsymbol{\theta}}, \boldsymbol{\lambda}) = \frac{1}{1-\gamma} d_{\pi_{\boldsymbol{\theta}}}^{\rho_0}(s) \pi_{\boldsymbol{\theta}}(a|s) \left( A_{\pi_{\boldsymbol{\theta}}}(s,a) - \boldsymbol{\lambda}^{\top} \mathbf{a}_{\pi_{\boldsymbol{\theta}}}^c(s,a) \right), \tag{69}$$

$$\frac{\partial}{\partial \boldsymbol{\lambda}} \mathcal{L}(\pi_{\boldsymbol{\theta}}, \boldsymbol{\lambda}) = \mathbf{b} - \mathbf{c}(\pi_{\boldsymbol{\theta}}). \tag{70}$$

We consider the following update rule with respect to $\boldsymbol{\lambda}$ and $\boldsymbol{\theta}$:

$$\boldsymbol{\lambda}_t \leftarrow \left\{ \boldsymbol{\lambda}_{t-1} - \eta \frac{\partial}{\partial \boldsymbol{\lambda}} \mathcal{L}(\pi_{\boldsymbol{\theta}_{t-1}}, \boldsymbol{\lambda}_{t-1}) \right\}_+ ; \tag{71}$$

$$\boldsymbol{\theta}_t \leftarrow \boldsymbol{\theta}_{t-1} + \eta \frac{\partial}{\partial \boldsymbol{\theta}} \mathcal{L}(\pi_{\boldsymbol{\theta}_{t-1}}, \boldsymbol{\lambda}_{t-1}), \tag{72}$$

where each $(s, a)$-component update rule of $\boldsymbol{\theta}_t$ is defined as follows,

$$\boldsymbol{\theta}_t[s,a] := \theta_{s,a}^{(t)} \leftarrow \theta_{s,a}^{(t-1)} + \eta \frac{\partial}{\partial \theta_{s,a}} \mathcal{L}(\pi_{\boldsymbol{\theta}_{t-1}}, \boldsymbol{\lambda}_{t-1}) \tag{73}$$

$$= \theta_{s,a}^{(t-1)} + \frac{\eta}{1-\gamma} d_{\pi_{t-1}}^{\rho_0}(s) \pi_{\boldsymbol{\theta}_{t-1}}(a|s) \left( A_{\pi_{t-1}}(s,a) - \boldsymbol{\lambda}_{t-1}^{\top} \mathbf{a}_{\pi_{t-1}}^c(s,a) \right), \tag{74}$$

where $\theta_{s,a}^{(t)} = \boldsymbol{\theta}_t[s,a]$.

To short the expression, we use the following notations:

$$\pi_t(a|s) := \pi_{\boldsymbol{\theta}_t}(a|s) = \frac{\exp\left\{ \theta_{s,a}^{(t)} \right\}}{\sum_{\tilde{a} \in \mathcal{A}} \exp\left\{ \theta_{s,\tilde{a}}^{(t)} \right\}}, \text{ and } \pi_t := \pi_{\boldsymbol{\theta}_t}. \tag{75}$$

# D  PROOF OF THEOREM 2

In this section, we provide the necessary proof details of Theorem 2. It is very technical to achieve the result of Theorem 2, we outline some necessary intermediate results in Section D.1 where we provide some basic lemmas in Section D.1, and the proof of Theorem 2 is shown in Section D.2.

## D.1  AUXILIARY LEMMAS

**Lemma 6.** *The sequence $\{\boldsymbol{\lambda}_t, \boldsymbol{\theta}_t\}_{t \geq 0}$ is generated by (71)-(72)/(74), and the softmax policy $\pi_t :=$ $\pi_{\boldsymbol{\theta}_t}$ is defined as (75), then update rule with respect to $\pi_t$ equals to*

$$\pi_t(a|s) = \pi_{t-1}(a|s) \frac{\exp\left\{\frac{\eta}{1-\gamma} d_{\pi_{t-1}}^{\rho_0}(s)\pi_{t-1}(a|s)A^{(t-1)}(s,a)\right\}}{Z_{t-1}(s)}, \tag{76}$$

*where $A^{(t-1)}(s,a) = A_{\pi_{t-1}}(s,a) - \boldsymbol{\lambda}_{t-1}^\top \mathbf{a}_{\pi_{t-1}}^c(s,a)$, and*

$$Z_{t-1}(s) = \sum_{\tilde{a} \in \mathcal{A}} \left( \pi_{t-1}(\tilde{a}|s) \exp\left\{ \frac{\eta}{1-\gamma} d_{\pi_{t-1}}^{\rho_0}(s)\pi_{t-1}(\tilde{a}|s) \left( A_{\pi_{t-1}}(s,\tilde{a}) - \boldsymbol{\lambda}_{t-1}^\top \mathbf{a}_{\pi_{t-1}}^c(s,\tilde{a}) \right) \right\} \right).$$

*Proof.* According to the update rules (74), we calculate policy $\pi_t(a|s)$ (75) as follows,

$$\pi_t(a|s) = \pi_{\boldsymbol{\theta}_t}(a|s) = \frac{\exp\left\{\theta_{s,a}^{(t)}\right\}}{\sum_{\tilde{a} \in \mathcal{A}} \exp\left\{\theta_{s,\tilde{a}}^{(t)}\right\}}$$

$$\overset{(74)}{=} \frac{\exp\left\{\theta_{s,a}^{(t-1)} + \frac{\eta}{1-\gamma} d_{\pi_{t-1}}^{\rho_0}(s)\pi_{t-1}(a|s)\left(A_{\pi_{t-1}}(s,a) - \boldsymbol{\lambda}_{t-1}^\top \mathbf{a}_{\pi_{t-1}}^c(s,a)\right)\right\}}{\sum_{\tilde{a} \in \mathcal{A}} \exp\left\{\theta_{s,a}^{(t-1)} + \frac{\eta}{1-\gamma} d_{\pi_{t-1}}^{\rho_0}(s)\pi_{t-1}(a|s)\left(A_{\pi_{t-1}}(s,a) - \boldsymbol{\lambda}_{t-1}^\top \mathbf{a}_{\pi_{t-1}}^c(s,a)\right)\right\}}$$

$$= \frac{\exp\left\{\theta_{s,a}^{(t-1)}\right\}\exp\left\{\frac{\eta}{1-\gamma} d_{\pi_{t-1}}^{\rho_0}(s)\pi_{t-1}(a|s)\left(A_{\pi_{t-1}}(s,a) - \boldsymbol{\lambda}_{t-1}^\top \mathbf{a}_{\pi_{t-1}}^c(s,a)\right)\right\}}{\sum_{\tilde{a} \in \mathcal{A}} \left( \exp\left\{\theta_{s,\tilde{a}}^{(t-1)}\right\}\exp\left\{\frac{\eta}{1-\gamma} d_{\pi_{t-1}}^{\rho_0}(s)\pi_{t-1}(\tilde{a}|s)\left(A_{\pi_{t-1}}(s,\tilde{a}) - \boldsymbol{\lambda}_{t-1}^\top \mathbf{a}_{\pi_{t-1}}^c(s,\tilde{a})\right)\right\} \right)}$$

$$= \pi_{t-1}(a|s) \frac{\sum_{\tilde{a} \in \mathcal{A}} \exp\left\{\theta_{s,\tilde{a}}^{(t-1)}\right\}\exp\left\{\frac{\eta}{1-\gamma} d_{\pi_{t-1}}^{\rho_0}(s)\pi_{t-1}(a|s)\left(A_{\pi_{t-1}}(s,a) - \boldsymbol{\lambda}_{t-1}^\top \mathbf{a}_{\pi_{t-1}}^c(s,a)\right)\right\}}{\sum_{\tilde{a} \in \mathcal{A}} \left( \exp\left\{\theta_{s,\tilde{a}}^{(t-1)}\right\}\exp\left\{\frac{\eta}{1-\gamma} d_{\pi_{t-1}}^{\rho_0}(s)\pi_{t-1}(\tilde{a}|s)\left(A_{\pi_{t-1}}(s,\tilde{a}) - \boldsymbol{\lambda}_{t-1}^\top \mathbf{a}_{\pi_{t-1}}^c(s,\tilde{a})\right)\right\} \right)} \tag{77}$$

$$= \pi_{t-1}(a|s) \frac{\exp\left\{\frac{\eta}{1-\gamma} d_{\pi_{t-1}}^{\rho_0}(s)\pi_{t-1}(a|s)\left(A_{\pi_{t-1}}(s,a) - \boldsymbol{\lambda}_{t-1}^\top \mathbf{a}_{\pi_{t-1}}^c(s,a)\right)\right\}}{\sum_{\tilde{a} \in \mathcal{A}} \left( \frac{\exp\left\{\theta_{s,\tilde{a}}^{(t-1)}\right\}}{\sum_{\tilde{a} \in \mathcal{A}} \exp\left\{\theta_{s,\tilde{a}}^{(t-1)}\right\}} \exp\left\{\frac{\eta}{1-\gamma} d_{\pi_{t-1}}^{\rho_0}(s)\pi_{t-1}(\tilde{a}|s)\left(A_{\pi_{t-1}}(s,\tilde{a}) - \boldsymbol{\lambda}_{t-1}^\top \mathbf{a}_{\pi_{t-1}}^c(s,\tilde{a})\right)\right\} \right)}$$

$$= \pi_{t-1}(a|s) \frac{\exp\left\{\frac{\eta}{1-\gamma} d_{\pi_{t-1}}^{\rho_0}(s)\pi_{t-1}(a|s)\left(A_{\pi_{t-1}}(s,a) - \boldsymbol{\lambda}_{t-1}^\top \mathbf{a}_{\pi_{t-1}}^c(s,a)\right)\right\}}{\sum_{\tilde{a} \in \mathcal{A}} \left( \pi_{t-1}(\tilde{a}|s) \exp\left\{\frac{\eta}{1-\gamma} d_{\pi_{t-1}}^{\rho_0}(s)\pi_{t-1}(\tilde{a}|s)\left(A_{\pi_{t-1}}(s,\tilde{a}) - \boldsymbol{\lambda}_{t-1}^\top \mathbf{a}_{\pi_{t-1}}^c(s,\tilde{a})\right)\right\} \right)}, \tag{78}$$

where Eq.(77) holds since we use the following definition

$$\pi_{t-1}(a|s) = \frac{\exp\left\{\theta_{s,a}^{(t-1)}\right\}}{\sum_{\tilde{a} \in \mathcal{A}} \exp\left\{\theta_{s,\tilde{a}}^{(t-1)}\right\}}.$$

To short the expression, we introduce two notations:

$$Z_{t-1}(s) = \sum_{\tilde{a} \in \mathcal{A}} \left( \pi_{t-1}(\tilde{a}|s) \exp \left\{ \frac{\eta}{1-\gamma} d_{\pi_{t-1}}^{\rho_0}(s) \pi_{t-1}(\tilde{a}|s) \left( A_{\pi_{t-1}}(s, \tilde{a}) - \boldsymbol{\lambda}_{t-1}^\top \mathbf{a}_{\pi_{t-1}}^c(s, \tilde{a}) \right) \right\} \right),$$
(79)

$$A^{(t-1)}(s, a) = A_{\pi_{t-1}}(s, a) - \boldsymbol{\lambda}_{t-1}^\top \mathbf{a}_{\pi_{t-1}}^c(s, a),$$
(80)

we rewrite (78) as follows

$$\pi_t(a|s) = \pi_{t-1}(a|s) \frac{\exp \left\{ \frac{\eta}{1-\gamma} d_{\pi_{t-1}}^{\rho_0}(s) \pi_{t-1}(a|s) A^{(t-1)}(s, a) \right\}}{Z_{t-1}(s)}.$$
(81)

This concludes the proof of Lemma 6. $\qquad \square$

Before we further discussions, we need to define some a notations: $\mathbb{H}_t(s)$ and a distance $\mathbb{D}[\cdot \| \cdot]$ between function $p(\cdot)$ and function $q(\cdot)$ over the action space $\mathcal{A}$,

$$\mathbb{H}_t(s) = \sum_{a \in \mathcal{A}} \frac{\pi_{t+1}(a|s)}{\pi_t(a|s)} \log \frac{\pi_{t+1}(a|s)}{\pi_t(a|s)}, \quad \mathbb{D}[p(\cdot) \| q(\cdot)] = \sum_{a \in \mathcal{A}} p(a) \log \frac{p(a)}{q(a)}.$$

Note that if $p(\cdot), q(\cdot)$ is reduced to probability distributions, then $\mathbb{D}(\cdot \| \cdot)$ is reduced to Kullback-Leibler divergence between the two distributions $p(\cdot), q(\cdot)$:

$$\mathbb{D}[p(\cdot) \| q(\cdot)] = \mathbb{E}_{x \sim p(\cdot)} \left[ \log \frac{p(x)}{q(x)} \right] := \mathbb{KL}[p(\cdot) \| q(\cdot)].$$

**Lemma 7.** *The sequence $\{\boldsymbol{\lambda}_t, \boldsymbol{\theta}_t\}_{t \geq 0}$ is generated by (71)-(72)/(74), and the softmax policy $\pi_t := \pi_{\boldsymbol{\theta}_t}$ is defined as (75), the $\pi_t$ and $\boldsymbol{\lambda}_t$ satisfy the following equation*

$$\mathbf{c}(\pi) = (C_1(\pi), C_2(\pi), \cdots, C_m(\pi))^\top,$$
$$\mathbf{a}_\pi^c(s, a) = (A_\pi^{c_1}(s, a), A_\pi^{c_2}(s, a), \cdots, A_\pi^{c_m}(s, a))^\top.$$

*Furthermore, applying Lemma 1 again, we rewrite (85) as follows*

$$J(\pi_\star) - J(\pi_t) = \frac{1}{\eta} \sum_{s \in \mathcal{S}} \frac{d_{\pi_\star}^{\rho_0}(s)}{d_{\pi_t}^{\rho_0}(s)} \left( \sum_{a \in \mathcal{A}} \frac{\pi_\star(a|s)}{\pi_t(a|s)} \log \frac{\pi_{t+1}(a|s)}{\pi_t(a|s)} \right)$$
$$+ \frac{1}{\eta} \sum_{s \in \mathcal{S}} \sum_{a \in \mathcal{A}} \frac{d_{\pi_\star}^{\rho_0}(s) \pi_\star(a|s)}{d_{\pi_t}^{\rho_0}(s) \pi_t(a|s)} \log Z_t(s) + \boldsymbol{\lambda}_t^\top (\mathbf{c}(\pi_\star) - \mathbf{c}(\pi_t)),$$
(82)

*where $\pi_\star$ is the optimal policy of primal problem (4), i.e., $\pi_\star = \arg \max_{\pi \in \Pi_C} J(\pi)$.*

*Proof.* According to Lemma 1, we calculate the performance difference between $J(\pi_\star)$ and $J(\pi_t)$ as follows,

$$J(\pi_\star) - J(\pi_t) = \frac{1}{1-\gamma} \mathbb{E}_{s \sim d_{\pi_\star}^{\rho_0}(\cdot), a \sim \pi_\star(\cdot|s)} [A_{\pi_t}(s, a)] = \frac{1}{1-\gamma} \sum_{s \in \mathcal{S}} \sum_{a \in \mathcal{A}} d_{\pi_\star}^{\rho_0}(s) \pi_\star(a|s) A_{\pi_t}(s, a).$$
(83)

Recall Eq.(80) and Eq.(81), we write it as follows,

$$\log \left( \frac{\pi_{t+1}(a|s)}{\pi_t(a|s)} Z_t(s) \right) = \frac{\eta}{1-\gamma} d_{\pi_t}^{\rho_0}(s) \pi_t(a|s) \left( A_{\pi_t}(s, a) - \boldsymbol{\lambda}_t^\top \mathbf{a}_{\pi_t}^c(s, a) \right).$$

Then, we rewrite the term $A_{\pi_t}(s, a)$ as follows,

$$A_{\pi_t}(s, a) = \frac{1-\gamma}{\eta} \cdot \frac{1}{d_{\pi_t}^{\rho_0}(s) \pi_t(a|s)} \cdot \log \left( \frac{\pi_{t+1}(a|s)}{\pi_t(a|s)} Z_t(s) \right) + \boldsymbol{\lambda}_t^\top \mathbf{a}_{\pi_t}^c(s, a).$$
(84)

Taking the results (84) into (83), we rewrite the performance difference between $J(\pi_\star)$ and $J(\pi_t)$ as follows,

$$
J(\pi_\star) - J(\pi_t) = \frac{1}{\eta} \sum_{s \in \mathcal{S}} \frac{d_{\pi_\star}^{\rho_0}(s)}{d_{\pi_t}^{\rho_0}(s)} \sum_{a \in \mathcal{A}} \frac{\pi_\star(a|s)}{\pi_t(a|s)} \cdot \log \left( \frac{\pi_{t+1}(a|s)}{\pi_t(a|s)} Z_t(s) \right)
$$

$$
+ \frac{1}{1 - \gamma} \sum_{s \in \mathcal{S}} \sum_{a \in \mathcal{A}} d_{\pi_\star}^{\rho_0}(s) \pi_\star(a|s) \boldsymbol{\lambda}_t^\top \mathbf{a}_{\pi_t}^c(s, a)
$$

$$
= \frac{1}{\eta} \sum_{s \in \mathcal{S}} \frac{d_{\pi_\star}^{\rho_0}(s)}{d_{\pi_t}^{\rho_0}(s)} \left( \sum_{a \in \mathcal{A}} \frac{\pi_\star(a|s)}{\pi_t(a|s)} \log \frac{\pi_{t+1}(a|s)}{\pi_t(a|s)} \right)
$$

$$
+ \frac{1}{\eta} \sum_{s \in \mathcal{S}} \sum_{a \in \mathcal{A}} \frac{d_{\pi_\star}^{\rho_0}(s) \pi_\star(a|s)}{d_{\pi_t}^{\rho_0}(s) \pi_t(a|s)} \log Z_t(s) + \frac{1}{1 - \gamma} \sum_{s \in \mathcal{S}} \sum_{a \in \mathcal{A}} d_{\pi_\star}^{\rho_0}(s) \pi_\star(a|s) \boldsymbol{\lambda}_t^\top \mathbf{a}_{\pi_t}^c(s, a). \quad (85)
$$

Recall the vectors $\mathbf{c}(\pi), \mathbf{a}_\pi^c(s, a)$ defined in previous sections, where

$$
\mathbf{c}(\pi) = (C_1(\pi), C_2(\pi), \cdots, C_m(\pi))^\top,
$$

$$
\mathbf{a}_\pi^c(s, a) = (A_\pi^{c_1}(s, a), A_\pi^{c_2}(s, a), \cdots, A_\pi^{c_m}(s, a))^\top.
$$

Furthermore, let us apply Lemma 1 again, we obtain

$$
\mathbf{c}(\pi_\star) - \mathbf{c}(\pi_t) = \frac{1}{1 - \gamma} \sum_{s \in \mathcal{S}} \sum_{a \in \mathcal{A}} d_{\pi_\star}^{\rho_0}(s) \pi_\star(a|s) \mathbf{a}_{\pi_t}^c(s, a),
$$

which implies,

$$
J(\pi_\star) - J(\pi_t) = \frac{1}{\eta} \sum_{s \in \mathcal{S}} \frac{d_{\pi_\star}^{\rho_0}(s)}{d_{\pi_t}^{\rho_0}(s)} \left( \sum_{a \in \mathcal{A}} \frac{\pi_\star(a|s)}{\pi_t(a|s)} \log \frac{\pi_{t+1}(a|s)}{\pi_t(a|s)} \right)
$$

$$
+ \frac{1}{\eta} \sum_{s \in \mathcal{S}} \sum_{a \in \mathcal{A}} \frac{d_{\pi_\star}^{\rho_0}(s) \pi_\star(a|s)}{d_{\pi_t}^{\rho_0}(s) \pi_t(a|s)} \log Z_t(s) + \boldsymbol{\lambda}_t^\top (\mathbf{c}(\pi_\star) - \mathbf{c}(\pi_t)).
$$

This concludes the proof of the result (82). $\qquad\square$

Similarly, we obtain the performance difference between $J(\pi_{t+1})$ and $J(\pi_t)$.

**Lemma 8.** *The performance difference between $J(\pi_{t+1})$ and $J(\pi_t)$ satisfies the following equation*

$$
J(\pi_{t+1}) - J(\pi_t) - \boldsymbol{\lambda}_t^\top (\mathbf{c}(\pi_{t+1}) - \mathbf{c}(\pi_t)) \quad (86)
$$

$$
= \frac{1}{\eta} \sum_{s \in \mathcal{S}} \frac{d_{\pi_{t+1}}^{\rho_0}(s)}{d_{\pi_t}^{\rho_0}(s)} \mathbb{H}_t(s) + \frac{1}{\eta} \sum_{s \in \mathcal{S}} \sum_{a \in \mathcal{A}} \frac{d_{\pi_{t+1}}^{\rho_0}(s) \pi_{t+1}(a|s)}{d_{\pi_t}^{\rho_0}(s) \pi_t(a|s)} \log Z_t(s).
$$

*Proof.* According to Lemma 1, we calculate the performance difference between $J(\pi_{t+1})$ and $J(\pi_t)$ as follows,

$$
J(\pi_{t+1}) - J(\pi_t) = \frac{1}{1 - \gamma} \mathbb{E}_{s \sim d_{\pi_{t+1}}^{\rho_0}(\cdot), a \sim \pi_{t+1}(\cdot|s)} [A_{\pi_t}(s, a)] \quad (87)
$$

$$
= \frac{1}{1 - \gamma} \sum_{s \in \mathcal{S}} \sum_{a \in \mathcal{A}} d_{\pi_{t+1}}^{\rho_0}(s) \pi_{t+1}(a|s) A_{\pi_t}(s, a). \quad (88)
$$

Recall Eq.(80) and Eq.(81), we have

$$
A_{\pi_t}(s, a) = \frac{1 - \gamma}{\eta} \cdot \frac{1}{d_{\pi_t}^{\rho_0}(s) \pi_t(a|s)} \cdot \log \left( \frac{\pi_{t+1}(a|s)}{\pi_t(a|s)} Z_t(s) \right) + \boldsymbol{\lambda}_t^\top \mathbf{a}_{\pi_t}^c(s, a). \quad (89)
$$

Taking (89) to (88), we have

$$
\begin{aligned}
J(\pi_{t+1}) - J(\pi_t) &= \frac{1}{\eta} \sum_{s \in \mathcal{S}} \frac{d_{\pi_{t+1}}^{\rho_0}(s)}{d_{\pi_t}^{\rho_0}(s)} \sum_{a \in \mathcal{A}} \left( \frac{\pi_{t+1}(a|s)}{\pi_t(a|s)} \cdot \log \frac{\pi_{t+1}(a|s)}{\pi_t(a|s)} + \frac{\pi_{t+1}(a|s)}{\pi_t(a|s)} \cdot \log Z_t(s) \right) \\
&\quad + \frac{1}{1-\gamma} \sum_{s \in \mathcal{S}} \sum_{a \in \mathcal{A}} d_{\pi_{t+1}}^{\rho_0}(s) \pi_{t+1}(a|s) \boldsymbol{\lambda}_t^\top \mathbf{a}_{\pi_t}^c(s, a) \\
&= -\frac{1}{\eta} \sum_{s \in \mathcal{S}} \frac{d_{\pi_{t+1}}^{\rho_0}(s)}{d_{\pi_t}^{\rho_0}(s)} \mathbb{H}_t(s) + \frac{1}{\eta} \sum_{s \in \mathcal{S}} \sum_{a \in \mathcal{A}} \frac{d_{\pi_{t+1}}^{\rho_0}(s) \pi_{t+1}(a|s)}{d_{\pi_t}^{\rho_0}(s) \pi_t(a|s)} \log Z_t(s) + \boldsymbol{\lambda}_t^\top (\mathbf{c}(\pi_{t+1}) - \mathbf{c}(\pi_t)).
\end{aligned}
$$

This concludes the proof of the result (86). $\qquad\square$

**Lemma 9.** *The sequence $\{\boldsymbol{\lambda}_t, \boldsymbol{\theta}_t\}_{t \geq 0}$ is generated by (71)-(72)/(74), and the softmax policy $\pi_t := \pi_{\boldsymbol{\theta}_t}$ is defined as (75), then the performance difference between $J(\pi_{t+1})$ and $J(\pi_t)$ satisfies*

$$
J(\pi_{t+1}) - J(\pi_t) - \boldsymbol{\lambda}_t^\top (\mathbf{c}(\pi_{t+1}) - \mathbf{c}(\pi_t)) \geq \frac{1}{\eta} \sum_{s \in \mathcal{S}} \sum_{a \in \mathcal{A}} \frac{d_{\pi_{t+1}}^{\rho_0}(s) \pi_{t+1}(a|s)}{d_{\pi_t}^{\rho_0}(s) \pi_t(a|s)} \log Z_t(s), \qquad (90)
$$

*and $\log Z_t(s) \geq 0$.*

*Proof.* Recall

$$
\mathbb{H}_t(s) = \sum_{a \in \mathcal{A}} \frac{\pi_{t+1}(a|s)}{\pi_t(a|s)} \log \frac{\pi_{t+1}(a|s)}{\pi_t(a|s)},
$$

since $\pi_t(a|s) \leq 1$, then

$$
\begin{aligned}
\mathbb{H}_t(s) &= \sum_{a \in \mathcal{A}} \frac{\pi_{t+1}(a|s)}{\pi_t(a|s)} \log \frac{\pi_{t+1}(a|s)}{\pi_t(a|s)} \\
&\geq \sum_{a \in \mathcal{A}} \pi_{t+1}(a|s) \log \frac{\pi_{t+1}(a|s)}{\pi_t(a|s)} = \mathbb{KL}\left[\pi_{t+1}(\cdot|s) \| \pi_t(\cdot|s)\right] \geq 0. \qquad (91)
\end{aligned}
$$

According to Eq.(86), and due to positivity of the terms $d_{\pi_t}^{\rho_0}(s)$ and $d_{\pi_{t+1}}^{\rho_0}(s)$, we have

$$
J(\pi_{t+1}) - J(\pi_t) - \boldsymbol{\lambda}_t^\top (\mathbf{c}(\pi_{t+1}) - \mathbf{c}(\pi_t)) \geq \frac{1}{\eta} \sum_{s \in \mathcal{S}} \sum_{a \in \mathcal{A}} \frac{d_{\pi_{t+1}}^{\rho_0}(s) \pi_{t+1}(a|s)}{d_{\pi_t}^{\rho_0}(s) \pi_t(a|s)} \log Z_t(s).
$$

Now, we need to show $\log Z_t(s) \geq 0$. In fact, the following holds

$$
\begin{aligned}
\log Z_t(s) &= \log \left( \sum_{\tilde{a} \in \mathcal{A}} \pi_t(\tilde{a}|s) \exp\left\{ \frac{\eta}{1-\gamma} d_{\pi_t}^{\rho_0}(s) \pi_t(\tilde{a}|s) \left( A_{\pi_t}(s, \tilde{a}) - \boldsymbol{\lambda}_t^\top \mathbf{a}_{\pi_t}^c(s, \tilde{a}) \right) \right\} \right) \\
&\geq \sum_{\tilde{a} \in \mathcal{A}} \pi_t(\tilde{a}|s) \log \left( \exp\left\{ \frac{\eta}{1-\gamma} d_{\pi_t}^{\rho_0}(s) \pi_t(\tilde{a}|s) \left( A_{\pi_t}(s, \tilde{a}) - \boldsymbol{\lambda}_t^\top \mathbf{a}_{\pi_t}^c(s, \tilde{a}) \right) \right\} \right) \qquad (92) \\
&= \frac{\eta}{1-\gamma} \sum_{\tilde{a} \in \mathcal{A}} \pi_t(\tilde{a}|s) d_{\pi_t}^{\rho_0}(s) \pi_t(\tilde{a}|s) \left( A_{\pi_t}(s, \tilde{a}) - \boldsymbol{\lambda}_t^\top \mathbf{a}_{\pi_t}^c(s, \tilde{a}) \right) \\
&\geq \min_{a \in \mathcal{A}} \{\pi_t(a|s)\} \cdot \frac{\eta}{1-\gamma} \sum_{\tilde{a} \in \mathcal{A}} d_{\pi_t}^{\rho_0}(s) \pi_t(\tilde{a}|s) \left( A_{\pi_t}(s, \tilde{a}) - \boldsymbol{\lambda}_t^\top \mathbf{a}_{\pi_t}^c(s, \tilde{a}) \right) \\
&\geq \min_{a \in \mathcal{A}} \{\pi_t(a|s)\} \cdot \frac{\eta}{1-\gamma} d_{\pi_t}^{\rho_0}(s) \left( \sum_{\tilde{a} \in \mathcal{A}} \pi_t(\tilde{a}|s) A_{\pi_t}(s, \tilde{a}) - \sum_{\tilde{a} \in \mathcal{A}} \pi_t(\tilde{a}|s) \boldsymbol{\lambda}_t^\top \mathbf{a}_{\pi_t}^c(s, \tilde{a}) \right) \\
&= 0, \qquad (93)
\end{aligned}
$$

where Eq.(92) holds due to the Jensen's inequality, and the last equation (93) holds since:

$$\sum_{\tilde{a}\in\mathcal{A}} \pi_t(\tilde{a}|s)A_{\pi_t}(s,\tilde{a}) = 0,$$

$$\sum_{\tilde{a}\in\mathcal{A}} \pi_t(\tilde{a}|s)A_{\pi_t}^{c_i}(s,\tilde{a}) = 0, \ \ i = 1,2,\cdots,m, \tag{94}$$

$$\sum_{\tilde{a}\in\mathcal{A}} \pi_t(\tilde{a}|s)\mathbf{a}_{\pi_t}^c(s,\tilde{a}) = \left( \underbrace{\sum_{\tilde{a}\in\mathcal{A}} \pi_t(\tilde{a}|s)A_{\pi_t}^{c_1}(s,\tilde{a})}_{\overset{(94)}{=}0}, \underbrace{\sum_{\tilde{a}\in\mathcal{A}} \pi_t(\tilde{a}|s)A_{\pi_t}^{c_2}(s,\tilde{a})}_{=0}, \cdots, \underbrace{\sum_{\tilde{a}\in\mathcal{A}} \pi_t(\tilde{a}|s)A_{\pi_t}^{c_m}(s,\tilde{a})}_{=0} \right)^{\top}$$

$$= \mathbf{0}_m^{\top},$$

the notation $\mathbf{0}_m^{\top} \in \mathbb{R}^{m\times 1}$ denotes a vector with the elements are all zero. $\qquad\square$

**Lemma 10.** *The term $\sum_{s\in\mathcal{S}} \log Z_t(s)$ is bounded as follows,*

$$\sum_{s\in\mathcal{S}} \log Z_t(s) \leq \frac{\eta}{(1-\gamma)\rho_{\min}} \left( J(\pi_{t+1}) - J(\pi_t) - \boldsymbol{\lambda}_t^{\top}(\mathbf{c}(\pi_{t+1}) - \mathbf{c}(\pi_t)) \right).$$

*Proof.* Since $d_{\pi_t}^{\rho_0}(s) \leq 1, \pi_t(a|s) \leq 1$ always holds, recall Eq.(90) in Lemma 9, we achieve the following inequality:

$$J(\pi_{t+1}) - J(\pi_t) - \boldsymbol{\lambda}_t^{\top}(\mathbf{c}(\pi_{t+1}) - \mathbf{c}(\pi_t)) \geq \frac{1}{\eta}\sum_{s\in\mathcal{S}}\sum_{a\in\mathcal{A}} d_{\pi_{t+1}}^{\rho_0}(s)\pi_{t+1}(a|s)\log Z_t(s). \tag{95}$$

According to Eq.(143), we rewrite Eq.(95) as follows

$$J(\pi_{t+1}) - J(\pi_t) - \boldsymbol{\lambda}_t^{\top}(\mathbf{c}(\pi_{t+1}) - \mathbf{c}(\pi_t))$$

$$\geq \frac{1-\gamma}{\eta}\sum_{s\in\mathcal{S}}\sum_{a\in\mathcal{A}} \rho_0(s)\pi_{t+1}(a|s)\log Z_t(s) \tag{96}$$

$$= \frac{1-\gamma}{\eta}\sum_{s\in\mathcal{S}} \rho_0(s)\log Z_t(s) \sum_{a\in\mathcal{A}} \pi_{t+1}(a|s) = \frac{1-\gamma}{\eta}\sum_{s\in\mathcal{S}} \rho_0(s)\log Z_t(s). \tag{97}$$

Then, under Assumption 2, for each $s \in \mathcal{S}$, $\rho_{\min} \leq \rho_0(s)$, result (97) implies

$$\frac{\eta}{(1-\gamma)\rho_{\min}} \left( J(\pi_{t+1}) - J(\pi_t) - \boldsymbol{\lambda}_t^{\top}(\mathbf{c}(\pi_{t+1}) - \mathbf{c}(\pi_t)) \right) \geq \sum_{s\in\mathcal{S}} \log Z_t(s). \tag{98}$$

$\qquad\square$

**Lemma 11.** *Let $\pi_{\boldsymbol{\theta}}$ and $\pi_{\boldsymbol{\theta}'}$ be softmax policy, then the following holds*

$$\left| V_{\pi_{\boldsymbol{\theta}'}}(s) - V_{\pi_{\boldsymbol{\theta}}}(s) - \left\langle \frac{\partial V_{\pi_{\boldsymbol{\theta}}}(s)}{\partial \boldsymbol{\theta}}, \boldsymbol{\theta}' - \boldsymbol{\theta} \right\rangle \right| \leq \frac{4}{(1-\gamma)^3}\|\boldsymbol{\theta}' - \boldsymbol{\theta}\|_2^2.$$

*Proof.* See (Mei et al., 2020, Lemma 7). $\qquad\square$

Let $\beta = \frac{4}{(1-\gamma)^3}$, and if $\boldsymbol{\theta}' = \boldsymbol{\theta} + \frac{1}{\beta}\frac{\partial V_{\pi_{\boldsymbol{\theta}}}(s)}{\partial \boldsymbol{\theta}}$, then we obtain

$$V_{\pi_{\boldsymbol{\theta}}}(s) - V_{\pi_{\boldsymbol{\theta}'}}(s) \leq -\frac{1}{2\beta}\left\| \frac{\partial V_{\pi_{\boldsymbol{\theta}}}(s)}{\partial \boldsymbol{\theta}} \right\|_2^2. \tag{99}$$

Let $\mathbf{q}_\pi^c(s,a) \in \mathbb{R}^m$ is defined as follows,

$$\mathbf{q}_\pi^c(s,a) = (Q_\pi^{c_2}(s,a), Q_\pi^{c_2}(s,a), \cdots, Q_\pi^{c_m}(s,a)), \ \mathbf{v}_\pi^c(s) = (V_\pi^{c_2}(s,a), V_\pi^{c_2}(s), \cdots, V_\pi^{c_m}(s))$$

$$Q_{\pi_\theta}(s, a, \boldsymbol{\lambda}) = Q_{\pi_\theta}(s, a) - \boldsymbol{\lambda}^\top \mathbf{q}_{\pi_\theta}^c(s), \quad V_{\pi_\theta}(s, \boldsymbol{\lambda}) = V_{\pi_\theta}(s) - \boldsymbol{\lambda}^\top \mathbf{v}_{\pi_\theta}^c(s).$$

Furthermore, we define

$$\Delta_\star^r(s) = \max_{\pi \in \Pi_C}\{Q_\pi(s, a)\} - \max_{\pi \in \Pi_C, a \neq a_\star(s)}\{Q_\pi(s, a)\} \tag{100}$$

$$= Q_\star(s, a_\star(s)) - \max_{\pi \in \Pi_C, a \neq a_\star(s)}\{Q_\pi(s, a)\} > 0. \tag{101}$$

Similarly, we define

$$\Delta_\star^{c_i}(s) = \min_{\pi \in \Pi_C}\{Q_\pi^{c_i}(s, a)\} - \min_{\pi \in \Pi_C, a \neq a_\star(s)}\{Q_\pi^{c_i}(s, a)\} \tag{102}$$

$$= Q_\star^{c_i}(s, a_\star(s)) - \min_{\pi \in \Pi_C, a \neq a_\star(s)}\{Q_\pi^{c_i}(s, a)\} > 0. \tag{103}$$

We define $\Delta_\star(s)$ as follows

$$\Delta_\star(s) = \Delta_\star^r(s) + \boldsymbol{\lambda}^\top (\Delta_\star^{c_1}(s), \Delta_\star^{c_2}(s), \cdots, \Delta_\star^{c_m}(s))^\top. \tag{104}$$

Recall the notation $\theta_{s,a} = \boldsymbol{\theta}[s, a]$, for all $(s, a) \in \mathcal{S} \times \mathcal{A}$, and we define

$$\Theta_1(s) = \left\{ \boldsymbol{\theta} : \frac{\partial \mathcal{L}(\pi_\theta, \boldsymbol{\lambda})}{\partial \boldsymbol{\theta}[s, a_\star(s)]} \geq \frac{\partial \mathcal{L}(\pi_\theta, \boldsymbol{\lambda})}{\partial \boldsymbol{\theta}[s, a]}, \ \forall \, a \neq a_\star \right\}, \tag{105}$$

$$\Theta_2(s) = \left\{ \boldsymbol{\theta} : \begin{array}{c} Q_{\pi_\theta}(s, a_\star(s)) \geq Q_\star(s, a_\star(s)) - \frac{1}{2}\Delta_\star^r \\ -Q_{\pi_\theta}^{c_i}(s, a_\star(s)) \geq -Q_\star^{c_i}(s, a_\star(s)) - \frac{1}{2}\Delta_\star^{c_i}, i = 1, 2, \cdots, m \end{array} \right\}, \tag{106}$$

$$\Theta_3(s) = \bigcap_{i=1}^m \left\{ \boldsymbol{\theta}_t : \begin{array}{c} V_{\pi_{\theta_t}}(s) \geq Q_{\pi_{\theta_t}}(s, a_\star(s)) - \frac{1}{2}\Delta_\star^r \\ -V_{\pi_{\theta_t}}^{c_i}(s) \geq -Q_{\pi_{\theta_t}}^{c_i}(s, a_\star(s)) - \frac{1}{2}\Delta_\star^{c_i} \end{array} \middle| \text{for } t \geq 0 \text{ is large enough} \right\}, \tag{107}$$

$$\Theta_c(s) = \left\{ \boldsymbol{\theta} : \pi_\theta(a_\star(s)|s) \geq \frac{c(s)}{c(s) + 1}, \ c(s) + 1 = \frac{|\mathcal{A}|}{(1 - \gamma)\Delta_\star(s)} \right\}. \tag{108}$$

**Lemma 12.** *Let $\boldsymbol{\theta}_t \in \Theta_1(s) \cap \Theta_2(s) \cap \Theta_3(s)$, then*

$$\boldsymbol{\theta}_{t+1} \in \Theta_1(s) \cap \Theta_2(s) \cap \Theta_3(s).$$

*Proof.* $\boldsymbol{\theta}_{t+1} \in \Theta_2(s)$

Since $\boldsymbol{\theta}_t \in \Theta_3(s)$, we obtain the following equation

$$Q_{\pi_t}(s, a_\star(s), \boldsymbol{\lambda}) \geq Q_\star(s, a_\star(s), \boldsymbol{\lambda}) - \frac{1}{2}\Delta_\star(s),$$

where $\pi_t$ is short for $\pi_{\boldsymbol{\theta}_t}$, and $Q_\star(s, a, \boldsymbol{\lambda}) = \max_{\pi \in \Pi_C} Q_\pi(s, a, \boldsymbol{\lambda})$.

Furthermore, we obtain

$$Q_{\pi_{t+1}}(s, a_\star(s), \boldsymbol{\lambda}) - Q_{\pi_t}(s, a_\star(s), \boldsymbol{\lambda}) = \gamma \sum_{s' \in \mathcal{S}} \mathbb{P}(s'|s, a_\star(s)) \left( V_{\pi_{t+1}}(s', \boldsymbol{\lambda}) - V_{\pi_t}(s', \boldsymbol{\lambda}) \right). \tag{109}$$

According to Lemma 11, Eq.(99), we know

$$Q_{\pi_{t+1}}(s, a_\star(s), \boldsymbol{\lambda}) - Q_{\pi_t}(s, a_\star(s), \boldsymbol{\lambda}) \geq 0 \geq -\frac{1}{2}\Delta_\star(s), \tag{110}$$

which implies $\boldsymbol{\theta}_{t+1} \in \Theta_2(s)$.

$\boldsymbol{\theta}_{t+1} \in \Theta_3(s)$

For any $a \neq a_\star(s)$, we know

$$Q_{\pi_t}(s, a_\star(s), \boldsymbol{\lambda}) - Q_{\pi_t}(s, a, \boldsymbol{\lambda}) \tag{111}$$

$$= Q_{\pi_t}(s, a_\star(s), \boldsymbol{\lambda}) - Q_\star(s, a_\star(s), \boldsymbol{\lambda}) + Q_\star(s, a_\star(s), \boldsymbol{\lambda}) - Q_{\pi_t}(s, a, \boldsymbol{\lambda}) \tag{112}$$

$$\geq -\frac{1}{2}\Delta_\star(s) + Q_\star(s, a_\star(s), \boldsymbol{\lambda}) - Q_\star(s, a) + Q_\star(s, a) - Q_{\pi_t}(s, a, \boldsymbol{\lambda}) \tag{113}$$

$$\geq -\frac{1}{2}\Delta_\star(s) + Q_\star(s, a_\star(s), \boldsymbol{\lambda}) - \max_{a \neq a_\star(s)} Q_\star(s, a) + Q_\star(s, a) - Q_{\pi_t}(s, a, \boldsymbol{\lambda}) \tag{114}$$

$$\overset{(100),(102),(104)}{=} -\frac{1}{2}\Delta_\star(s) + \Delta_\star(s) + \gamma \sum_{s' \in \mathcal{S}} \mathbb{P}(s'|s, a_\star(s)) \left( V_{\pi_{t+1}}(s', \boldsymbol{\lambda}) - V_{\pi_t}(s', \boldsymbol{\lambda}) \right) \tag{115}$$

$$\geq \frac{1}{2}\Delta_\star(s). \tag{116}$$

Similarly, we obtain

$$Q_{\pi_{t+1}}(s, a_\star(s), \boldsymbol{\lambda}) - Q_{\pi_{t+1}}(s, a, \boldsymbol{\lambda}) \geq \frac{1}{2}\Delta_\star(s), \tag{117}$$

which implies $\boldsymbol{\theta}_{t+1} \in \Theta_3(s)$.

$\boldsymbol{\theta}_{t+1} \in \Theta_1(s)$

According to Proposition 1,

$$\frac{\partial \mathcal{L}(\pi_{\boldsymbol{\theta}}, \boldsymbol{\lambda})}{\partial \theta_{s,a}} = \frac{1}{1 - \gamma} d_{\pi_{\boldsymbol{\theta}}}^{\rho_0}(s) \pi_{\boldsymbol{\theta}}(a|s) A_{\pi_{\boldsymbol{\theta}}}(s, a, \boldsymbol{\lambda}),$$

if $\boldsymbol{\theta}_t \in \Theta_1(s)$, i.e., $\dfrac{\partial \mathcal{L}(\pi_{\boldsymbol{\theta}_t}, \boldsymbol{\lambda}_t)}{\partial \boldsymbol{\theta}_t[s, a_\star(s)]} \geq \dfrac{\partial \mathcal{L}(\pi_{\boldsymbol{\theta}_t}, \boldsymbol{\lambda}_t)}{\partial \boldsymbol{\theta}_t[s, a]}$, we obtain: for $a \neq a_\star$

$$\pi_t(a_\star(s)|s) A_{\pi_t}(s, a_\star(s), \boldsymbol{\lambda}) \geq \pi_t(a|s) A_{\pi_t}(s, a, \boldsymbol{\lambda}),$$

where $\pi_t$ is short for $\pi_{\boldsymbol{\theta}_t}$.

**Case (i)**: $\pi_t(a_\star(s)|s) \geq \pi_t(a|s)$.

If $\pi_t(a_\star(s)|s) \geq \pi_t(a|s)$, according to softmax parameterization (9), we obtain

$$\boldsymbol{\theta}_t[s, a_\star(s)] \geq \boldsymbol{\theta}_t[s, a] \tag{118}$$

Recall the update rule of Algorithm 1, we know

$$\boldsymbol{\theta}_{t+1}[s, a_\star(s)] = \boldsymbol{\theta}_t[s, a_\star(s)] + \eta \frac{\partial \mathcal{L}(\pi_{\boldsymbol{\theta}_t}, \boldsymbol{\lambda}_t)}{\partial \boldsymbol{\theta}_t[s, a_\star(s)]} \overset{(105),(118)}{\geq} \boldsymbol{\theta}_t[s, a] + \eta \frac{\partial \mathcal{L}(\pi_{\boldsymbol{\theta}_t}, \boldsymbol{\lambda}_t)}{\partial \boldsymbol{\theta}_t[s, a]} = \boldsymbol{\theta}_{t+1}[s, a],$$

which implies

$$\pi_{t+1}(a_\star(s)|s) = \frac{\exp\{\boldsymbol{\theta}_t[s, a_\star(s)]\}}{\sum_{a \in \mathcal{A}} \exp\{\boldsymbol{\theta}_t[s, a]\}} \geq \frac{\exp\{\boldsymbol{\theta}_t[s, a]\}}{\sum_{a \in \mathcal{A}} \exp\{\boldsymbol{\theta}_t[s, a]\}} = \pi_{t+1}(a|s). \tag{119}$$

Recall (117), we obtain

$$A_{\pi_{t+1}}(s, a_\star(s), \boldsymbol{\lambda}) \geq A_{\pi_{t+1}}(s, a, \boldsymbol{\lambda}). \tag{120}$$

Eq.(119)-Eq.(120) implies

$$\frac{\partial \mathcal{L}(\pi_{\boldsymbol{\theta}_{t+1}}, \boldsymbol{\lambda}_{t+1})}{\partial \boldsymbol{\theta}_{t+1}[s, a_\star(s)]} \geq \frac{\partial \mathcal{L}(\pi_{\boldsymbol{\theta}_{t+1}}, \boldsymbol{\lambda}_{t+1})}{\partial \boldsymbol{\theta}_{t+1}[s, a]},$$

which implies $\boldsymbol{\theta}_{t+1} \in \Theta_1(s)$.

**Case (ii)**: $\pi_t(a_\star(s)|s) < \pi_t(a|s)$.

According to

$$\pi_t(a_\star(s)|s) A_{\pi_t}(s, a_\star(s), \boldsymbol{\lambda}) \geq \pi_t(a|s) A_{\pi_t}(s, a, \boldsymbol{\lambda}),$$

we obtain

$$\pi_t(a_\star(s)|s)\left(Q_{\pi_t}(s, a_\star(s), \boldsymbol{\lambda}) - V_{\pi_t}(s, \boldsymbol{\lambda})\right) \geq \pi_t(a|s)\left(Q_{\pi_t}(s, a, \boldsymbol{\lambda}) - V_{\pi_t}(s, \boldsymbol{\lambda})\right) \quad (121)$$
$$= \pi_t(a|s)\left(Q_{\pi_t}(s, a_\star(s), \boldsymbol{\lambda}) - V_{\pi_t}(s, \boldsymbol{\lambda}) + Q_{\pi_t}(s, a, \boldsymbol{\lambda}) - Q_{\pi_t}(s, a_\star(s), \boldsymbol{\lambda})\right), \quad (122)$$

i.e.,

$$\left(1 - \frac{\pi_t(a_\star(s)|s)}{\pi_t(a|s)}\right)\left(Q_{\pi_t}(s, a_\star(s), \boldsymbol{\lambda}) - V_{\pi_t}(s, \boldsymbol{\lambda})\right) \quad (123)$$
$$= \left(1 - \exp\left\{\boldsymbol{\theta}_t[s, a_\star(s)] - \boldsymbol{\theta}_t[s, a]\right\}\right)\left(Q_{\pi_t}(s, a_\star(s), \boldsymbol{\lambda}) - V_{\pi_t}(s, \boldsymbol{\lambda})\right) \quad (124)$$
$$\leq Q_{\pi_t}(s, a_\star(s), \boldsymbol{\lambda}) - Q_{\pi_t}(s, a, \boldsymbol{\lambda}). \quad (125)$$

Recall the update rule of Algorithm 1, we know

$$\boldsymbol{\theta}_{t+1}[s, a_\star(s)] = \boldsymbol{\theta}_t[s, a_\star(s)] + \eta\frac{\partial\mathcal{L}(\pi_{\boldsymbol{\theta}_t}, \boldsymbol{\lambda}_t)}{\partial\boldsymbol{\theta}_t[s, a_\star(s)]}, \boldsymbol{\theta}_{t+1}[s, a] = \boldsymbol{\theta}_t[s, a] + \eta\frac{\partial\mathcal{L}(\pi_{\boldsymbol{\theta}_t}, \boldsymbol{\lambda}_t)}{\partial\boldsymbol{\theta}_t[s, a]}. \quad (126)$$

Since $\boldsymbol{\theta}_t \in \Theta_1(s)$, i.e., $\dfrac{\partial\mathcal{L}(\pi_{\boldsymbol{\theta}_t}, \boldsymbol{\lambda}_t)}{\partial\boldsymbol{\theta}_t[s, a_\star(s)]} \geq \dfrac{\partial\mathcal{L}(\pi_{\boldsymbol{\theta}_t}, \boldsymbol{\lambda}_t)}{\partial\boldsymbol{\theta}_t[s, a]}$, Eq.(126) implies

$$\boldsymbol{\theta}_{t+1}[s, a_\star(s)] - \boldsymbol{\theta}_{t+1}[s, a] \geq \boldsymbol{\theta}_t[s, a_\star(s)] - \boldsymbol{\theta}_t[s, a]. \quad (127)$$

Since we consider $\pi_t(a_\star(s)|s) < \pi_t(a|s)$, then

$$1 - \exp\{\boldsymbol{\theta}_t[s, a_\star(s)] - \boldsymbol{\theta}_t[s, a]\} = 1 - \frac{\pi_t(a_\star(s)|s)}{\pi_t(a|s)} > 0,$$

which implies

$$\left(1 - \exp\{\boldsymbol{\theta}_{t+1}[s, a_\star(s)] - \boldsymbol{\theta}_{t+1}[s, a]\}\right)\left(Q_{\pi_{t+1}}(s, a_\star(s), \boldsymbol{\lambda}) - V_{\pi_{t+1}}(s, \boldsymbol{\lambda})\right)$$
$$\leq Q_{\pi_{t+1}}(s, a_\star(s), \boldsymbol{\lambda}) - Q_{\pi_{t+1}}(s, a, \boldsymbol{\lambda}).$$

Rearranging it, we obtain

$$\pi_{t+1}(a_\star(s)|s)A_{\pi_{t+1}}(s, a_\star(s), \boldsymbol{\lambda}) \geq \pi_{t+1}(a|s)A_{\pi_{t+1}}(s, a, \boldsymbol{\lambda}),$$

which is

$$\frac{\partial\mathcal{L}(\pi_{\boldsymbol{\theta}_{t+1}}, \boldsymbol{\lambda}_{t+1})}{\partial\boldsymbol{\theta}_{t+1}[s, a_\star(s)]} \geq \frac{\partial\mathcal{L}(\pi_{\boldsymbol{\theta}_{t+1}}, \boldsymbol{\lambda}_{t+1})}{\partial\boldsymbol{\theta}_{t+1}[s, a]},$$

which implies $\boldsymbol{\theta}_{t+1} \in \Theta_1(s)$. $\qquad\square$

**Lemma 13.** *Let $\boldsymbol{\theta}_t \in \Theta_1(s) \cap \Theta_2(s) \cap \Theta_3(s)$, then $\pi_{t+1}(a_\star(s)|s) \geq \pi_t(a_\star(s)|s)$.*

*Proof.* Recall $\dfrac{\partial\mathcal{L}(\pi_{\boldsymbol{\theta}_t}, \boldsymbol{\lambda}_t)}{\partial\boldsymbol{\theta}_t[s, a_\star(s)]} \geq \dfrac{\partial\mathcal{L}(\pi_{\boldsymbol{\theta}_t}, \boldsymbol{\lambda}_t)}{\partial\boldsymbol{\theta}_t[s, a]}$, we obtain

$$\pi_{t+1}(a_\star(s)|s) = \frac{\exp\left\{\boldsymbol{\theta}_{t+1}[s, a_\star(s)]\right\}}{\sum_{a\in\mathcal{A}}\exp\left\{\boldsymbol{\theta}_{t+1}[s, a]\right\}} = \frac{\exp\left\{\boldsymbol{\theta}_t[s, a_\star(s)] + \eta\dfrac{\partial\mathcal{L}(\pi_{\boldsymbol{\theta}_t}, \boldsymbol{\lambda}_t)}{\partial\boldsymbol{\theta}_t[s, a_\star(s)]}\right\}}{\sum_{a\in\mathcal{A}}\exp\left\{\boldsymbol{\theta}_t(s, a) + \eta\dfrac{\partial\mathcal{L}(\pi_{\boldsymbol{\theta}_t}, \boldsymbol{\lambda}_t)}{\partial\boldsymbol{\theta}_t[s, a]}\right\}} \quad (128)$$

$$\geq \frac{\exp\left\{\boldsymbol{\theta}_t[s, a_\star(s)] + \eta\dfrac{\partial\mathcal{L}(\pi_{\boldsymbol{\theta}_t}, \boldsymbol{\lambda}_t)}{\partial\boldsymbol{\theta}_t[s, a_\star(s)]}\right\}}{\sum_{a\in\mathcal{A}}\exp\left\{\boldsymbol{\theta}_t(s, a) + \eta\dfrac{\partial\mathcal{L}(\pi_{\boldsymbol{\theta}_t}, \boldsymbol{\lambda}_t)}{\partial\boldsymbol{\theta}_t[s, a_\star(s)]}\right\}} \quad (129)$$

$$= \frac{\exp\left\{\boldsymbol{\theta}_t(s, a_\star(s))\right\}}{\sum_{a\in\mathcal{A}}\exp\left\{\boldsymbol{\theta}_t(s, a)\right\}} = \pi_t(a_\star(s)|s). \quad (130)$$

$\square$

**Lemma 14.** $\Theta_c(s) \cap \Theta_2(s) \cap \Theta_3(s) \subset \Theta_1(s) \cap \Theta_2(s) \cap \Theta_3(s)$

*Proof.* Let $\boldsymbol{\theta} \in \Theta_c(s) \cap \Theta_2(s) \cap \Theta_3(s)$, we consider the two following cases:

Case (i): $\pi_{\boldsymbol{\theta}}(a_\star(s)|s) \geq \max_{a \neq a_\star(s)} \pi_{\boldsymbol{\theta}}(a|s)$.

$$\frac{\partial \mathcal{L}(\pi_{\boldsymbol{\theta}_t}, \boldsymbol{\lambda}_t)}{\partial \boldsymbol{\theta}_t[s, a_\star(s)]} = \frac{1}{1-\gamma} d_{\pi_{\boldsymbol{\theta}}}^{\rho_0}(s) \pi_{\boldsymbol{\theta}}(a_\star(s)|s) A_{\pi_{\boldsymbol{\theta}}}(s, a_\star(s), \boldsymbol{\lambda}) \tag{131}$$

$$\overset{(106),(107)}{>} \frac{1}{1-\gamma} d_{\pi_{\boldsymbol{\theta}}}^{\rho_0}(s) \pi_{\boldsymbol{\theta}}(a|s) A_{\pi_{\boldsymbol{\theta}}}(s, a, \boldsymbol{\lambda}) = \frac{\partial \mathcal{L}(\pi_{\boldsymbol{\theta}_t}, \boldsymbol{\lambda}_t)}{\partial \boldsymbol{\theta}_t[s, a]}, \tag{132}$$

where the last equation holds since the same analysis from (111)-(116), we have

$$Q_{\pi_t}(s, a_\star(s), \boldsymbol{\lambda}) - Q_{\pi_t}(s, a, \boldsymbol{\lambda}) \geq \frac{1}{2}\Delta_\star(s).$$

Case (ii): $\pi_{\boldsymbol{\theta}}(a_\star(s)|s) \geq \max_{a < a_\star(s)} \pi_{\boldsymbol{\theta}}(a|s)$, which is impossible, since if this case hold, we obtain

$$\pi_{\boldsymbol{\theta}}(a_\star(s)|s) + \pi_{\boldsymbol{\theta}}(a|s) > \frac{2c(s)}{c(s)+1} > 1.$$

$\square$

**Lemma 15.** *Under Assumption 2, updating $\pi_t$ according to Algorithm 1, we obtain*

$$c_\star =: \inf_{s \in \mathcal{S}, t \geq 1} \{\pi_t(a_\star(s)|s)\} > 0.$$

*Proof.* According to (Agarwal et al., 2021, Lemma E.2-Lemma12), we know $\pi_t(a_\star(s)|s) \to 1$, which implies there exists $\mathtt{T}_1(s) \geq 1$, such that

$$\pi_{\theta_{\mathtt{T}_1(s)}}(a_\star(s)|s) \geq \frac{c(s)}{c(s)+1}.$$

Furthermore, since $Q_{\pi_{\boldsymbol{\theta}_t}}(s, a_\star(s)) \to Q_\star(s, a_\star(s))$, as $t \to \infty$, then there exists $\mathtt{T}_2(s) \geq 1$, s.t

$$Q_{\pi_{\theta_{\mathtt{T}_2(s)}}}(s, a_\star(s)) \geq Q_\star(s, a_\star(s)) - \frac{1}{2}\Delta_\star(s).$$

Finally, since $Q_{\pi_{\boldsymbol{\theta}_t}}(s, a_\star(s)) \to V_\star(s)$, and $V_{\pi_{\boldsymbol{\theta}_t}}(s) \to V_\star(s)$, as $t \to \infty$, then there exists $\mathtt{T}_3(s) \geq 1$, such that $\forall t \geq \mathtt{T}_3(s)$,

$$Q_{\pi_{\boldsymbol{\theta}_t}}(s, a_\star(s)) - V_{\pi_{\boldsymbol{\theta}_t}}(s) \leq \frac{1}{2}\Delta_\star(s).$$

Define $\mathtt{T}_0(s) = \max\{\mathtt{T}_1(s), \mathtt{T}_2(s), \mathtt{T}_3(s)\}$, then we obtain

$$\boldsymbol{\theta}_{\mathtt{T}_0(s)} \in \Theta_c(s) \cap \Theta_2(s) \cap \Theta_3(s), \boldsymbol{\theta}_{\mathtt{T}_0(s)} \in \Theta_1(s) \cap \Theta_2(s) \cap \Theta_3(s).$$

According to Lemma 12-14,, i.e., if $\boldsymbol{\theta}_t \in \Theta_1(s) \cap \Theta_2(s) \cap \Theta_3(s)$, then $\boldsymbol{\theta}_{t+1} \in \Theta_1(s) \cap \Theta_2(s) \cap \Theta_3(s)$, and the policy $\pi_{\boldsymbol{\theta}_t}(a_\star(s)|s)$ is increasing in the space $\Theta_1(s) \cap \Theta_2(s) \cap \Theta_3(s)$, we have

$$\inf_{t \geq 0} \pi_{\boldsymbol{\theta}_t}(a_\star(s)|s) = \min_{1 \leq t \leq \mathtt{T}_0(s)} \pi_{\boldsymbol{\theta}_t}(a_\star(s)|s).$$

$\mathtt{T}_0(s)$ only depends on initialization and $c(s)$, which only depends on the CMDP and state $s$.

Finally, we have

$$\inf_{s \in \mathcal{S}, t \geq 1} \{\pi_t(a_\star(s)|s)\} = \min_{s \in \mathcal{S}} \min_{1 \leq t \leq \mathtt{T}_0(s)} \pi_{\boldsymbol{\theta}_t}(a_\star(s)|s) > 0.$$

$\square$

**Lemma 16.** *For any fixed $T > 0$, let $\boldsymbol{\theta}_0 = \mathbf{0}$, $\boldsymbol{\lambda}_0 = \mathbf{0}$. The sequence $\{\boldsymbol{\lambda}_t, \boldsymbol{\theta}_t\}_{t \geq 0}$ is generated by (71)-(72)/(74), and the softmax policy $\pi_t := \pi_{\boldsymbol{\theta}_t}$ is defined as (75). Let $\chi = \frac{1}{(1-\gamma)c_\star} \left\| \frac{d_{\pi_\star}^{\rho_0}}{\rho_0} \right\|_\infty$, $C_1 = \frac{m}{(1-\gamma)^2}\left(1 + \frac{1}{1-\gamma}\right)$. Then*

$$J(\pi_\star) - \frac{1}{T}\sum_{t=0}^{T-1} J(\pi_t) - \frac{1}{T}\sum_{t=0}^{T-1} \boldsymbol{\lambda}_t^\top (\mathbf{c}(\pi_\star) - \mathbf{c}(\pi_t)) \leq \frac{1}{T}\left(\frac{\chi|\mathcal{S}|\log|\mathcal{A}|}{\eta} + \frac{2\chi}{\rho_{\min}(1-\gamma)^2}\right) + 2\eta\chi C_1. \tag{133}$$

*Proof.* According to Lemma 7, we obtain

$$
J(\pi_\star) - J(\pi_t) = \frac{1}{\eta} \sum_{s \in \mathcal{S}} \frac{d_{\pi_\star}^{\rho_0}(s)}{d_{\pi_t}^{\rho_0}(s)} \left( \sum_{a \in \mathcal{A}} \frac{\pi_\star(a|s)}{\pi_t(a|s)} \log \frac{\pi_{t+1}(a|s)}{\pi_t(a|s)} \right)
$$
$$
+ \frac{1}{\eta} \sum_{s \in \mathcal{S}} \sum_{a \in \mathcal{A}} \frac{d_{\pi_\star}^{\rho_0}(s)\pi_\star(a|s)}{d_{\pi_t}^{\rho_0}(s)\pi_t(a|s)} \log Z_t(s) + \boldsymbol{\lambda}_t^\top (\mathbf{c}(\pi_\star) - \mathbf{c}(\pi_t)), \tag{134}
$$

and summing (134) as $t$ ranges from $0$ to $T-1$, we have

$$
J(\pi_\star) - \frac{1}{T} \sum_{t=0}^{T-1} J(\pi_t) = \frac{1}{\eta T} \sum_{t=0}^{T-1} \sum_{s \in \mathcal{S}} \frac{d_{\pi_\star}^{\rho_0}(s)}{d_{\pi_t}^{\rho_0}(s)} \left( \sum_{a \in \mathcal{A}} \frac{\pi_\star(a|s)}{\pi_t(a|s)} \log \frac{\pi_{t+1}(a|s)}{\pi_t(a|s)} \right)
$$
$$
+ \frac{1}{\eta T} \sum_{t=0}^{T-1} \sum_{s \in \mathcal{S}} \sum_{a \in \mathcal{A}} \frac{d_{\pi_\star}^{\rho_0}(s)\pi_\star(a|s)}{d_{\pi_t}^{\rho_0}(s)\pi_t(a|s)} \log Z_t(s) + \frac{1}{T} \sum_{t=0}^{T-1} \boldsymbol{\lambda}_t^\top (\mathbf{c}(\pi_\star) - \mathbf{c}(\pi_t))
$$
$$
\leq \frac{1}{\eta(1-\gamma)T} \sum_{t=0}^{T-1} \sum_{s \in \mathcal{S}} \frac{d_{\pi_\star}^{\rho_0}(s)}{\rho_0(s)} \left( \sum_{a \in \mathcal{A}} \frac{\pi_\star(a|s)}{\pi_t(a|s)} \log \frac{\pi_{t+1}(a|s)}{\pi_t(a|s)} \right)
$$
$$
+ \frac{1}{\eta(1-\gamma)T} \sum_{t=0}^{T-1} \sum_{s \in \mathcal{S}} \sum_{a \in \mathcal{A}} \frac{d_{\pi_\star}^{\rho_0}(s)\pi_\star(a|s)}{\rho_0(s)\pi_t(a|s)} \log Z_t(s) + \frac{1}{T} \sum_{t=0}^{T-1} \boldsymbol{\lambda}_t^\top (\mathbf{c}(\pi_\star) - \mathbf{c}(\pi_t)) \tag{135}
$$
$$
\leq \frac{1}{\eta(1-\gamma)T} \left\| \frac{d_{\pi_\star}^{\rho_0}}{\rho_0} \right\|_\infty \sum_{t=0}^{T-1} \sum_{s \in \mathcal{S}} \sum_{a \in \mathcal{A}} \frac{\pi_\star(a|s)}{\pi_t(a|s)} \log \frac{\pi_{t+1}(a|s)}{\pi_t(a|s)} \tag{136}
$$
$$
+ \frac{1}{\eta(1-\gamma)T} \left\| \frac{d_{\pi_\star}^{\rho_0}}{\rho_0} \right\|_\infty \sum_{t=0}^{T-1} \sum_{s \in \mathcal{S}} \sum_{a \in \mathcal{A}} \frac{\pi_\star(a|s)}{\pi_t(a|s)} \log Z_t(s) + \frac{1}{T} \sum_{t=0}^{T-1} \boldsymbol{\lambda}_t^\top (\mathbf{c}(\pi_\star) - \mathbf{c}(\pi_t)) \tag{137}
$$
$$
\leq \frac{1}{\eta(1-\gamma)c_\star T} \left\| \frac{d_{\pi_\star}^{\rho_0}}{\rho_0} \right\|_\infty \sum_{t=0}^{T-1} \sum_{s \in \mathcal{S}} \sum_{a \in \mathcal{A}} \pi_\star(a|s) \log \frac{\pi_{t+1}(a|s)}{\pi_t(a|s)} \tag{138}
$$
$$
+ \frac{1}{\eta(1-\gamma)c_\star T} \left\| \frac{d_{\pi_\star}^{\rho_0}}{\rho_0} \right\|_\infty \sum_{t=0}^{T-1} \sum_{s \in \mathcal{S}} \log Z_t(s) + \frac{1}{T} \sum_{t=0}^{T-1} \boldsymbol{\lambda}_t^\top (\mathbf{c}(\pi_\star) - \mathbf{c}(\pi_t)) \tag{139}
$$
$$
= \frac{1}{\eta(1-\gamma)c_\star T} \left\| \frac{d_{\pi_\star}^{\rho_0}}{\rho_0} \right\|_\infty \sum_{t=0}^{T-1} \sum_{s \in \mathcal{S}} \Big( \mathbb{KL}\left[ \pi_\star(\cdot|s) \| \pi_t(\cdot|s) \right] - \mathbb{KL}\left[ \pi_\star(\cdot|s) \| \pi_{t+1}(\cdot|s) \right] \Big) \tag{140}
$$
$$
+ \frac{1}{\eta(1-\gamma)c_\star T} \left\| \frac{d_{\pi_\star}^{\rho_0}}{\rho_0} \right\|_\infty \sum_{t=0}^{T-1} \sum_{s \in \mathcal{S}} \log Z_t(s) + \frac{1}{T} \sum_{t=0}^{T-1} \boldsymbol{\lambda}_t^\top (\mathbf{c}(\pi_\star) - \mathbf{c}(\pi_t))
$$
$$
= \frac{1}{\eta(1-\gamma)c_\star T} \left\| \frac{d_{\pi_\star}^{\rho_0}}{\rho_0} \right\|_\infty \sum_{s \in \mathcal{S}} \Big( \mathbb{KL}\left[ \pi_\star(\cdot|s) \| \pi_0(\cdot|s) \right] - \mathbb{KL}\left[ \pi_\star(\cdot|s) \| \pi_T(\cdot|s) \right] \Big)
$$
$$
+ \frac{1}{\eta(1-\gamma)c_\star T} \left\| \frac{d_{\pi_\star}^{\rho_0}}{\rho_0} \right\|_\infty \sum_{t=0}^{T-1} \sum_{s \in \mathcal{S}} \log Z_t(s) + \frac{1}{T} \sum_{t=0}^{T-1} \boldsymbol{\lambda}_t^\top (\mathbf{c}(\pi_\star) - \mathbf{c}(\pi_t))
$$
$$
\leq \frac{1}{\eta(1-\gamma)c_\star T} \left\| \frac{d_{\pi_\star}^{\rho_0}}{\rho_0} \right\|_\infty \sum_{s \in \mathcal{S}} \mathbb{KL}\left[ \pi_\star(\cdot|s) \| \pi_0(\cdot|s) \right] + \frac{1}{T} \sum_{t=0}^{T-1} \boldsymbol{\lambda}_t^\top (\mathbf{c}(\pi_\star) - \mathbf{c}(\pi_t)) \tag{141}
$$
$$
+ \frac{1}{\eta(1-\gamma)c_\star T} \left\| \frac{d_{\pi_\star}^{\rho_0}}{\rho_0} \right\|_\infty \sum_{t=0}^{T-1} \sum_{s \in \mathcal{S}} \log Z_t(s), \tag{142}
$$

where Eq.(135) holds since: for any Markov stationary policy $\pi$,

$$
\begin{aligned}
d_\pi^{\rho_0}(s) = \mathbb{E}_{s_0 \sim \rho_0(\cdot)}[d_\pi^{s_0}(s)] = \mathbb{E}_{s_0 \sim \rho_0(\cdot)} & \left[(1 - \gamma) \sum_{t=0}^{\infty} \gamma^t \mathbb{P}_\pi(s_t = s | s_0)\right] \\
& \geq \mathbb{E}_{s_0 \sim \rho_0(\cdot)}\left[(1 - \gamma) \mathbb{P}_\pi(s_0 = s | s_0)\right] \\
& = (1 - \gamma)\rho_0(s);
\end{aligned}
\tag{143}
$$

Eq.(137) holds since we use $\left\|\dfrac{d_{\pi_\star}^{\rho_0}}{\rho_0}\right\|_\infty$ to denote the distribution mismatch coefficient, i.e.,

$$
\left\|\frac{d_{\pi_\star}^{\rho_0}}{\rho_0}\right\|_\infty := \max_{s \in \mathcal{S}} \left\{\frac{d_{\pi_\star}^{\rho_0}(s)}{\rho_0(s)}\right\},
$$

Due to the Assumption 2, the term $\dfrac{d_{\pi_\star}^{\rho_0}(s)}{\rho_0(s)}$ is well-defined;

Eq.(138) holds since $\pi_\star$ is a deterministic optimal policy, and we denote it as $\pi_\star(a_\star(s)|s) = 1$, otherwise, i.e., if $a \neq a_\star(s)$, $\pi_\star(a|s) = 0$. Recall Proposition 2, we know $c_\star = \inf_{s \in \mathcal{S}, t \geq 1} \{\pi_t(a_\star(s)|s)\} > 0$

$$
\sum_{a \in \mathcal{A}} \frac{\pi_\star(a|s)}{\pi_t(a|s)} \log \frac{\pi_{t+1}(a|s)}{\pi_t(a|s)} = \frac{\pi_\star(a_\star(s)|s)}{\pi_t(a_\star(s)|s)} \log \frac{\pi_{t+1}(a_\star(s)|s)}{\pi_t(a_\star(s)|s)}
\tag{144}
$$

$$
\leq \frac{\pi_\star(a_\star(s)|s)}{c_\star} \log \frac{\pi_{t+1}(a_\star(s)|s)}{\pi_t(a_\star(s)|s)}
\tag{145}
$$

$$
= \frac{1}{c_\star} \sum_{a \in \mathcal{A}} \pi_\star(a|s) \log \frac{\pi_{t+1}(a|s)}{\pi_t(a|s)};
\tag{146}
$$

Similarly, Eq.(139) hold since

$$
\sum_{a \in \mathcal{A}} \frac{\pi_\star(a|s)}{\pi_t(a|s)} = \frac{\pi_\star(a_\star(s)|s)}{\pi_t(a_\star(s)|s)} \leq \frac{1}{c_\star};
$$

Eq(139) holds since:

$$
\sum_{a \in \mathcal{A}} \pi_\star(a|s) \log \frac{\pi_{t+1}(a|s)}{\pi_t(a|s)} = \mathbb{KL}\left[\pi_\star(\cdot|s)\|\pi_t(\cdot|s)\right] - \mathbb{KL}\left[\pi_\star(\cdot|s)\|\pi_{t+1}(\cdot|s)\right];
$$

Eq.(141) holds since we omit the term $\mathbb{KL}\left[\pi_\star(\cdot|s)\|\pi_T(\cdot|s)\right]$.

Taking Eq.(98) into Eq.(142), we achieve

$$
\begin{aligned}
J(\pi_\star) - \frac{1}{T} \sum_{t=0}^{T-1} J(\pi_t) \leq {}& \frac{1}{\eta(1 - \gamma)c_\star T} \left\|\frac{d_{\pi_\star}^{\rho_0}}{\rho_0}\right\|_\infty \sum_{s \in \mathcal{S}} \mathbb{KL}\left[\pi_\star(\cdot|s)\|\pi_0(\cdot|s)\right] + \frac{1}{T} \sum_{t=0}^{T-1} \boldsymbol{\lambda}_t^\top (\mathbf{c}(\pi_\star) - \mathbf{c}(\pi_t)) \\
& + \frac{1}{\rho_{\min}(1 - \gamma)c_\star^2 T} \left\|\frac{d_{\pi_\star}^{\rho_0}}{\rho_0}\right\|_\infty \sum_{t=0}^{T-1} \left(J(\pi_{t+1}) - J(\pi_t) - \boldsymbol{\lambda}_t^\top (\mathbf{c}(\pi_{t+1}) - \mathbf{c}(\pi_t))\right) \\
= {}& \frac{1}{\eta(1 - \gamma)c_\star T} \left\|\frac{d_{\pi_\star}^{\rho_0}}{\rho_0}\right\|_\infty \sum_{s \in \mathcal{S}} \mathbb{KL}\left[\pi_\star(\cdot|s)\|\pi_0(\cdot|s)\right] + \frac{1}{T} \sum_{t=0}^{T-1} \boldsymbol{\lambda}_t^\top (\mathbf{c}(\pi_\star) - \mathbf{c}(\pi_t)) \\
& + \frac{1}{\rho_{\min}(1 - \gamma)c_\star^2 T} \left\|\frac{d_{\pi_\star}^{\rho_0}}{\rho_0}\right\|_\infty \left(J(\pi_T) - J(\pi_0) + \sum_{t=0}^{T-1} \boldsymbol{\lambda}_t^\top (\mathbf{c}(\pi_t) - \mathbf{c}(\pi_{t+1}))\right).
\end{aligned}
\tag{147}
$$

Now, we need to bound the term $\frac{1}{T}\sum_{t=0}^{T-1}\boldsymbol{\lambda}_t^\top(\mathbf{c}(\pi_t)-\mathbf{c}(\pi_{t+1}))$ in Eq.(147), our proof is adaptive to (Ding et al., 2020).

$$
\begin{aligned}
&\frac{1}{T}\left|\sum_{t=0}^{T-1}\boldsymbol{\lambda}_t^\top\left(\mathbf{c}(\pi_t-\mathbf{c}(\pi_{t+1}))\right)\right| \\
=&\frac{1}{T}\left|\sum_{t=0}^{T-1}\boldsymbol{\lambda}_t^\top\left(\mathbf{c}(\pi_{t+1})-\mathbf{c}(\pi_t)\right)\right| \\
=&\frac{1}{T}\left|\sum_{t=0}^{T-1}\left(\boldsymbol{\lambda}_{t+1}^\top\mathbf{c}(\pi_{t+1})-\boldsymbol{\lambda}_t^\top\mathbf{c}(\pi_t)\right)+\frac{1}{T}\sum_{t=0}^{T-1}\left(\boldsymbol{\lambda}_t^\top-\boldsymbol{\lambda}_{t+1}^\top\right)\mathbf{c}(\pi_{t+1})\right| \\
=&\frac{1}{T}\left|\left(\boldsymbol{\lambda}_T^\top\mathbf{c}(\pi_T)-\cancel{\boldsymbol{\lambda}_0^\top\mathbf{c}(\pi_0)}\right)+\frac{1}{T}\sum_{t=0}^{T-1}\left(\boldsymbol{\lambda}_t^\top-\boldsymbol{\lambda}_{t+1}^\top\right)\mathbf{c}(\pi_{t+1})\right| && (148) \\
\leq&\frac{1}{T}\left\|\boldsymbol{\lambda}_T^\top\right\|_2\left\|\mathbf{c}(\pi_T)\right\|_2+\frac{1}{T}\sum_{t=0}^{T-1}\left\|\boldsymbol{\lambda}_t^\top-\boldsymbol{\lambda}_{t+1}^\top\right\|_2\left\|\mathbf{c}(\pi_{t+1})\right\|_2, && (149)
\end{aligned}
$$

where Eq.(148) holds since the initial value $\boldsymbol{\lambda}_0=\mathbf{0}$, and Eq.(149) due to Cauchy-Schwarz inequality. Recall the update rule (71) with respect to $\boldsymbol{\lambda}$:

$$
\boldsymbol{\lambda}_{t+1}\leftarrow\left\{\boldsymbol{\lambda}_t-\eta\frac{\partial}{\partial\boldsymbol{\lambda}}L(\pi_t,\boldsymbol{\lambda}_t)\right\}_+=\left\{\boldsymbol{\lambda}_t-\eta(\mathbf{b}-\mathbf{c}(\pi_t))\right\}_+,
$$

which implies

$$
\|\boldsymbol{\lambda}_{t+1}-\boldsymbol{\lambda}_t\|_2\leq\eta\|\mathbf{b}-\mathbf{c}(\pi_t)\|_2\leq\eta\left(\|\mathbf{b}\|_2+\|\mathbf{c}(\pi_t)\|_2\right). \tag{150}
$$

Now, we need to bound

$$
\begin{aligned}
\|\mathbf{c}(\pi_t)\|_2&=\sqrt{\sum_{i=1}^m|C_i(\pi_t)|^2}=\sqrt{\left|\sum_{i=1}^m\mathbb{E}_{s_0\sim\rho_0(\cdot)}[V_{\pi_t}^{c_i}(s_0)]\right|^2} \\
&=\sqrt{\sum_{i=1}^m\left|\mathbb{E}_{s_0\sim\rho_0(\cdot),s_t\sim\mathbb{P}_{\pi_t}^{(t)}(\cdot|s_0),a_t\sim\pi_t(\cdot|s_t)}\left[\sum_{t=0}^\infty\gamma^t c_i(s_t,a_t)\Big|s_0\right]\right|^2} \\
&\leq\sqrt{m}\frac{1}{1-\gamma}, && (151)
\end{aligned}
$$

where last equation holds since the cost function $c_i(\cdot)$ is bounded by 1.

Recall $\mathbf{b}=(b_1,b_2,\cdots,b_m)^\top$, let

$$
b_{\max}:=\max\{b_1,b_2,\cdots,b_m\},
$$

then, according to (150), we achieve

$$
\|\boldsymbol{\lambda}_{t+1}-\boldsymbol{\lambda}_t\|_2\leq\eta\|\mathbf{b}-\mathbf{c}(\pi_t)\|_2\leq\eta\sqrt{m}\left(b_{\max}+\frac{1}{1-\gamma}\right). \tag{152}
$$

Furthermore,

$$
\|\boldsymbol{\lambda}_T\|_2=\left\|\sum_{t=0}^{T-1}(\boldsymbol{\lambda}_{t+1}-\boldsymbol{\lambda}_t)+\boldsymbol{\lambda}_0\right\|_2\leq\eta\sqrt{m}\left(b_{\max}+\frac{1}{1-\gamma}\right)T. \tag{153}
$$

Taking Eq.(151), Eq.(152) and Eq.(153) to Eq.(149), we have

$$
\frac{1}{T}\sum_{t=0}^{T-1}\boldsymbol{\lambda}_t^\top(\mathbf{c}(\pi_t)-\mathbf{c}(\pi_{t+1}))\leq2\eta m\frac{1}{1-\gamma}\left(b_{\max}+\frac{1}{1-\gamma}\right). \tag{154}
$$

Finally, recall the result (147), and taking (154) to it, we obtain the following equation,

$$J(\pi_\star) - \frac{1}{T}\sum_{t=0}^{T-1} J(\pi_t) \le \frac{1}{\eta(1-\gamma)c_\star T}\left\|\frac{d_{\pi_\star}^{\rho_0}}{\rho_0}\right\|_\infty \sum_{s\in\mathcal{S}} \mathbb{KL}\left[\pi_\star(\cdot|s)\|\pi_0(\cdot|s)\right] + \frac{1}{T}\sum_{t=0}^{T-1}\boldsymbol{\lambda}_t^\top(\mathbf{c}(\pi_\star)-\mathbf{c}(\pi_t))$$

$$+ \frac{1}{\rho_{\min}(1-\gamma)c_\star{}^2}\left\|\frac{d_{\pi_\star}^{\rho_0}}{\rho_0}\right\|_\infty\left(\frac{1}{T}\left(J(\pi_T)-J(\pi_0)\right)-\frac{1}{T}\sum_{t=0}^{T-1}\boldsymbol{\lambda}_t^\top(\mathbf{c}(\pi_{t+1})-\mathbf{c}(\pi_t))\right)$$

$$= \frac{1}{\eta(1-\gamma)c_\star T}\left\|\frac{d_{\pi_\star}^{\rho_0}}{\rho_0}\right\|_\infty \sum_{s\in\mathcal{S}} \mathbb{KL}\left[\pi_\star(\cdot|s)\|\pi_0(\cdot|s)\right] + \frac{1}{T}\sum_{t=0}^{T-1}\boldsymbol{\lambda}_t^\top(\mathbf{c}(\pi_\star)-\mathbf{c}(\pi_t))$$

$$+ \frac{1}{\rho_{\min}(1-\gamma)c_\star{}^2}\left\|\frac{d_{\pi_\star}^{\rho_0}}{\rho_0}\right\|_\infty\left(\frac{1}{T}\left(J(\pi_T)-J(\pi_0)\right)+\frac{1}{T}\sum_{t=0}^{T-1}\boldsymbol{\lambda}_t^\top(\mathbf{c}(\pi_t)-\mathbf{c}(\pi_{t+1}))\right)$$

$$\le \frac{1}{\eta(1-\gamma)c_\star T}\left\|\frac{d_{\pi_\star}^{\rho_0}}{\rho_0}\right\|_\infty |\mathcal{S}|\log|\mathcal{A}| + \frac{1}{T}\sum_{t=0}^{T-1}\boldsymbol{\lambda}_t^\top(\mathbf{c}(\pi_\star)-\mathbf{c}(\pi_t)) \tag{155}$$

$$+ \frac{1}{\rho_{\min}(1-\gamma)c_\star{}^2}\left\|\frac{d_{\pi_\star}^{\rho_0}}{\rho_0}\right\|_\infty\left(\frac{2}{T}\cdot\frac{1}{1-\gamma}+2\eta m\frac{1}{1-\gamma}\left(b_{\max}+\frac{1}{1-\gamma}\right)\right), \tag{156}$$

where Eq.(155) holds since: according to the initial value $\boldsymbol{\lambda}_0 = 0$, then the initial policy $\pi_0$ is reduced to uniform distribution, and the probability of each action $a$ is $\pi_0(a|s) = \frac{1}{|\mathcal{A}|}$,

$$\mathbb{KL}\left[\pi_\star(\cdot|s)\|\pi_0(\cdot|s)\right] = \sum_{a\in\mathcal{A}}\pi_\star(\cdot|s)\log\frac{\pi_\star(a|s)}{\pi_0(a|s)} = \sum_{a\in\mathcal{A}}\pi_\star(\cdot|s)\log\left(|\mathcal{A}|\pi_\star(a|s)\right) \le \log|\mathcal{A}|,$$

which implies

$$\sum_{s\in\mathcal{S}}\mathbb{KL}\left[\pi_\star(\cdot|s)\|\pi_0(\cdot|s)\right] \le |\mathcal{S}|\log|\mathcal{A}|;$$

Eq.(156) holds since: for any policy $\pi$, the objective $J(\pi)$ satisfies

$$J(\pi) \le \frac{1}{1-\gamma},$$

and the result (154) shows the boundedness of

$$\frac{1}{T}\sum_{t=0}^{T-1}\boldsymbol{\lambda}_t^\top\left(\mathbf{c}(\pi_t)-\mathbf{c}(\pi_{t+1})\right) \le 2\eta m\frac{1}{1-\gamma}\left(b_{\max}+\frac{1}{1-\gamma}\right).$$

Finally, let

$$\chi = \frac{1}{(1-\gamma)c_\star}\left\|\frac{d_{\pi_\star}^{\rho_0}}{\rho_0}\right\|_\infty, \quad C_1 = \frac{m}{(1-\gamma)^2}\left(b_{\max}+\frac{1}{1-\gamma}\right), \tag{157}$$

then we rewrite Eq.(156) as follows

$$J(\pi_\star) - \frac{1}{T}\sum_{t=0}^{T-1} J(\pi_t) - \frac{1}{T}\sum_{t=0}^{T-1}\boldsymbol{\lambda}_t^\top(\mathbf{c}(\pi_\star)-\mathbf{c}(\pi_t)) \le \left(\frac{\chi|\mathcal{S}|\log|\mathcal{A}|}{\eta}+\frac{2\chi}{\rho_{\min}(1-\gamma)^2}\right)\cdot\frac{1}{T}+2\eta\chi C_1, \tag{158}$$

This concludes the proof of (133). $\qquad\square$

### D.2 DETAILS FOR PROOF OF THEOREM 2

**Theorem 2** *Under Assumption 1-2, $\pi_{\boldsymbol{\theta}}$ is the softmax policy defined in (9). The time-step $T$ satisfies*

$$T \ge \max\left\{\frac{F}{(1-\gamma)^2}, \frac{F}{((1-\gamma)^2+2m/\iota^2)^2}\right\},$$

where $F := \frac{D_2^2}{|\mathcal{S}| \log |\mathcal{A}| \rho_{\min}^2 D_1}$, $D_1$ and $D_2$ are positive scalars will be special later. The initial $\boldsymbol{\lambda}_0 = \mathbf{0}$, $\boldsymbol{\theta}_0 = \mathbf{0}$, the parameter sequence $\{\boldsymbol{\lambda}_t, \boldsymbol{\theta}_t\}_{t=0}^{T-1}$ is generated according to Algorithm 1. Let $\eta$, $\beta$ satisfy

$$\eta = \sqrt{\left\|\frac{d_{\pi_\star}^{\rho_0}}{\rho_0}\right\|_\infty \frac{|\mathcal{S}| \log |\mathcal{A}|}{C(1-\gamma)T}}, \beta = \sqrt{\left\|\frac{d_{\pi_\star}^{\rho_0}}{\rho_0}\right\|_\infty \frac{4|\mathcal{S}| \log |\mathcal{A}|}{(1-\gamma)^3 \iota^2 c_\star}},$$

where $C$ is a positive scalar will be special later. Then for all $i \in \{1, 2, \cdots, m\}$, $\pi_t := \pi_{\boldsymbol{\theta}_t}$ satisfies

$$\min_{t<T} \{J(\pi_\star) - J(\pi_t)\} \le 4 \left\|\frac{d_{\pi_\star}^{\rho_0}}{\rho_0}\right\|_\infty \sqrt{\frac{|\mathcal{S}| \log |\mathcal{A}|}{c_\star(1-\gamma)^4 T}}, \tag{159}$$

$$\min_{t<T} \{C_i(\pi_t) - b_i\}_+ \le \frac{4\varrho \left\|\frac{d_{\pi_\star}^{\rho_0}}{\rho_0}\right\|_\infty}{\beta - \|\boldsymbol{\lambda}_\star\|_\infty} \sqrt{\frac{|\mathcal{S}| \log |\mathcal{A}|}{c_\star(1-\gamma)^4 T}}. \tag{160}$$

In this section, we show the details for the proof of Theorem 2. The proof contains two key steps: bounding the optimality gap and bounding the constraint violation. Finally, we summary the hyper-parameter setting for us to obtain the results presented in Theorem 2.

**Bounding the Optimality Gap**.

Recall the dual update (71)-(72), we have

$$\|\boldsymbol{\lambda}_T\|_2^2 = \sum_{t=0}^{T-1} \left(\|\boldsymbol{\lambda}_{t+1}\|_2^2 - \|\boldsymbol{\lambda}_t\|_2^2\right)$$

$$\stackrel{(71)}{=} \sum_{t=0}^{T-1} \left(\left\|\left\{\boldsymbol{\lambda}_t - \eta \frac{\partial}{\partial \boldsymbol{\lambda}} L(\pi_t, \boldsymbol{\lambda}_t)\right\}_+\right\|_2^2 - \|\boldsymbol{\lambda}_t\|_2^2\right)$$

$$\stackrel{(70)}{=} \sum_{t=0}^{T-1} \left(\left\|\{\boldsymbol{\lambda}_t - \eta (\mathbf{b} - \mathbf{c}(\pi_t))\}_+\right\|_2^2 - \|\boldsymbol{\lambda}_t\|_2^2\right)$$

$$\le \sum_{t=0}^{T-1} \left(\left\|\boldsymbol{\lambda}_t - \eta (\mathbf{b} - \mathbf{c}(\pi_t))\right\|_2^2 - \|\boldsymbol{\lambda}_t\|_2^2\right)$$

$$= \sum_{t=0}^{T-1} \left(\eta^2 \|\mathbf{b} - \mathbf{c}(\pi_t)\|_2^2 + 2\eta \boldsymbol{\lambda}_t^\top (\mathbf{c}(\pi_t) - \mathbf{b})\right)$$

$$\le \sum_{t=0}^{T-1} \left(\eta^2 m \left(b_{\max} + \frac{1}{1-\gamma}\right)^2 + 2\eta \boldsymbol{\lambda}_t^\top (\mathbf{c}(\pi_t) - \mathbf{c}(\pi_\star))\right) \tag{161}$$

$$= \eta^2 m \left(b_{\max} + \frac{1}{1-\gamma}\right)^2 T + 2\eta \sum_{t=0}^{T-1} \left(\boldsymbol{\lambda}_t^\top (\mathbf{c}(\pi_t) - \mathbf{c}(\pi_\star))\right), \tag{162}$$

where Eq.(161) holds since: $\mathbf{c}(\pi_\star) \preceq \mathbf{b}$, then $\boldsymbol{\lambda}_t^\top (\mathbf{c}(\pi_\star) - \mathbf{b}) \le 0$, and

$$\boldsymbol{\lambda}_t^\top (\mathbf{c}(\pi_t) - \mathbf{b}) = \boldsymbol{\lambda}_t^\top (\mathbf{c}(\pi_t) - \mathbf{c}(\pi_\star) + \mathbf{c}(\pi_\star) - \mathbf{b}) \le \boldsymbol{\lambda}_t^\top (\mathbf{c}(\pi_t) - \mathbf{c}(\pi_\star)); \tag{163}$$

Eq.(152) implies

$$\|\mathbf{b} - \mathbf{c}(\pi_t)\|_2 \le \sqrt{m} \left(b_{\max} + \frac{1}{1-\gamma}\right). \tag{164}$$

Combining the results (163) and (164), we obtain Eq.(161).

Since $\|\boldsymbol{\lambda}_T\|_2^2 \ge 0$, Eq.(162) implies the following boundedness w.r.t. $\boldsymbol{\lambda}_t^\top (\mathbf{c}(\pi_\star)) - \mathbf{c}(\pi_t)$:

$$\frac{1}{2}\eta m \left(b_{\max} + \frac{1}{1-\gamma}\right)^2 \ge \frac{1}{T} \sum_{t=0}^{T-1} \left(\boldsymbol{\lambda}_t^\top (\mathbf{c}(\pi_\star)) - \mathbf{c}(\pi_t)\right). \tag{165}$$

Recall Lemma 16, we have

$$
J(\pi_\star) - \frac{1}{T}\sum_{t=0}^{T-1} J(\pi_t) \leq \frac{1}{T}\left(\frac{\chi|\mathcal{S}|\log|\mathcal{A}|}{\eta} + \frac{2\chi}{\rho_{\min}(1-\gamma)^2}\right) + 2\chi\eta C_1 + \frac{1}{T}\sum_{t=0}^{T-1} \boldsymbol{\lambda}_t^\top (\mathbf{c}(\pi_\star) - \mathbf{c}(\pi_t))
$$

$$
\overset{(165)}{\leq} \frac{1}{T}\left(\frac{\chi|\mathcal{S}|\log|\mathcal{A}|}{\eta} + \frac{2\chi}{\rho_{\min}(1-\gamma)^2}\right) + 2\chi\eta C_1 + \frac{1}{2}\eta m\left(b_{\max} + \frac{1}{1-\gamma}\right)^2
$$

$$
= \frac{1}{T}\left(\frac{\chi|\mathcal{S}|\log|\mathcal{A}|}{\eta} + \frac{2\chi}{\rho_{\min}(1-\gamma)^2}\right) + \eta\left(2\chi C_1 + \frac{1}{2}m\left(b_{\max} + \frac{1}{1-\gamma}\right)^2\right),
$$
$$\tag{166}$$

which implies

$$
\min_{t<T}\{J(\pi_\star) - J(\pi_t)\} \leq \frac{1}{T}\left(\frac{\chi|\mathcal{S}|\log|\mathcal{A}|}{\eta} + \frac{2\chi}{\rho_{\min}(1-\gamma)^2}\right) + \eta\left(2\chi C_1 + \frac{1}{2}m\left(b_{\max} + \frac{1}{1-\gamma}\right)^2\right).
$$
$$\tag{167}$$

Furthermore, let

$$
\eta = \sqrt{\frac{1}{T}\frac{\chi|\mathcal{S}|\log|\mathcal{A}|}{2\chi C_1 + \frac{1}{2}m\left(b_{\max} + \frac{1}{1-\gamma}\right)^2}} = \sqrt{\left\|\frac{d_{\pi_\star}^{\rho_0}}{\rho_0}\right\|_\infty \frac{|\mathcal{S}|\log|\mathcal{A}|}{(1-\gamma)C}\frac{1}{T}}, \tag{168}
$$

where the positive scalar $C$ is defined as follows,

$$
C = c_\star\left(2\chi C_1 + \frac{1}{2}m\left(b_{\max} + \frac{1}{1-\gamma}\right)^2\right) < +\infty. \tag{169}
$$

Then we achieve the optimal gap as follows

$$
\min_{t<T}\{J(\pi_\star) - J(\pi_t)\} \leq \sqrt{\frac{\chi|\mathcal{S}|\log|\mathcal{A}|}{T}\left(2\chi C_1 + \frac{1}{2}m\left(b_{\max} + \frac{1}{1-\gamma}\right)^2\right)} + \frac{2\chi}{\rho_{\min}(1-\gamma)^2 T}
$$
$$\tag{170}$$

$$
= \sqrt{\frac{\chi|\mathcal{S}|\log|\mathcal{A}|}{T}(M_1 + 1)2\chi C_1} + \frac{2\chi}{\rho_{\min}(1-\gamma)^2 T}, \tag{171}
$$

where the constant $M_1$ is special as follows: $2\chi C_1 M_1 = \frac{1}{2}m\left(b_{\max} + \frac{1}{1-\gamma}\right)^2$, i.e.,

$$
M_1 = (1-\gamma)c_\star{}^3\left\|\frac{d_{\pi_\star}^{\rho_0}}{\rho_0}\right\|_\infty^{-1}\left(\frac{b_{\max} + \frac{1}{1-\gamma}}{1}\right).
$$

Recall $\chi$ and $C_1$ defined in Eq.(157),

$$
\chi = \frac{1}{(1-\gamma)c_\star}\left\|\frac{d_{\pi_\star}^{\rho_0}}{\rho_0}\right\|_\infty, \quad C_1 = \frac{m}{(1-\gamma)^2}\left(b_{\max} + \frac{1}{1-\gamma}\right),
$$

taking them into Eq.(171), we rewrite Eq.(171) as follows

$$
\min_{t<T}\{J(\pi_\star) - J(\pi_t)\} \leq \frac{1}{(1-\gamma)^2}\left\|\frac{d_{\pi_\star}^{\rho_0}}{\rho_0}\right\|_\infty \sqrt{\frac{|\mathcal{S}|\log|\mathcal{A}|}{T}\cdot\frac{(M_1+1)2m}{(c_\star)^2}\cdot\left(b_{\max} + \frac{1}{1-\gamma}\right)}
$$

$$
+ \frac{2}{c_\star}\frac{1}{\rho_{\min}(1-\gamma)^3 T}\left\|\frac{d_{\pi_\star}^{\rho_0}}{\rho_0}\right\|_\infty
$$

$$
= \frac{1}{(1-\gamma)^2}\left\|\frac{d_{\pi_\star}^{\rho_0}}{\rho_0}\right\|_\infty \sqrt{\frac{|\mathcal{S}|\log|\mathcal{A}|D_1}{T}} + \frac{D_2}{\rho_{\min}(1-\gamma)^3 T}\left\|\frac{d_{\pi_\star}^{\rho_0}}{\rho_0}\right\|_\infty, \quad (172)
$$

where $D_1$ and $D_2$ are two positive constants defined as follows

$$D_1 := \frac{(M_1 + 1)2m}{(c_\star)^2} \left( b_{\max} + \frac{1}{1 - \gamma} \right) < +\infty, \ \ D_2 := \frac{2}{c_\star} < +\infty. \tag{173}$$

Finally, let [1]

$$T \geq \frac{D_2^2}{(1 - \gamma)^2 |\mathcal{S}| \log |\mathcal{A}| \rho_{\min}^2 D_1}, \tag{174}$$

which implies

$$\min_{t < T} \{J(\pi_\star) - J(\pi_t)\} \leq \frac{2}{(1 - \gamma)^2} \left\| \frac{d_{\pi_\star}^{\rho_0}}{\rho_0} \right\|_\infty \sqrt{\frac{|\mathcal{S}| \log |\mathcal{A}| D_1}{T}}. \tag{175}$$

**Bounding the Constraint Violation**.

We consider the parameter $\|\tilde{\boldsymbol{\lambda}}\|_2 \in [0, \beta]$, where the positive scalar $\beta$ will be special later. According to the update rule (71) w.r.t. parameter $\boldsymbol{\lambda}$, we obtain the following the equation,

$$\begin{aligned}
\left\| \boldsymbol{\lambda}_{t+1} - \tilde{\boldsymbol{\lambda}} \right\|_2^2 &= \left\| \{\boldsymbol{\lambda}_t - \eta(\mathbf{b} - \mathbf{c}(\pi_t))\}_+ - \tilde{\boldsymbol{\lambda}} \right\|_2^2 \\
&\leq \left\| \boldsymbol{\lambda}_t - \eta(\mathbf{b} - \mathbf{c}(\pi_t)) - \tilde{\boldsymbol{\lambda}} \right\|_2^2 \\
&= \left\| \boldsymbol{\lambda}_t - \tilde{\boldsymbol{\lambda}} \right\|_2^2 - 2\eta(\boldsymbol{\lambda}_t - \tilde{\boldsymbol{\lambda}})^\top (\mathbf{b} - \mathbf{c}(\pi_t)) + \eta^2 \|\mathbf{b} - \mathbf{c}(\pi_t)\|_2^2 \\
&\overset{(152)}{\leq} \left\| \boldsymbol{\lambda}_t - \tilde{\boldsymbol{\lambda}} \right\|_2^2 - 2\eta(\boldsymbol{\lambda}_t - \tilde{\boldsymbol{\lambda}})^\top (\mathbf{b} - \mathbf{c}(\pi_t)) + \eta^2 m \left( b_{\max} + \frac{1}{1 - \gamma} \right)^2,
\end{aligned}$$

which is equal to

$$\left\| \boldsymbol{\lambda}_{t+1} - \tilde{\boldsymbol{\lambda}} \right\|_2^2 - \left\| \boldsymbol{\lambda}_t - \tilde{\boldsymbol{\lambda}} \right\|_2^2 \leq -2\eta(\boldsymbol{\lambda}_t - \tilde{\boldsymbol{\lambda}})^\top (\mathbf{b} - \mathbf{c}(\pi_t)) + \eta^2 m \left( b_{\max} + \frac{1}{1 - \gamma} \right)^2. \tag{176}$$

Summing Eq.(176) from $t = 0$ to $T - 1$, we achieve the following equation

$$0 \leq \left\| \boldsymbol{\lambda}_T - \tilde{\boldsymbol{\lambda}} \right\|_2^2 \leq \left\| \boldsymbol{\lambda}_0 - \tilde{\boldsymbol{\lambda}} \right\|_2^2 - 2\eta \sum_{t=0}^{T-1} (\boldsymbol{\lambda}_t - \tilde{\boldsymbol{\lambda}})^\top (\mathbf{b} - \mathbf{c}(\pi_t)) + T\eta^2 m \left( b_{\max} + \frac{1}{1 - \gamma} \right)^2,$$

which implies

$$\frac{1}{T} \sum_{t=0}^{T-1} (\boldsymbol{\lambda}_t - \tilde{\boldsymbol{\lambda}})^\top (\mathbf{b} - \mathbf{c}(\pi_t)) \leq \frac{1}{2\eta T} \left\| \boldsymbol{\lambda}_0 - \tilde{\boldsymbol{\lambda}} \right\|_2^2 + \frac{\eta}{2} m \left( b_{\max} + \frac{1}{1 - \gamma} \right)^2. \tag{177}$$

Due to $\mathbf{c}(\pi_\star) \preceq \mathbf{b}$, and $\boldsymbol{\lambda}_t \succeq \mathbf{0}$, then the following equation holds,

$$-\frac{1}{T} \sum_{t=0}^{T-1} \boldsymbol{\lambda}_t^\top (\mathbf{c}(\pi_\star) - \mathbf{c}(\pi_t)) = -\frac{1}{T} \sum_{t=0}^{T-1} \boldsymbol{\lambda}_t^\top (\mathbf{c}(\pi_\star) - \mathbf{b} + \mathbf{b} - \mathbf{c}(\pi_t)) \geq -\frac{1}{T} \sum_{t=0}^{T-1} \boldsymbol{\lambda}_t^\top (\mathbf{b} - \mathbf{c}(\pi_t)). \tag{178}$$

Recall Lemma 16,

$$J(\pi_\star) - \frac{1}{T} \sum_{t=0}^{T-1} J(\pi_t) - \frac{1}{T} \sum_{t=0}^{T-1} \boldsymbol{\lambda}_t^\top (\mathbf{c}(\pi_\star) - \mathbf{c}(\pi_t)) \leq \frac{1}{T} \left( \frac{\chi |\mathcal{S}| \log |\mathcal{A}|}{\eta} + \frac{2\chi}{\rho_{\min}(1 - \gamma)^2} \right) + 2\eta\chi C_1,$$

---

[1] where we obtain the term $T$ (174) by solving the inequality:

$$\frac{1}{(1 - \gamma)^2} \left\| \frac{d_{\pi_\star}^{\rho_0}}{\rho_0} \right\|_\infty \sqrt{\frac{|\mathcal{S}| \log |\mathcal{A}| D_1}{T}} \geq \frac{D_2}{\rho_{\min}(1 - \gamma)^3 T} \left\| \frac{d_{\pi_\star}^{\rho_0}}{\rho_0} \right\|_\infty.$$

and taking Eq.(178) into above equation, we achieve

$$J(\pi_\star) - \frac{1}{T}\sum_{t=0}^{T-1}J(\pi_t) - \frac{1}{T}\sum_{t=0}^{T-1}\boldsymbol{\lambda}_t^\top(\mathbf{b}-\mathbf{c}(\pi_t)) \leq \frac{1}{T}\left(\frac{\chi|\mathcal{S}|\log|\mathcal{A}|}{\eta} + \frac{2\chi}{\rho_{\min}(1-\gamma)^2}\right) + 2\eta\chi C_1.$$
(179)

We rewrite Eq.(179) as follows,

$$J(\pi_\star) - \frac{1}{T}\sum_{t=0}^{T-1}J(\pi_t) - \frac{1}{T}\sum_{t=0}^{T-1}\boldsymbol{\lambda}_t^\top(\mathbf{b}-\mathbf{c}(\pi_t))$$

$$=J(\pi_\star) - \frac{1}{T}\sum_{t=0}^{T-1}J(\pi_t) - \frac{1}{T}\sum_{t=0}^{T-1}(\boldsymbol{\lambda}_t-\tilde{\boldsymbol{\lambda}})^\top(\mathbf{b}-\mathbf{c}(\pi_t)) - \frac{1}{T}\sum_{t=0}^{T-1}\tilde{\boldsymbol{\lambda}}^\top(\mathbf{b}-\mathbf{c}(\pi_t))$$

$$\leq \frac{1}{T}\left(\frac{\chi|\mathcal{S}|\log|\mathcal{A}|}{\eta} + \frac{2\chi}{\rho_{\min}(1-\gamma)^2}\right) + 2\eta\chi C_1.$$
(180)

Let $\tilde{\boldsymbol{\lambda}} = (\tilde{\lambda}_1, \tilde{\lambda}_2, \cdots, \tilde{\lambda}_m)^\top$, taking the result (177) into Eq.(180), we have

$$J(\pi_\star) - \frac{1}{T}\sum_{t=0}^{T-1}J(\pi_t) - \frac{1}{T}\sum_{t=0}^{T-1}\tilde{\boldsymbol{\lambda}}^\top(\mathbf{b}-\mathbf{c}(\pi_t)) = J(\pi_\star) - \frac{1}{T}\sum_{t=0}^{T-1}J(\pi_t) + \frac{1}{T}\sum_{t=0}^{T-1}\tilde{\boldsymbol{\lambda}}^\top(\mathbf{c}(\pi_t)-\mathbf{b})$$
(181)

$$=J(\pi_\star) - \frac{1}{T}\sum_{t=0}^{T-1}J(\pi_t) + \sum_{i=1}^{m}\tilde{\lambda}_i\left(\frac{1}{T}\sum_{t=0}^{T-1}(C_i(\pi_t)-b_i)\right)$$
(182)

$$\leq \frac{1}{T}\left(\frac{\chi|\mathcal{S}|\log|\mathcal{A}|}{\eta} + \frac{2\chi}{\rho_{\min}(1-\gamma)^2}\right) + 2\eta\chi C_1 + \frac{1}{2\eta T}\|\boldsymbol{\lambda}_0 - \boldsymbol{\lambda}\|_2^2 + \frac{\eta}{2}m\left(b_{\max} + \frac{1}{1-\gamma}\right)^2.$$
(183)

For any policy $\pi$, the objective function $J(\pi)$ is a linear function in an occupancy measure induced by such policy $\pi$. Since the set of occupancy measures is convex and compact, the average of occupancy measures is another occupancy measure that yields a policy, which implies there exists a policy $\tilde{\pi}_t$ such that

$$\frac{1}{T}\sum_{t=0}^{T-1}J(\pi_t) = J(\tilde{\pi}_t), \quad \frac{1}{T}\sum_{t=0}^{T-1}(C_i(\pi_t)-b_i) = C_i(\tilde{\pi}_t)-b_i.$$
(184)

Furthermore, let

$$\tilde{\lambda}_i = \begin{cases} \beta = \sqrt{\chi|\mathcal{S}|\log|\mathcal{A}|}\dfrac{2}{(1-\gamma)\iota}, & \text{if } \sum_{t=0}^{T-1}(C_i(\pi_t)-b_i) \geq 0, \\ \\ 0, & \text{if } \sum_{t=0}^{T-1}(C_i(\pi_t)-b_i) < 0, \end{cases}$$
(185)

recall $\eta$ defined in (168), i.e., $\eta = \sqrt{\dfrac{1}{T}\dfrac{\chi|\mathcal{S}|\log|\mathcal{A}|}{2\chi C_1 + \frac{1}{2}m\left(b_{\max} + \frac{1}{1-\gamma}\right)^2}}$, then we rewrite Eq.(182) as follows,

$$J(\pi_\star) - \frac{1}{T}\sum_{t=0}^{T-1} J(\pi_t) + \sum_{i=1}^{m} \tilde{\lambda}_i \left(\frac{1}{T}\sum_{t=0}^{T-1}(C_i(\pi_t) - b_i)\right)$$

$$\overset{(184)}{=} J(\pi_\star) - J(\tilde{\pi}_t) + \sum_{i=1}^{m} \tilde{\lambda}_i \left(C_i(\tilde{\pi}_t) - b_i\right)$$

$$\overset{(185)}{=} J(\pi_\star) - J(\tilde{\pi}_t) + \beta \sum_{i=1}^{m} \{C_i(\tilde{\pi}_t) - b_i\}_+ \tag{186}$$

$$= J(\pi_\star) - J(\tilde{\pi}_t) + \beta \mathbf{1}_m^\top \{\mathbf{c}(\tilde{\pi}_t) - \mathbf{b}\}_+ \qquad \blacktriangleright \text{ the vector version of Eq.(186)}$$

$$\overset{(183)}{\leq} \frac{1}{T}\left(\frac{\chi|\mathcal{S}|\log|\mathcal{A}|}{\eta} + \frac{2\chi}{\rho_{\min}(1-\gamma)^2}\right) + 2\eta\chi C_1 + \frac{1}{2\eta T}\left\|\boldsymbol{\lambda}_0 - \tilde{\boldsymbol{\lambda}}\right\|_2^2 + \frac{\eta}{2}m\left(b_{\max} + \frac{1}{1-\gamma}\right)^2 \tag{187}$$

$$\overset{(185)}{\leq} \frac{1}{T}\left(\frac{\chi|\mathcal{S}|\log|\mathcal{A}|}{\eta} + \frac{2\chi}{\rho_{\min}(1-\gamma)^2}\right) + 2\eta\chi C_1 + \frac{m}{2\eta T}\beta^2 + \frac{\eta}{2}m\left(b_{\max} + \frac{1}{1-\gamma}\right)^2 \tag{188}$$

$$\overset{(168)}{=} \sqrt{\frac{\chi|\mathcal{S}|\log|\mathcal{A}|}{T}\left(2\chi C_1 + \frac{1}{2}m\left(b_{\max} + \frac{1}{1-\gamma}\right)^2\right)} + \frac{2\chi}{\rho_{\min}(1-\gamma)^2 T} + \frac{m}{2\eta T}\beta^2 \tag{189}$$

$$\overset{(168)}{=} \left(1 + \frac{2m}{(1-\gamma)^2\iota^2}\right)\sqrt{\frac{\chi|\mathcal{S}|\log|\mathcal{A}|}{T}\left(2\chi C_1 + \frac{1}{2}m\left(b_{\max} + \frac{1}{1-\gamma}\right)^2\right)} + \frac{2\chi}{\rho_{\min}(1-\gamma)^2 T} \tag{190}$$

$$\overset{(170),(172)}{=} \left(\frac{1}{(1-\gamma)^2} + \frac{2m}{(1-\gamma)^4\iota^2}\right)\left\|\frac{d_{\pi_\star}^{\rho_0}}{\rho_0}\right\|_\infty \sqrt{\frac{|\mathcal{S}|\log|\mathcal{A}|D_1}{T}} + \frac{D_2}{\rho_{\min}(1-\gamma)^3 T}\left\|\frac{d_{\pi_\star}^{\rho_0}}{\rho_0}\right\|_\infty, \tag{191}$$

where Eq.(188) holds since: by the definition of $\tilde{\boldsymbol{\lambda}}$ in Eq.(185), and initial $\boldsymbol{\lambda}_0 = \mathbf{0}$, we have

$$\left\|\boldsymbol{\lambda}_0 - \tilde{\boldsymbol{\lambda}}\right\|_2^2 = \left\|\tilde{\boldsymbol{\lambda}}\right\|_2^2 \leq m\beta^2;$$

Eq.(190) holds since we replace the term $\dfrac{m}{2\eta T}\beta^2$ as follows: recall $\beta = \sqrt{\chi|\mathcal{S}|\log|\mathcal{A}|}\dfrac{2}{(1-\gamma)\iota}$ defined in (185) we have

$$\frac{m}{2\eta T}\beta^2 = \frac{m}{2}\beta^2 \sqrt{\frac{2\chi C_1 + \frac{1}{2}m\left(b_{\max} + \frac{1}{1-\gamma}\right)^2}{T\chi|\mathcal{S}|\log|\mathcal{A}|}}$$

$$= \frac{2m}{(1-\gamma)^2\iota^2}\sqrt{\frac{\chi|\mathcal{S}|\log|\mathcal{A}|}{T}\left(2\chi C_1 + \frac{1}{2}m\left(b_{\max} + \frac{1}{1-\gamma}\right)^2\right)}.$$

Finally, let

$$\delta := \left(\frac{1}{(1-\gamma)^2} + \frac{2m}{(1-\gamma)^4\iota^2}\right)\left\|\frac{d_{\pi_\star}^{\rho_0}}{\rho_0}\right\|_\infty \sqrt{\frac{|\mathcal{S}|\log|\mathcal{A}|D_1}{T}} + \frac{D_2}{\rho_{\min}(1-\gamma)^3 T}\left\|\frac{d_{\pi_\star}^{\rho_0}}{\rho_0}\right\|_\infty,$$

then the results (191) can be represented simply as follows,

$$J(\pi_\star) - J(\tilde{\pi}_t) + \beta \mathbf{1}_m^\top \{\mathbf{c}(\tilde{\pi}_t) - \mathbf{b}\}_+ \leq \delta; \tag{192}$$

recall Lemma 3 that reveals the boundedness of $\boldsymbol{\lambda}_\star$, the definition of $\beta = \sqrt{\chi|\mathcal{S}|\log|\mathcal{A}|}\dfrac{2}{(1-\gamma)\iota}$ implies $\beta > \|\boldsymbol{\lambda}_\star\|_\infty$, then applying Lemma 4, we have

$$\mathbf{1}_m^\top\{\mathbf{c}(\tilde{\pi}_t) - \mathbf{b}\}_+ < \frac{\delta}{\beta - \|\boldsymbol{\lambda}_\star\|_\infty}. \tag{193}$$

Since $\mathbf{1}_m^\top\{\mathbf{c}(\tilde{\pi}) - \mathbf{b}\}_+ = \sum_{i=1}^m \{C_i(\tilde{\pi}) - b_i\}_+$ and each $\{C_i(\tilde{\pi}) - b_i\}_+ \geq 0$, then we have

$$\{\mathbf{c}(\tilde{\pi}_t) - \mathbf{b}\}_+ \preceq \frac{\delta}{\beta - \|\boldsymbol{\lambda}_\star\|_\infty}\mathbf{1}_m. \tag{194}$$

Eq.(194) implies for each $i \in \{1, 2, \cdots, m\}$: $\{C_i(\tilde{\pi}_t) - b_i\}_+ \leq \dfrac{\delta}{\beta - \|\boldsymbol{\lambda}_\star\|_\infty}$, i.e.,

$$\{C_i(\tilde{\pi}_t) - b_i\}_+ \stackrel{(184)}{=} \left\{\frac{1}{T}\sum_{t=0}^{T-1}(C_i(\pi_t) - b_i)\right\}_+ \leq \frac{\delta}{\beta - \|\boldsymbol{\lambda}_\star\|_\infty}, \tag{195}$$

which implies the *Best-Case Constraint Violation* as follows: for each $i \in \{1, 2, \cdots, m\}$, we have

$$\min_{t<T}\{C_i(\pi_t) - b_i\}_+ \tag{196}$$

$$\leq \frac{1}{\beta - \|\boldsymbol{\lambda}_\star\|_\infty}\left(\left(\frac{1}{(1-\gamma)^2} + \frac{2m}{(1-\gamma)^4\iota^2}\right)\left\|\frac{d_{\pi_\star}^{\rho_0}}{\rho_0}\right\|_\infty\sqrt{\frac{|\mathcal{S}|\log|\mathcal{A}|D_1}{T}} + \frac{D_2}{\rho_{\min}(1-\gamma)^3T}\left\|\frac{d_{\pi_\star}^{\rho_0}}{\rho_0}\right\|_\infty\right).$$

Furthermore, let [2]

$$T \geq \frac{D_2^2}{\left((1-\gamma)^2 + 2m/\iota^2\right)^2\rho_{\min}^2 D_1|\mathcal{S}|\log|\mathcal{A}|}, \tag{197}$$

then we obtain

$$\min_{t<T}\{C_i(\pi_t) - b_i\}_+ \leq \frac{2}{\beta - \|\boldsymbol{\lambda}_\star\|_\infty}\left(1 + \frac{2m}{(1-\gamma)^2\iota^2}\right)\left\|\frac{d_{\pi_\star}^{\rho_0}}{\rho_0}\right\|_\infty\sqrt{\frac{|\mathcal{S}|\log|\mathcal{A}|D_1}{(1-\gamma)^4T}}$$

$$= \frac{2\left(1 + \dfrac{2m}{(1-\gamma)^2\iota^2}\right)}{\beta - \|\boldsymbol{\lambda}_\star\|_\infty}\left\|\frac{d_{\pi_\star}^{\rho_0}}{\rho_0}\right\|_\infty\sqrt{\frac{|\mathcal{S}|\log|\mathcal{A}|D_1}{(1-\gamma)^4T}}. \tag{198}$$

**Summarizing the Conclusion under Special Hyper-Parameter Setting**.

Finally, recall the condition for the term $T$ in (174), (197), we conclude if the time-step $T$ satisfies

$$T \geq \max\left\{\frac{1}{(1-\gamma)^2}, \frac{1}{((1-\gamma)^2 + 2m/\iota^2)^2}\right\}\cdot\frac{D_2^2}{|\mathcal{S}|\log|\mathcal{A}|\rho_{\min}^2 D_1},$$

the step-size $\eta$ defined in (168) satisfies

$$\eta = \sqrt{\left\|\frac{d_{\pi_\star}^{\rho_0}}{\rho_0}\right\|_\infty\frac{|\mathcal{S}|\log|\mathcal{A}|}{(1-\gamma)C}\frac{1}{T}},$$

and the constant term $\beta$ satisfies

$$\beta \stackrel{(185)}{=} \sqrt{\chi|\mathcal{S}|\log|\mathcal{A}|}\frac{2}{(1-\gamma)\iota} \tag{199}$$

$$\stackrel{(157)}{=} \sqrt{\frac{1}{(1-\gamma)c_\star}\left\|\frac{d_{\pi_\star}^{\rho_0}}{\rho_0}\right\|_\infty|\mathcal{S}|\log|\mathcal{A}|}\frac{2}{(1-\gamma)\iota} \tag{200}$$

$$:= \sqrt{\left\|\frac{d_{\pi_\star}^{\rho_0}}{\rho_0}\right\|_\infty\frac{D|\mathcal{S}|\log|\mathcal{A}|}{(1-\gamma)^3\iota^2}}, \tag{201}$$

---

[2]where we obtain the term $T$ (197) by solving the inequality:

$$\left(\frac{1}{(1-\gamma)^2} + \frac{2m}{(1-\gamma)^4\iota^2}\right)\left\|\frac{d_{\pi_\star}^{\rho_0}}{\rho_0}\right\|_\infty\sqrt{\frac{|\mathcal{S}|\log|\mathcal{A}|D_1}{T}} \geq \frac{D_2}{\rho_{\min}(1-\gamma)^3T}\left\|\frac{d_{\pi_\star}^{\rho_0}}{\rho_0}\right\|_\infty.$$

where we define the constant $D$ as follows

$$D_1 := \frac{4}{c_\star}.$$

Then, according to (175) and (198), the following holds

$$\min_{t<T} \left\{ J(\pi_\star) - J(\pi_t) \right\} \le 2 \left\| \frac{d^{\rho_0}_{\pi_\star}}{\rho_0} \right\|_\infty \sqrt{\frac{|\mathcal{S}| \log |\mathcal{A}| D_1}{(1-\gamma)^4 T}},$$

$$\min_{t<T} \left\{ C_i(\pi_t) - b_i \right\}_+ \le \frac{2}{\beta - \|\boldsymbol{\lambda}_\star\|_\infty} \left( 1 + \frac{2m}{(1-\gamma)^2 \iota^2} \right) \left\| \frac{d^{\rho_0}_{\pi_\star}}{\rho_0} \right\|_\infty \sqrt{\frac{|\mathcal{S}| \log |\mathcal{A}| D_1}{(1-\gamma)^4 T}},$$

where each $i \in \{1, 2, \cdots, m\}$. This concludes the proof of Theorem 2.

**Remark 3.** *Recall Eq.(143): $d^{\rho_0}_\pi(s) \ge (1-\gamma)\rho_0(s)$, which implies $\left\| \frac{d^{\rho_0}_{\pi_\star}}{\rho_0} \right\|_\infty \ge (1-\gamma)$. Thus,*

$$\beta \ge \sqrt{\frac{1}{c_\star} |\mathcal{S}| \log |\mathcal{A}|} \frac{2}{(1-\gamma)\iota} > \frac{2}{(1-\gamma)\iota}.$$

*Lemma 3 shows that*

$$\|\boldsymbol{\lambda}_\star\|_\infty \le \frac{2}{(1-\gamma)\iota}.$$

*Thus $\beta > \|\boldsymbol{\lambda}_\star\|_\infty$.*

# E    PROOF OF THEOREM 3

## E.1    GEOMETRIC DISTRIBUTION

Before we show the details of the proof, we introduce some basic notations about geometric distribution $\text{Geo}(\gamma)$, which is defined as the following discrete probability distributions: the probability distribution of the number $\tau$ of failures before the first success, supported on the set $\{0, 1, 2, \cdots \}$, i.e.,

$$\mathbb{P}(\tau = t) = (1 - \gamma)^t \gamma, \quad \gamma \in (0, 1), \quad t = 0, 1, 2, 3, \cdots . \tag{202}$$

To understand $\text{Geo}(\gamma)$ (202) clearly, we list the distribution column of the distribution $\text{Geo}(\gamma)$ in the following Table 2.

Table 2: Distribution column of the distribution $\text{Geo}(\gamma)$.

| $\tau$ | 0 | 1 | 2 | 3 | 4 | $\cdots\cdots$ | $t$ | $\cdots\cdots$ |
|---|---|---|---|---|---|---|---|---|
|  | $\gamma$ | $(1-\gamma)\gamma$ | $(1-\gamma)^2\gamma$ | $(1-\gamma)^3\gamma$ | $(1-\gamma)^4\gamma$ | $\cdots\cdots$ | $(1-\gamma)^t\gamma$ | $\cdots\cdots$ |

## E.2    ROLLOUT WITH FINITE HORIZONS

---
**Algorithm 4** $\texttt{EstQ}(\pi, g, s, a)$: Estimate $Q$ Value Function
---
1: **Input**: Policy $\pi$; Reward function or Cost function $g(\cdot, \cdot)$; State-action pair $(s, a)$;
2: **Initialization**: $\widehat{Q}(s, a) = 0$, $(s_0, a_0) = (s, a)$;
3: Draw an integer $\tau$ from a geometric distribution with parameter $(1 - \gamma)$: $\mathbb{P}(\tau = t) = (1 - \gamma)\gamma^t$;
4: **for** $t = 0, 1, 2, \cdots, \tau - 1$ **do**
5:     Collect reward (or cost) $g(s_t, a_t)$ and add to estimate: $\widehat{Q}(s, a) \leftarrow \widehat{Q}(s, a) + g(s_t, a_t)$;
6:     Simulate the next state and next action as follows: $s_{t+1} \sim \mathbb{P}(\cdot | s_t, a_t)$; $a_{t+1} \sim \pi(\cdot | s_{t+1})$;
7: **end for**
8: Collect last reward (or cost) $g(s_\tau, a_\tau)$, add to estimate: $\widehat{Q}(s, a) \leftarrow \widehat{Q}(s, a) + g(s_\tau, a_\tau)$.
9: **Output**: $\widehat{Q}(s, a)$.

---

---
**Algorithm 5** $\texttt{EstV}(\pi, g, s)$: Estimate $V$ Value Function
---
1: **Input**: Policy $\pi$ to be evaluated; Reward function or Cost function $g(\cdot, \cdot)$; State $s$;
2: **Initialization**: $\widehat{V}(s) = 0$, $s_0 = s$, $a_0 \sim \pi(\cdot | s_0)$;
3: Draw an integer $\widetilde{\tau}$ from a geometric distribution with parameter $(1 - \gamma)$: $\mathbb{P}(\tilde{\tau} = t) = (1 - \gamma)\gamma^t$;
4: **for** $t = 0, 1, 2, \cdots, \widetilde{\tau} - 1$ **do**
5:     Collect reward (or cost) $g(s_t, a_t)$ and add to estimate: $\widehat{V}(s) \leftarrow \widehat{V}(s) + g(s_t, a_t)$;
6:     Simulate the next state and next action as follows: $s_{t+1} \sim \mathbb{P}(\cdot | s_t, a_t)$; $a_{t+1} \sim \pi(\cdot | s_{t+1})$;
7: **end for**
8: Collect last reward (or cost) $g(s_{\widetilde{\tau}}, a_{\widetilde{\tau}})$, add to estimate: $\widehat{V}(s) \leftarrow \widehat{V}(s) + g(s_{\widetilde{\tau}}, a_{\widetilde{\tau}})$.
9: **Output**: $\widehat{V}(s)$.

---

## E.3    PROOF OF PROPOSITION 3

We need the following Lemma 17 to show Proposition 3.

**Lemma 17** (Dominated Convergence Theorem). *Let $\{X_n\}_{n \geq 0}$ be a random variable sequence, and $X_n \to X$ almost surely, as $n \to \infty$. Furthermore, if $|X_n| \leq Y$ for all $n$, and $\mathbb{E}[Y] \leq \infty$, then $\mathbb{E}[X_n] \to \mathbb{E}[X]$, as $n \to \infty$.*

*Proof.* See (Durrett, 2019, Theorem 1.6.7). □

**Proposition 3**. *The output of Algorithm 4 (also see Algorithm 4) is an unbiased estimator of $Q_\pi(s,a)$ or $Q_\pi^c(s,a)$, i.e., let*

$$\widehat{Q}_\pi(s,a) = \texttt{EstQ}(\pi, r, s, a), \ \widehat{Q}_\pi^{c_i}(s,a) = \texttt{EstQ}(\pi, c_i, s, a), \tag{203}$$

*then the following holds*

$$\mathbb{E}[\widehat{Q}_\pi(s,a)] = Q_\pi(s,a), \ \mathbb{E}[\widehat{Q}_\pi^{c_i}(s,a)] = Q_\pi^{c_i}(s,a). \tag{204}$$

*Proof.* This proof is adaptive to Paternain (2018); Zhang et al. (2020). Without losing generality, we only need to show the case of $g(\cdot, \cdot) = r(\cdot, \cdot)$, i.e., $\mathbb{E}[\widehat{Q}_\pi(s,a)] = Q_\pi(s,a)$.

According to the iteration from Algorithm 4, we obtain the estimator of $Q_\pi(s,a)$ as follows,

$$\widehat{Q}_\pi(s,a) = \sum_{t=0}^{\tau} r(s_t, a_t), \ (s_0, a_0) = (s,a), \ \tau \sim \text{Geo}(1-\gamma). \tag{205}$$

We consider the expectation of (205):

$$\mathbb{E}[\widehat{Q}_\pi(s,a)] = \mathbb{E}\left[\sum_{t=0}^{\tau} r(s_t, a_t) \Big| \pi, s_0 = s, a_0 = a\right] \tag{206}$$

$$= \mathbb{E}\left[\sum_{t=0}^{\infty} \mathbb{I}\{t \le \tau\} r(s_t, a_t) \Big| \pi, s_0 = s, a_0 = a\right] \tag{207}$$

$$= \sum_{t=0}^{\infty} \mathbb{E}\left[\mathbb{I}\{t \le \tau\} r(s_t, a_t) \Big| \pi, s_0 = s, a_0 = a\right], \tag{208}$$

where Eq.(207) holds since: we have substituted $\infty$ for the $\tau$ via the indicator function $\mathbb{I}\{\cdot\}$ such that the summand for $t > \tau$ is vanished;

Eq.(208) holds since we use the dominated convergence theorem (see Lemma 17): let

$$X_n = \sum_{t=0}^{n} \mathbb{I}\{t \le \tau\} r(s_t, a_t),$$

then we obtain

$$|X_n| = \left|\sum_{t=0}^{n} \mathbb{I}\{t \le \tau\} r(s_t, a_t)\right| \le \sum_{t=0}^{n} \mathbb{I}\{t \le \tau\} := Y_n, \tag{209}$$

and recall $\tau \sim \text{Geo}(1-\gamma)$, then we obtain

$$\mathbb{P}(t \le \tau) = \sum_{\tau=t}^{\infty} \gamma^\tau (1-\gamma) = \gamma^t, \tag{210}$$

$$\mathbb{E}[Y_n] = \mathbb{E}\left[\sum_{t=0}^{n} \mathbb{I}\{t \le \tau\}\right] = \sum_{t=0}^{n} \mathbb{P}(t \le \tau) \stackrel{(210)}{=} \sum_{t=0}^{n} \gamma^t \le \frac{1}{1-\gamma}; \tag{211}$$

according to the results (209) and (211), and applying Lemma 17, we obtain

$$\mathbb{E}\left[\sum_{t=0}^{\infty} \mathbb{I}\{t \le \tau\} r(s_t, a_t) \Big| \pi, s_0 = s, a_0 = a\right] = \sum_{t=0}^{\infty} \mathbb{E}\left[\mathbb{I}\{t \le \tau\} r(s_t, a_t) \Big| \pi, s_0 = s, a_0 = a\right],$$

i.e., we have checked exchange condition for the sum and the expectation in the previous expression from Eq.(207) to Eq.(208).

Furthermore, we consider the result (208) as follows,

$$
\begin{aligned}
\mathbb{E}[\widehat{Q}_\pi(s,a)] &= \sum_{t=0}^{\infty} \mathbb{E}\left[\mathbb{I}\left\{t \le \tau\right\} r(s_t, a_t)\Big|\pi, s_0 = s, a_0 = a\right] \\
&= \sum_{t=0}^{\infty} \mathbb{E}\left[\mathbb{E}_{\tau\sim\mathrm{Geo}(1-\gamma)}\left[\mathbb{I}\left\{t \le \tau\right\} r(s_t, a_t)\right]\Big|\pi, s_0 = s, a_0 = a\right] & (212) \\
&= \sum_{t=0}^{\infty} \mathbb{E}\left[\mathbb{E}_{\tau\sim\mathrm{Geo}(1-\gamma)}\left[\mathbb{I}\left\{t \le \tau\right\}\right] r(s_t, a_t)\Big|\pi, s_0 = s, a_0 = a\right] & (213) \\
&= \sum_{t=0}^{\infty} \mathbb{E}\left[\gamma^t r(s_t, a_t)\Big|\pi, s_0 = s, a_0 = a\right] & (214) \\
&= Q_\pi(s,a), & (215)
\end{aligned}
$$

where Eq(212) holds due to the *double expectation formula*: $\mathbb{E}[X] = \mathbb{E}_Y[\mathbb{E}_X[X|Y]]$, which implies that we find the expected value of $X$ by conditioning it on another random variable $Y$;

Eq.(213) holds since: the horizon $\tau$ is drawn independently of the MDP sequence $\{s_t, a_t, r(s_t, a_t)\}$;

Eq.(214) holds since: $\tau \sim \mathrm{Geo}(1-\gamma)$, then we obtain

$$
\mathbb{E}_{\tau\sim\mathrm{Geo}(1-\gamma)}\left[\mathbb{I}\left\{t \le \tau\right\}\right] = \mathbb{P}(t \le \tau) = \sum_{\tau=t}^{\infty} \gamma^\tau(1-\gamma) = \gamma^t.
$$

This concludes the result $\mathbb{E}[\widehat{Q}_\pi(s,a)] = Q_\pi(s,a)$.

If we replace all the term $r(s_t, a_t)$ to $c_i(s_t, a_t)$ from Eq.(205)-Eq.(215), we obtain $\mathbb{E}[\widehat{Q}_\pi^{c_i}(s,a)] = Q_\pi^{c_i}(s,a)$. $\qquad\square$

### E.4    Proof of Theorem 3

Since return objective $J(\pi_{\boldsymbol{\theta}})$ and cost function $C_i(\pi_{\boldsymbol{\theta}})$ share a similar structure, all the Eq.(22)-(24) can be extended to $C_i(\pi_{\boldsymbol{\theta}})$ if we use $c_i$ to replace $r$ respectively. In this section, we only show the case of the reward function $J(\pi_{\boldsymbol{\theta}})$ in Theorem 3. Before we show the details, we present some insights of those unbiased esitimators.

**Rollout Algorithm**

We rollout a policy evaluation with respect to $\pi_{\boldsymbol{\theta}}$ according to Algorithm 4,

$$
\widehat{Q}_{\pi_{\boldsymbol{\theta}}}(s,a) = \mathtt{EstQ}(\pi_{\boldsymbol{\theta}}, r, s, a), \tag{216}
$$

we use $\tau \sim \mathrm{Geo}(1-\gamma)$ to denote the terminal time of the horizon of the rollout (216).

Furthermore, let $\widehat{G}_{\pi_{\boldsymbol{\theta}}}(s,a)$ be an estimator defined as follows,

$$
\widehat{G}_{\pi_{\boldsymbol{\theta}}}(s,a) = \frac{1}{1-\gamma}\widehat{Q}_{\pi_{\boldsymbol{\theta}}}(s_\tau, a_\tau)\frac{\partial \log \pi_{\boldsymbol{\theta}}(a_\tau|s_\tau)}{\partial \theta_{s,a}}, \tag{217}
$$

where we obtain $\widehat{Q}_{\pi_{\boldsymbol{\theta}}}(s_\tau, a_\tau)$ according to Algorithm 4,

$$
\widehat{Q}_{\pi_{\boldsymbol{\theta}}}(s_\tau, a_\tau) = \mathtt{EstQ}(\pi_{\boldsymbol{\theta}}, r, s_\tau, a_\tau). \tag{218}
$$

Let $\tau' \sim \mathrm{Geo}(1-\gamma)$ be the terminal time of the horizon of the rollout (218), and we denote the rollout trajectory as follows,

$$
\mathcal{D}' = \left\{\left(s_j', a_j', r(s_j', a_j')\right)\right\}_{j=0}^{\tau'}, \quad \text{where initial state-action pair } (s_0', a_0') = (s_\tau, a_\tau).
$$

Then, we rewrite the value $\widehat{Q}_{\pi_{\boldsymbol{\theta}}}(s_\tau, a_\tau)$ (218) as follows,

$$
\widehat{Q}_{\pi_{\boldsymbol{\theta}}}(s_\tau, a_\tau) = \sum_{j=0}^{\tau'} r(s_j', a_j'), \quad \left(s_0', a_0'\right) = (s_\tau, a_\tau), \quad \tau' \sim \mathrm{Geo}(1-\gamma). \tag{219}
$$

---

**Algorithm 6** $\texttt{EstPG}(\pi, g, s, a)$: Estimate Policy Gradient

1: **Input**: A policy $\pi_{\boldsymbol{\theta}}$ with given parameter $\boldsymbol{\theta}$, $(s, a) \in \mathcal{S} \times \mathcal{A}$;
2: **Policy Evaluation Rollout for** $(s, a)$**-Pair**: $\widehat{Q}_{\pi_{\boldsymbol{\theta}}}(s, a) = \texttt{EstQ}(\pi_{\boldsymbol{\theta}}, r, s, a)$, and $\tau \sim \text{Geo}(1 - \gamma)$ denotes the terminal time of the horizon of such a policy evaluation rollout;
3: **Policy Evaluation Rollout for** $(s_\tau, a_\tau)$**-Pair**: $\widehat{Q}_{\pi_{\boldsymbol{\theta}}}(s_\tau, a_\tau) = \texttt{EstQ}(\pi_{\boldsymbol{\theta}}, r, s_\tau, a_\tau)$, and $\tau^{'} \sim \text{Geo}(1 - \gamma)$ denotes the terminal time of the horizon of such a policy evaluation rollout;
4: Collect the trajectory $\mathcal{D}^{'} = \left\{ (s_j^{'}, a_j^{'}, r(s_j^{'}, a_j^{'})) \right\}_{j=0:\tau^{'}}$, where initial state-action pair $(s_0^{'}, a_0^{'}) = (s_\tau, a_\tau)$;
5: **Output**: $\widehat{G}_{\pi_{\boldsymbol{\theta}}}(s, a)$ defined as follows,

$$\widehat{G}_{\pi_{\boldsymbol{\theta}}}(s, a) = \frac{1}{1 - \gamma} \widehat{Q}_{\pi_{\boldsymbol{\theta}}}(s_\tau, a_\tau) \frac{\partial \log \pi_{\boldsymbol{\theta}}(a_\tau | s_\tau)}{\partial \theta_{s,a}} = \frac{1}{1 - \gamma} \sum_{j=0}^{\tau^{'}} r(s_j^{'}, a_j^{'}) \frac{\partial}{\partial \theta_{s,a}} \log \pi_{\boldsymbol{\theta}}(a_\tau | s_\tau). \tag{221}$$

---

Taking Eq.(219) to (217), we obtain the expression of $\widehat{G}_{\pi_{\boldsymbol{\theta}}}(s, a)$ (217) as follows,

$$\widehat{G}_{\pi_{\boldsymbol{\theta}}}(s, a) = \frac{1}{1 - \gamma} \sum_{j=0}^{\tau^{'}} r(s_j^{'}, a_j^{'}) \frac{\partial}{\partial \theta_{s,a}} \log \pi_{\boldsymbol{\theta}}(a_\tau | s_\tau), \text{ where } \left(s_0^{'}, a_0^{'}\right) = (s_\tau, a_\tau), \ \tau^{'} \sim \text{Geo}(1 - \gamma). \tag{220}$$

**Remark 4.** *Since Algorithm 6 plays two rollouts that cause randomness of the estimator $\widehat{G}_{\pi_{\boldsymbol{\theta}}}(s, a)$ (220) with respect to $\tau$ and $\tau^{'}$, which implies $\widehat{G}_{\pi_{\boldsymbol{\theta}}}(s, a)$ (220) is a random variable with respect to $\tau$ and $\tau^{'}$.*

**Unbiased Analysis**

We consider the expectation of $\widehat{G}_{\pi_{\boldsymbol{\theta}}}(s, a)$ (217) as follows, for any given $\boldsymbol{\theta}$,

$$\mathbb{E}\left[\widehat{G}_{\pi_{\boldsymbol{\theta}}}(s, a)\right] = \mathbb{E}_{\tau, \tau^{'}} \left[ \frac{1}{1 - \gamma} \sum_{j=0}^{\tau^{'}} r(s_j^{'}, a_j^{'}) \frac{\partial \log \pi_{\boldsymbol{\theta}}(a_\tau | s_\tau)}{\partial \theta_{s,a}} \right] \tag{222}$$

$$\overset{(220)}{=} \mathbb{E}_\tau \left[ \frac{1}{1 - \gamma} \underbrace{\mathbb{E}_{\tau^{'}} \left[ \left( \sum_{j=0}^{\tau^{'}} r(s_j^{'}, a_j^{'}) \right) \frac{\partial \log \pi_{\boldsymbol{\theta}}(a_\tau | s_\tau)}{\partial \theta_{s,a}} \middle| \left(s_0^{'}, a_0^{'}\right) = (s_\tau, a_\tau) \right]}_{:= \text{E}_1} \right], \tag{223}$$

where $\mathbb{E}_{\tau, \tau^{'}}[\cdot]$ is short for the expectation over the randomness from the variables $\tau \sim \text{Geo}(1 - \gamma), \tau^{'} \sim \text{Geo}(1 - \gamma)$, similarly, $\mathbb{E}_\tau[\cdot]$ denotes the expectation over the randomness from the trajectory $\tau \sim \text{Geo}(1 - \gamma)$.

As a similar analysis from (207) to (215), we consider the term $\text{E}_1$ (223) as follows,

$$\text{E}_1 (223) = \mathbb{E}_{\tau^{'} \sim \text{Geo}(1-\gamma)} \left[ \left( \sum_{j=0}^{\tau^{'}} r(s_j^{'}, a_j^{'}) \right) \frac{\partial \log \pi_{\boldsymbol{\theta}}(a_\tau | s_\tau)}{\partial \theta_{s,a}} \middle| \left(s_0^{'}, a_0^{'}\right) = (s_\tau, a_\tau) \right]$$

$$= \mathbb{E}_{\tau^{'} \sim \text{Geo}(1-\gamma)} \left[ \sum_{j=0}^{\infty} \mathbb{I}\{j \leq \tau^{'}\} r(s_j^{'}, a_j^{'}) \middle| \left(s_0^{'}, a_0^{'}\right) = (s_\tau, a_\tau) \right] \frac{\partial \log \pi_{\boldsymbol{\theta}}(a_\tau | s_\tau)}{\partial \theta_{s,a}}$$

$$= Q_{\pi_{\boldsymbol{\theta}}}(s_\tau, a_\tau) \frac{\partial \log \pi_{\boldsymbol{\theta}}(a_\tau | s_\tau)}{\partial \theta_{s,a}}. \tag{224}$$

According to Eq.(223) and Eq.(224), we obtain

$$\mathbb{E}\left[\widehat{G}_{\pi_{\boldsymbol{\theta}}}(s,a)\right] = \frac{1}{1-\gamma}\mathbb{E}_{\tau}\left[Q_{\pi_{\boldsymbol{\theta}}}(s_{\tau},a_{\tau})\frac{\partial\log\pi_{\boldsymbol{\theta}}(a_{\tau}|s_{\tau})}{\partial\theta_{s,a}}\right] \tag{225}$$

$$= \frac{1}{1-\gamma}\mathbb{E}_{\tau,s_t,a_t}\left[\sum_{t=0}^{\infty}\mathbb{I}\{t=\tau\}Q_{\pi_{\boldsymbol{\theta}}}(s_t,a_t)\frac{\partial\log\pi_{\boldsymbol{\theta}}(a_t|s_t)}{\partial\theta_{s,a}}\right], \tag{226}$$

where $\mathbb{E}_{\tau,s_t,a_t}[\cdot]$ is short for the expectation over the randomness from the variables $\tau\sim\mathrm{Geo}(1-\gamma)$, $s_t\sim\mathbb{P}_{\pi_{\boldsymbol{\theta}}}^{(t)}(\cdot|s_0)$, and $a_t\sim\pi_{\boldsymbol{\theta}}(\cdot|s_t)$.

Recall the result (57), we obtain

$$\frac{\partial\pi_{\boldsymbol{\theta}}(a_t|s_t)}{\partial\theta_{s,a}} = \begin{cases} \pi_{\boldsymbol{\theta}}(a|s) - (\pi_{\boldsymbol{\theta}}(a|s))^2 & \text{if } s_t = s \text{ and } a_t = a \\[2mm] -\pi_{\boldsymbol{\theta}}(a_t|s)\pi_{\boldsymbol{\theta}}(a|s) & \text{if } s_t = s \text{ and } a_t \neq a \\[2mm] 0 & \text{if } s_t \neq s \text{ or } a_t \neq a \end{cases} \tag{227}$$

$$= \begin{cases} \pi_{\boldsymbol{\theta}}(a_t|s_t) - (\pi_{\boldsymbol{\theta}}(a_t|s_t))^2 & \text{if } s_t = s \text{ and } a_t = a \\[2mm] -\pi_{\boldsymbol{\theta}}(a_t|s_t)\pi_{\boldsymbol{\theta}}(a|s) & \text{if } s_t = s \text{ and } a_t \neq a \\[2mm] 0 & \text{if } s_t \neq s \text{ or } a_t \neq a, \end{cases} \tag{228}$$

which implies

$$\frac{\partial\log\pi_{\boldsymbol{\theta}}(a_t|s_t)}{\partial\theta_{s,a}} = \begin{cases} 1 - \pi_{\boldsymbol{\theta}}(a_t|s_t) & \text{if } s_t = s \text{ and } a_t = a \\[2mm] -\pi_{\boldsymbol{\theta}}(a|s) & \text{if } s_t = s \text{ and } a_t \neq a \\[2mm] 0 & \text{if } s_t \neq s \text{ or } a_t \neq a. \end{cases} \tag{229}$$

According to the result (229), it is similar to (209)-(211), it is easy to check that Eq.(226) satisfies the condition of dominated convergence theorem (see Lemma 17), thus we rewrite Eq.(226) as follows,

$$\mathbb{E}\left[\widehat{G}_{\pi_{\boldsymbol{\theta}}}(s,a)\right] = \sum_{t=0}^{\infty}\frac{1}{1-\gamma}\mathbb{E}_{\tau,s_t,a_t}\left[\mathbb{I}\{t=\tau\}Q_{\pi_{\boldsymbol{\theta}}}(s_t,a_t)\frac{\partial\log\pi_{\boldsymbol{\theta}}(a_t|s_t)}{\partial\theta_{s,a}}\right] \tag{230}$$

$$= \sum_{t=0}^{\infty}\frac{1}{1-\gamma}\mathbb{E}_{\tau\sim\mathrm{Geo}(1-\gamma)}\left[\mathbb{I}\{t=\tau\}\right]\mathbb{E}_{s_t,a_t}\left[Q_{\pi_{\boldsymbol{\theta}}}(s_t,a_t)\frac{\partial\log\pi_{\boldsymbol{\theta}}(a_t|s_t)}{\partial\theta_{s,a}}\right] \tag{231}$$

$$= \sum_{t=0}^{\infty}\gamma^t\mathbb{E}_{s_t\sim\mathbb{P}_{\pi_{\boldsymbol{\theta}}}^{(t)}(\cdot|s_0),a_t\sim\pi_{\boldsymbol{\theta}}(\cdot|s_t)}\left[Q_{\pi_{\boldsymbol{\theta}}}(s_t,a_t)\frac{\partial\log\pi_{\boldsymbol{\theta}}(a_t|s_t)}{\partial\theta_{s,a}}\right] \tag{232}$$

$$= \sum_{t=0}^{\infty}\gamma^t\left(\sum_{s'\in\mathcal{S}}\mathbb{P}_{\pi_{\boldsymbol{\theta}}}(s_t=s'|s_0)\sum_{a'\in\mathcal{A}}\pi_{\boldsymbol{\theta}}(a'|s')\left(Q_{\pi_{\boldsymbol{\theta}}}(s',a')\frac{\partial\log\pi_{\boldsymbol{\theta}}(a'|s')}{\partial\theta_{s,a}}\right)\right) \tag{233}$$

$$= \sum_{s'\in\mathcal{S}}\sum_{a'\in\mathcal{A}}\sum_{t=0}^{\infty}\gamma^t\mathbb{P}_{\pi_{\boldsymbol{\theta}}}(s_t=s'|s_0)\pi_{\boldsymbol{\theta}}(a'|s')Q_{\pi_{\boldsymbol{\theta}}}(s',a')\frac{\partial\log\pi_{\boldsymbol{\theta}}(a'|s')}{\partial\theta_{s,a}}, \tag{234}$$

where Eq.(232) holds since

$$\mathbb{E}_{\tau\sim\mathrm{Geo}(1-\gamma)}\left[\mathbb{I}\{t=\tau\}\right] = \mathbb{P}(t=\tau) = \gamma^t(1-\gamma);$$

where Eq.(234) holds since we use the dominated convergence theorem (see Lemma 17).

We should notice that the last Eq.(234) share the same expression in the previous in Eq.(65), and following the same analysis from Eq.(65)-(68), we obtain the unbiasedness of $\widehat{G}_{\pi_{\boldsymbol{\theta}}}(s,a)$.

**Boundedness Analysis**

Recall the expectation of $\widehat{G}_{\pi_{\boldsymbol{\theta}}}(s,a)$ (217) as follows, for any given $\boldsymbol{\theta}$,

$$\mathbb{E}\left[(\widehat{G}_{\pi_{\boldsymbol{\theta}}}(s,a))^2\right] = \mathbb{E}_{\tau,\tau'}\left[\left(\frac{1}{1-\gamma}\sum_{j=0}^{\tau'}r(s_j',a_j')\frac{\partial\log\pi_{\boldsymbol{\theta}}(a_\tau|s_\tau)}{\partial\theta_{s,a}}\right)^2\right] \tag{235}$$

$$\leq \frac{4}{(1-\gamma)^2}\mathbb{E}_{\tau'}\left[\left(\sum_{j=0}^{\tau'}r(s_j',a_j')\right)^2\right] \tag{236}$$

$$\leq \frac{4}{(1-\gamma)^2}\mathbb{E}_{\tau'\sim\mathrm{Geo}(1-\gamma)}\left[\left(\tau'\right)^2\right] \leq \frac{4}{(1-\gamma)^3}, \tag{237}$$

where Eq.(236) holds since: Eq.(229) implies the boundedness of $\left|\dfrac{\partial\log\pi_{\boldsymbol{\theta}}(a_t|s_t)}{\partial\theta_{s,a}}\right| \leq 2$;

Eq.(237) holds since: recall $\tau' \sim \mathrm{Geo}(1-\gamma)$ can be expressed as follows,

Table 3: Distribution column of the distribution $\tau' \sim \mathrm{Geo}(1-\gamma)$.

| $\tau'$ | 0 | 1 | 2 | 3 | 4 | $\cdots\cdots$ | $t$ | $\cdots\cdots$ |
|---|---|---|---|---|---|---|---|---|
| | $1-\gamma$ | $(1-\gamma)\gamma$ | $\gamma^2(1-\gamma)$ | $\gamma^3(1-\gamma)$ | $\gamma^4(1-\gamma)$ | $\cdots\cdots$ | $\gamma^t(1-\gamma)$ | $\cdots\cdots$ |

which implies the distribution of $(\tau')^2$ can be presented as follows,

Table 4: Distribution column of the distribution $(\tau')^2$.

| $(\tau')^2$ | 0 | 1 | 4 | 9 | 16 | $\cdots\cdots$ | $t^2$ | $\cdots\cdots$ |
|---|---|---|---|---|---|---|---|---|
| | $1-\gamma$ | $(1-\gamma)\gamma$ | $\gamma^4(1-\gamma)$ | $\gamma^9(1-\gamma)$ | $\gamma^{16}(1-\gamma)$ | $\cdots\cdots$ | $\gamma^{t^2}(1-\gamma)$ | $\cdots\cdots$ |

thus

$$\mathbb{E}_{\tau'\sim\mathrm{Geo}(1-\gamma)}\left[\left(\tau'\right)^2\right] \leq \mathbb{E}_{\tau'\sim\mathrm{Geo}(1-\gamma)}\left[\tau'\right] = \frac{1}{1-\gamma}. \tag{238}$$

# F PROOF OF THEOREM 4

In this section, we provide the necessary proof details of Theorem 4. It is very technical to achieve the result of Theorem 4, we outline some necessary intermediate results in Section F.1 where we provide some basic lemmas, and the proof of Theorem 4 is shown in Section F.2.

**Theorem 4** *Under Assumption 1-2, $\pi_{\boldsymbol{\theta}}$ is the softmax policy defined in (9). The time-step $T$ shares a fixed low bound similar to (16). The initial $\boldsymbol{\lambda}_0 = \mathbf{0}$, $\boldsymbol{\theta}_0 = \mathbf{0}$, the parameter sequence $\{\boldsymbol{\lambda}_t, \boldsymbol{\theta}_t\}_{t=0}^{T-1}$ is generated according to Algorithm 3. Let $\eta$, $\beta$ satisfy*

$$\eta = \sqrt{\left\| \frac{d_{\pi_\star}^{\rho_0}}{\rho_0} \right\|_\infty \frac{|\mathcal{S}| \log |\mathcal{A}|}{C'(1-\gamma)T}}, \beta = \sqrt{\left\| \frac{d_{\pi_\star}^{\rho_0}}{\rho_0} \right\|_\infty \frac{4|\mathcal{S}| \log |\mathcal{A}|}{(1-\gamma)^3 \iota^2 c_\star}},$$

*where $C'$ is a positive scalar will be special later. Then for all $i \in \{1, 2, \cdots, m\}$, $\pi_t := \pi_{\boldsymbol{\theta}_t}$ satisfies*

$$\mathbb{E}\left[ \min_{t<T} \{ J(\pi_\star) - J(\pi_t) \} \right] \leq 4 \left\| \frac{d_{\pi_\star}^{\rho_0}}{\rho_0} \right\|_\infty \sqrt{\frac{|\mathcal{S}| \log |\mathcal{A}|}{c_\star(1-\gamma)^4 T}},$$

$$\mathbb{E}\left[ \min_{t<T} \{ C_i(\pi_t) - b_i \}_+ \right] \leq \frac{4\varrho \left\| \frac{d_{\pi_\star}^{\rho_0}}{\rho_0} \right\|_\infty}{\beta - \|\boldsymbol{\lambda}_\star\|_\infty} \sqrt{\frac{|\mathcal{S}| \log |\mathcal{A}|}{c_\star(1-\gamma)^4 T}},$$

*where the notation $\mathbb{E}[\cdot]$ is short for $\mathbb{E}_{\mathcal{D}_0:\mathcal{D}_{T-1}}[\cdot]$ that denotes the expectation with respect to the randomness over the trajectories $\{\mathcal{D}_t\}_{t=0}^{T-1}$.*

## F.1 AUXILIARY LEMMA

Recall Algorithm 3, at current time $t$, we obtain cost value estimator $\hat{\mathbf{c}}(\pi_{\boldsymbol{\theta}_t})$ according to (25),

$$\hat{\mathbf{c}}(\pi_{\boldsymbol{\theta}_t}) = \left( \widehat{C}_1(\pi_{\boldsymbol{\theta}_t}), \widehat{C}_2(\pi_{\boldsymbol{\theta}_t}), \cdots, \widehat{C}_m(\pi_{\boldsymbol{\theta}_t}) \right)^\top, \tag{239}$$

where each $\widehat{C}_i(\pi_{\boldsymbol{\theta}_t})$ is a rollout estimator according to:

$$\widehat{C}_i(\pi_{\boldsymbol{\theta}_t}) = \mathbb{E}_{s\sim\rho_0(\cdot)}\left[ \widehat{V}_{\pi_{\boldsymbol{\theta}_t}}^{c_i}(s) \right] = \sum_{s\in\mathcal{S}} \rho_0(s) \widehat{V}_{\pi_{\boldsymbol{\theta}_t}}^{c_i}(s), \tag{240}$$

$$\widehat{V}_{\pi_{\boldsymbol{\theta}_t}}^{c_i}(s) = \texttt{EstV}(\pi_{\boldsymbol{\theta}_t}, c_i, s). \tag{241}$$

According to Proposition 4, after some simple algebra, we obtain the unbiasedness of $\hat{\mathbf{c}}(\pi_{\boldsymbol{\theta}_t})$:

$$\mathbb{E}\left[ \hat{\mathbf{c}}(\pi_{\boldsymbol{\theta}_t}) \right] = \mathbf{c}(\pi_{\boldsymbol{\theta}_t}). \tag{242}$$

Now, we provide the boundedness of $\hat{\mathbf{c}}(\pi_{\boldsymbol{\theta}_t})$ as follows,

$$\|\hat{\mathbf{c}}(\pi_{\boldsymbol{\theta}_t})(\pi_{\boldsymbol{\theta}_t})\|_2^2 \overset{(239)}{\leq} \sum_{i=1}^m \left| \widehat{C}_i(\pi_{\boldsymbol{\theta}_t}) \right|^2, \tag{243}$$

which implies we need to bound each $|\widehat{C}_i(\pi_{\boldsymbol{\theta}_t})|$, where $i \in \{1, 2, \cdot, m\}$.

Recall the definition of $\widehat{C}_i(\pi_{\boldsymbol{\theta}_t})$ in (240), we expand it as follows,

$$\widehat{C}_i(\pi_{\boldsymbol{\theta}_t}) \overset{(240)}{=} \mathbb{E}_{s\sim\rho_0(\cdot)}\left[ \widehat{V}_{\pi_{\boldsymbol{\theta}_t}}^{c_i}(s) \right] = \sum_{s\in\mathcal{S}} \rho_0(s) \widehat{V}_{\pi_{\boldsymbol{\theta}_t}}^{c_i}(s). \tag{244}$$

According to the iteration from Algorithm 5, we rewrite $\widehat{V}_{\pi_{\boldsymbol{\theta}_t}}^{c_i}(s)$ as follows,

$$\widehat{V}_{\pi_{\boldsymbol{\theta}_t}}^{c_i}(s) = \sum_{t=0}^\tau r(s_t, a_t), \ s_0 = s, \ \tau \sim \text{Geo}(1-\gamma). \tag{245}$$

Now, we bound the expectation of $|\widehat{C}_i(\pi_{\boldsymbol{\theta}_t})|^2$ as follows,

$$\mathbb{E}\left[\left(\widehat{C}_i(\pi_{\boldsymbol{\theta}_t})\right)^2\right] \stackrel{(245)}{=} \mathbb{E}_{\tau\sim\text{Geo}(1-\gamma)}\left[\left(\sum_{s\in\mathcal{S}}\rho_0(s)\sum_{t=0}^{\tau}r(s_t,a_t)\right)^2\right]$$

$$\leq\mathbb{E}_{\tau\sim\text{Geo}(1-\gamma)}\left[\left(\sum_{s\in\mathcal{S}}\sum_{t=0}^{\tau}r(s_t,a_t)\right)^2\right] \tag{246}$$

$$\leq|\mathcal{S}|\mathbb{E}_{\tau\sim\text{Geo}(1-\gamma)}\left[\tau^2\right] \stackrel{(238)}{\leq} \frac{|\mathcal{S}|}{1-\gamma}. \tag{247}$$

Collect the results (242), (243) and (247), we achieve the next Lemma 18.

**Lemma 18.** *Let $\pi_{\boldsymbol{\theta}_t}$ be the softmax policy defined in (9). For each parameter $\boldsymbol{\theta}_t$, let $\widehat{C}_i(\pi_{\boldsymbol{\theta}_t}), \hat{\mathbf{c}}(\pi_{\boldsymbol{\theta}_t})$ be the estimator of cost value function defined in (240) and (239):*

$$\widehat{C}_i(\pi_{\boldsymbol{\theta}_t}) = \mathbb{E}_{s\sim\rho_0(\cdot)}\left[\widehat{V}_{\pi_{\boldsymbol{\theta}_t}}^{c_i}(s)\right] = \sum_{s\in\mathcal{S}}\rho_0(s)\widehat{V}_{\pi_{\boldsymbol{\theta}_t}}^{c_i}(s), \tag{248}$$

$$\hat{\mathbf{c}}(\pi_{\boldsymbol{\theta}_t}) = \left(\widehat{C}_1(\pi_{\boldsymbol{\theta}_t}), \widehat{C}_2(\pi_{\boldsymbol{\theta}_t}), \cdots, \widehat{C}_m(\pi_{\boldsymbol{\theta}_t})\right)^{\top}, \tag{249}$$

*where $\widehat{V}_{\pi_{\boldsymbol{\theta}_t}}^{c_i}(s)$ is defined in Eq.(241). Then $\hat{\mathbf{c}}(\pi_{\boldsymbol{\theta}_t})$ is an unbiased and bounded of cost value function, i.e.,*

$$\mathbb{E}\left[\hat{\mathbf{c}}(\pi_{\boldsymbol{\theta}_t})\right] = \mathbf{c}(\pi_{\boldsymbol{\theta}_t}), \tag{250}$$

$$\mathbb{E}\left[\left(\widehat{C}_i(\pi_{\boldsymbol{\theta}_t})\right)^2\right] \leq \frac{|\mathcal{S}|}{1-\gamma}, \tag{251}$$

$$\mathbb{E}\left[\|\hat{\mathbf{c}}(\pi_{\boldsymbol{\theta}_t})\|_2^2\right] \leq \frac{m|\mathcal{S}|}{1-\gamma}. \tag{252}$$

Recall Algorithm 3, we obtain the unbiased estimator as follows,

$$\widehat{\nabla_{\boldsymbol{\lambda}_t}}\mathcal{L}(\pi_{\boldsymbol{\theta}_t}, \boldsymbol{\lambda}_t) = \mathbf{b} - \hat{\mathbf{c}}(\pi_{\boldsymbol{\theta}_t}), \tag{253}$$

$$\mathbb{E}\left[\widehat{\nabla_{\boldsymbol{\lambda}_t}}\mathcal{L}(\pi_{\boldsymbol{\theta}_t}, \boldsymbol{\lambda}_t)\right] = \mathbb{E}[\mathbf{b} - \hat{\mathbf{c}}(\pi_{\boldsymbol{\theta}_t})] = \frac{\partial\mathcal{L}(\pi_{\boldsymbol{\theta}_t}, \boldsymbol{\lambda}_t)}{\partial\boldsymbol{\lambda}_t}. \tag{254}$$

According to the estimators (26) and (26), we obtain the policy gradient estimators , i.e., for each $(s,a)\in\mathcal{S}\times\mathcal{A}$,

$$\widehat{G}_{\pi_{\boldsymbol{\theta}_t}}(s,a) = \text{PG}(\pi_{\boldsymbol{\theta}_t}, r, s, a), \quad \widehat{G}_{\pi_{\boldsymbol{\theta}_t}}^{c_i}(s,a) = \text{PG}(\pi_{\boldsymbol{\theta}_t}, c_i, s, a), i = 1, 2, \cdots, m.$$

Furthermore, let the vector $\mathbf{g}_{\pi_{\boldsymbol{\theta}_t}}^c(s,a) \in \mathbb{R}^m$ collect all the policy gradient estimators of cost value function, i.e.,

$$\mathbf{g}_{\pi_{\boldsymbol{\theta}_t}}^c(s,a) = \left(\widehat{G}_{\pi_{\boldsymbol{\theta}_t}}^{c_1}(s,a), \cdots, \widehat{G}_{\pi_{\boldsymbol{\theta}_t}}^{c_m}(s,a)\right)^{\top}.$$

Let $\widehat{\mathbf{G}}(\pi_{\boldsymbol{\theta}_t}, \boldsymbol{\lambda}_t) \in \mathbb{R}^{|\mathcal{S}|\times|\mathcal{A}|}$, each $(s,a)$-element is defined as follows,

$$\widehat{\mathbf{G}}(\pi_{\boldsymbol{\theta}_t}, \boldsymbol{\lambda}_t)[s,a] = \widehat{G}_{\pi_{\boldsymbol{\theta}_t}}(s,a) - \boldsymbol{\lambda}_t^{\top}\mathbf{g}_{\pi_{\boldsymbol{\theta}_t}}^c(s,a), \tag{255}$$

then we obtain the policy gradient estimator of $\frac{\partial\mathcal{L}(\pi_{\boldsymbol{\theta}_t}, \boldsymbol{\lambda}_t)}{\partial\boldsymbol{\theta}_t}$:

$$\widehat{\nabla_{\boldsymbol{\theta}_t}}\mathcal{L}(\pi_{\boldsymbol{\theta}_t}, \boldsymbol{\lambda}_t) = \widehat{\mathbf{G}}(\pi_{\boldsymbol{\theta}_t}, \boldsymbol{\lambda}_t), \quad \mathbb{E}\left[\widehat{\mathbf{G}}(\pi_{\boldsymbol{\theta}_t}, \boldsymbol{\lambda}_t)\right] = \frac{\partial\mathcal{L}(\pi_{\boldsymbol{\theta}_t}, \boldsymbol{\lambda}_t)}{\partial\boldsymbol{\theta}_t}. \tag{256}$$

Collect the results (254), and (256), we achieve the next Lemma 19.

**Lemma 19.** *Let $\pi_{\boldsymbol{\theta}_t}$ be the softmax policy defined in (9). For each parameter $\boldsymbol{\theta}_t$, let $\widehat{C}_i(\pi_{\boldsymbol{\theta}_t}), \hat{\mathbf{c}}(\pi_{\boldsymbol{\theta}_t})$ be the estimator of cost value function defined in (240) and (239). Then, the following holds Recall Algorithm 3, we obtain the unbiased estimator as follows,*

$$\mathbb{E}[\mathbf{b} - \hat{\mathbf{c}}(\pi_{\boldsymbol{\theta}_t})] = \frac{\partial\mathcal{L}(\pi_{\boldsymbol{\theta}_t}, \boldsymbol{\lambda}_t)}{\partial\boldsymbol{\lambda}_t}. \tag{257}$$

Let the policy gradient $\widehat{\mathbf{G}}(\pi_{\boldsymbol{\theta}_t}, \boldsymbol{\lambda}_t)$ be defined in (255), then

$$
\mathbb{E}\left[\widehat{\mathbf{G}}(\pi_{\boldsymbol{\theta}_t}, \boldsymbol{\lambda}_t)\right] = \frac{\partial \mathcal{L}(\pi_{\boldsymbol{\theta}_t}, \boldsymbol{\lambda}_t)}{\partial \boldsymbol{\theta}_t}. \tag{258}
$$

### F.2 PROOF OF THEOREM 4

In this section, we show the details for the proof of Theorem 4. The proof contains two key steps: bounding the optimality gap and bounding the constraint violation. Finally, we summary the hyper-parameter setting for us to obtain the results presented in Theorem 4.

We rewrite the iteration (12) as the following stochastic version,

$$
\boldsymbol{\lambda}_{t+1} = \{\boldsymbol{\lambda}_t - \eta(\mathbf{b} - \hat{\mathbf{c}}(\pi_{\boldsymbol{\theta}_t}))\}_+ , \tag{259}
$$

$$
\boldsymbol{\theta}_{t+1} = \boldsymbol{\theta}_t + \eta \widehat{\mathbf{G}}(\pi_{\boldsymbol{\theta}_t}, \boldsymbol{\lambda}_t), \tag{260}
$$

where we calculate $\hat{\mathbf{c}}(\pi_{\boldsymbol{\theta}_t})$ and $\widehat{\mathbf{G}}(\pi_{\boldsymbol{\theta}_t}, \boldsymbol{\lambda}_t)$ according to (253) and (256). To short the expression, as before, we introduce the following notations:

$$
\pi_t(a|s) := \pi_{\boldsymbol{\theta}_t}(a|s) = \frac{\exp\left\{\theta_{s,a}^{(t)}\right\}}{\sum_{\tilde{a}\in\mathcal{A}} \exp\left\{\theta_{s,\tilde{a}}^{(t)}\right\}}, \text{ and } \pi_t := \pi_{\boldsymbol{\theta}_t}.
$$

For each time $t$, we notice the estimator $\hat{\mathbf{c}}(\pi_t)$ in the inner loop (see Line 3) involves $m$ trajectories, and estimator $\widehat{\mathbf{G}}(\pi_{\boldsymbol{\theta}_t}, \boldsymbol{\lambda}_t)$ (see Line 5) involves $(2|\mathcal{S}||\mathcal{A}| + m)$ trajectories. We use $\mathcal{D}_t$ to collect all those $(2|\mathcal{S}||\mathcal{A}| + 2m)$ trajectories,

$$
\mathcal{D}_t = \{\mathcal{T}_{t,i}\}_{i=1}^{(2|\mathcal{S}||\mathcal{A}|+2m)}.
$$

According to rollout rule in Algorithm 4, Algorithm 5, and Algorithm 2, those $(2|\mathcal{S}||\mathcal{A}| + 2m)$ trajectories among $\mathcal{D}_t$ are independent with each other.

**Bounding the Optimality Gap**.

**Lemma 20.** *The average term* $-\frac{1}{T}\sum_{t=0}^{T-1} \boldsymbol{\lambda}_t^\top (\mathbf{c}(\pi_t) - \mathbf{c}(\pi_\star))$ *is bounded as follows,*

$$
-\mathbb{E}_{\mathcal{D}_0:\mathcal{D}_{T-1}}\left[\frac{1}{T}\sum_{t=0}^{T-1} \boldsymbol{\lambda}_t^\top (\mathbf{c}(\pi_t) - \mathbf{c}(\pi_\star))\right] \leq \frac{\eta}{2}\left(b_{\max}^2 + \frac{m|\mathcal{S}|}{1-\gamma}\right), \tag{261}
$$

*where* $\mathbb{E}_{\mathcal{D}_0:\mathcal{D}_{T-1}}[\cdot]$ *denotes the expectation with respect to the randomness over the trajectories* $\{\mathcal{D}_t\}_{t=0}^{T-1}$.

*Proof.* According to the dual update (259), we have

$$
\|\boldsymbol{\lambda}_T\|_2^2 = \sum_{t=0}^{T-1}\left(\|\boldsymbol{\lambda}_{t+1}\|_2^2 - \|\boldsymbol{\lambda}_t\|_2^2\right)
$$

$$
\overset{(259)}{=} \sum_{t=0}^{T-1}\left(\left\|\{\boldsymbol{\lambda}_t - \eta(\mathbf{b} - \hat{\mathbf{c}}(\pi_t))\}_+\right\|_2^2 - \|\boldsymbol{\lambda}_t\|_2^2\right)
$$

$$
\leq \sum_{t=0}^{T-1}\left(\|\boldsymbol{\lambda}_t - \eta(\mathbf{b} - \hat{\mathbf{c}}(\pi_t))\|_2^2 - \|\boldsymbol{\lambda}_t\|_2^2\right)
$$

$$
= \sum_{t=0}^{T-1}\left(\eta^2 \|\mathbf{b} - \hat{\mathbf{c}}(\pi_t)\|_2^2 + 2\eta\boldsymbol{\lambda}_t^\top (\hat{\mathbf{c}}(\pi_t) - \mathbf{b})\right)
$$

$$
= \sum_{t=0}^{T-1}\left(\eta^2 \|\mathbf{b} - \hat{\mathbf{c}}(\pi_t)\|_2^2 + 2\eta\boldsymbol{\lambda}_t^\top (\hat{\mathbf{c}}(\pi_t) - \mathbf{b})\right)
$$

$$
\leq \sum_{t=0}^{T-1}\left(\eta^2 \|\mathbf{b} - \hat{\mathbf{c}}(\pi_t)\|_2^2 + 2\eta\boldsymbol{\lambda}_t^\top (\hat{\mathbf{c}}(\pi_t) - \mathbf{c}(\pi_t)) + 2\eta\boldsymbol{\lambda}_t^\top (\mathbf{c}(\pi_t) - \mathbf{c}(\pi_\star))\right), \tag{262}
$$

where Eq.(262) holds since we express $(\hat{\mathbf{c}}(\pi_t) - \mathbf{b})$ as follows,

$$(\hat{\mathbf{c}}(\pi_t) - \mathbf{b}) = \left(\hat{\mathbf{c}}(\pi_t) - \mathbf{c}(\pi_t)\right) + \left(\mathbf{c}(\pi_\star) - \mathbf{b}\right) + \left(\mathbf{c}(\pi_t) - \mathbf{c}(\pi_\star)\right), \tag{263}$$

and the fact $\boldsymbol{\lambda}_t \succeq \mathbf{0}$ and $\mathbf{c}(\pi_\star) \preceq \mathbf{b}$, then $\boldsymbol{\lambda}_t^\top(\mathbf{c}(\pi_\star) - \mathbf{b}) \leq 0$ implies

$$\boldsymbol{\lambda}_t^\top(\hat{\mathbf{c}}(\pi_t) - \mathbf{b}) \leq \boldsymbol{\lambda}_t^\top(\hat{\mathbf{c}}(\pi_t) - \mathbf{c}(\pi_t)) + \boldsymbol{\lambda}_t^\top(\mathbf{c}(\pi_t) - \mathbf{c}(\pi_\star)). \tag{264}$$

For each given $\boldsymbol{\theta}_{t-1}$, the estimator $\hat{\mathbf{c}}(\pi_{\boldsymbol{\theta}_t})$ is independent of $\boldsymbol{\lambda}_t$, and $\boldsymbol{\lambda}_t$ is independent of $(\hat{\mathbf{c}}(\pi_t) - \mathbf{c}(\pi_t))$. Thus according to (242), for each time $t$, the next equation holds

$$\mathbb{E}\left[\boldsymbol{\lambda}_t^\top(\hat{\mathbf{c}}(\pi_t) - \mathbf{c}(\pi_t))\right] = 0, \tag{265}$$

where $t \in \{0, 1, 2, \cdots, T-1\}$.

Furthermore, we consider the expectation of $\sum_{t=0}^{T-1} \boldsymbol{\lambda}_t^\top(\hat{\mathbf{c}}(\pi_t) - \mathbf{c}(\pi_t))$ over the trajectory $\{\mathcal{D}_t\}_{t=0}^{T-1}$ as follows,

$$\mathbb{E}_{\mathcal{D}_0:\mathcal{D}_{T-1}}\left[\sum_{t=1}^{T-1} \boldsymbol{\lambda}_t^\top(\hat{\mathbf{c}}(\pi_t) - \mathbf{c}(\pi_t))\right]$$

$$= \mathbb{E}_{\mathcal{D}_0:a_{T-2}(s)}\left[\sum_{t=0}^{T-2} \boldsymbol{\lambda}_t^\top(\hat{\mathbf{c}}(\pi_t) - \mathbf{c}(\pi_t)) + \underbrace{\mathbb{E}_{\mathcal{D}_{T-1}}\left[\boldsymbol{\lambda}_{T-1}^\top(\hat{\mathbf{c}}(\pi_{T-1}) - \mathbf{c}(\pi_{T-1}))\right]}_{\overset{(265)}{=}0}\right] \tag{266}$$

$$= \mathbb{E}_{\mathcal{D}_0:a_{T-2}(s)}\left[\sum_{t=1}^{T-2} \boldsymbol{\lambda}_t^\top(\hat{\mathbf{c}}(\pi_t) - \mathbf{c}(\pi_t))\right], \tag{267}$$

where Eq.(266) holds since the term $\sum_{t=0}^{T-2} \boldsymbol{\lambda}_t^\top(\hat{\mathbf{c}}(\pi_t) - \mathbf{c}(\pi_t))$ is independent of the trajectories $\mathcal{D}_{T-1}$, which implies

$$\mathbb{E}_{\mathcal{D}_0:\mathcal{D}_{T-1}}\left[\sum_{t=1}^{T-1} \boldsymbol{\lambda}_t^\top(\hat{\mathbf{c}}(\pi_t) - \mathbf{c}(\pi_t))\right]$$

$$= \mathbb{E}_{\mathcal{D}_0:a_{T-2}(s)}\left[\sum_{t=1}^{T-2} \boldsymbol{\lambda}_t^\top(\hat{\mathbf{c}}(\pi_t) - \mathbf{c}(\pi_t)) + \mathbb{E}_{\mathcal{D}_{T-1}}\left[\boldsymbol{\lambda}_{T-1}^\top(\hat{\mathbf{c}}(\pi_{T-1}) - \mathbf{c}(\pi_{T-1}))\right]\right].$$

Let us expand recurrently according to the mathematical induction, we achieve

$$\mathbb{E}_{\mathcal{D}_0:\mathcal{D}_{T-1}}\left[\sum_{t=1}^{T-1} \boldsymbol{\lambda}_t^\top(\hat{\mathbf{c}}(\pi_t) - \mathbf{c}(\pi_t))\right] = 0. \tag{268}$$

Recall the result (262), which implies

$$\sum_{t=0}^{T-1}\left(\eta^2 \|\mathbf{b} - \hat{\mathbf{c}}(\pi_t)\|_2^2 + 2\eta\boldsymbol{\lambda}_t^\top(\hat{\mathbf{c}}(\pi_t) - \mathbf{c}(\pi_t)) + 2\eta\boldsymbol{\lambda}_t^\top(\mathbf{c}(\pi_t) - \mathbf{c}(\pi_\star))\right) \geq 0. \tag{269}$$

Recall the result (268), and we consider to take expectation of (269) over the trajectory $\{\mathcal{D}_t\}_{t=0}^{T-1}$, then we achieve the next equation

$$\mathbb{E}_{\mathcal{D}_0:\mathcal{D}_{T-1}}\left[\sum_{t=0}^{T-1}\left(\eta^2 \|\mathbf{b} - \hat{\mathbf{c}}(\pi_t)\|_2^2 + 2\eta\boldsymbol{\lambda}_t^\top(\hat{\mathbf{c}}(\pi_t) - \mathbf{c}(\pi_t)) + 2\eta\boldsymbol{\lambda}_t^\top(\mathbf{c}(\pi_t) - \mathbf{c}(\pi_\star))\right)\right] \geq 0, \tag{270}$$

rewriting (270), we obtain the following equation,

$$-\mathbb{E}_{\mathcal{D}_0:\mathcal{D}_{T-1}} \left[ \frac{1}{T} \sum_{t=0}^{T-1} \boldsymbol{\lambda}_t^\top (\mathbf{c}(\pi_t) - \mathbf{c}(\pi_\star)) \right]$$

$$\leq \mathbb{E}_{\mathcal{D}_0:\mathcal{D}_{T-1}} \left[ \frac{\eta}{2T} \sum_{t=0}^{T-1} \|\mathbf{b} - \hat{\mathbf{c}}(\pi_t)\|_2^2 \right] \tag{271}$$

$$\leq \mathbb{E}_{\mathcal{D}_0:\mathcal{D}_{T-1}} \left[ \frac{\eta}{2T} \sum_{t=0}^{T-1} \left( \|\mathbf{b}\|_2^2 + \|\hat{\mathbf{c}}(\pi_t)\|_2^2 \right) \right]$$

$$\leq \frac{1}{2} \eta b_{\max}^2 + \frac{\eta}{2T} \mathbb{E}_{\mathcal{D}_0:\mathcal{D}_{T-1}} \left[ \sum_{t=0}^{T-1} \|\hat{\mathbf{c}}(\pi_t)\|_2^2 \right] \tag{272}$$

$$\leq \frac{1}{2} \eta b_{\max}^2 + \frac{\eta}{2} \frac{m|\mathcal{S}|}{1-\gamma} = \frac{\eta}{2} \left( b_{\max}^2 + \frac{m|\mathcal{S}|}{1-\gamma} \right), \tag{273}$$

where last Eq.(273) holds since

$$\mathbb{E}_{\mathcal{D}_0:\mathcal{D}_{T-1}} \left[ \sum_{t=0}^{T-1} \|\hat{\mathbf{c}}(\pi_t)\|_2^2 \right] = \mathbb{E}_{\mathcal{D}_0:a_{T-2}(s)} \left[ \sum_{t=0}^{T-2} \|\hat{\mathbf{c}}(\pi_t)\|_2^2 + \mathbb{E}_{\mathcal{D}_{T-1}}[\|\hat{\mathbf{c}}(\pi_{T-1})\|_2^2] \right] \tag{274}$$

$$\overset{(252)}{\leq} \mathbb{E}_{\mathcal{D}_0:a_{T-2}(s)} \left[ \sum_{t=0}^{T-2} \|\hat{\mathbf{c}}(\pi_t)\|_2^2 \right] + \frac{m|\mathcal{S}|}{1-\gamma} \tag{275}$$

$$\cdots$$

$$\leq \frac{m|\mathcal{S}|}{1-\gamma} T, \tag{276}$$

where Eq.(274) holds since the term $\sum_{t=0}^{T-2} \|\hat{\mathbf{c}}(\pi_t)\|_2^2$ is independent of $\mathcal{D}_{T-1}$;

Eq.(276) holds since we expand recurrently according to (275) by the mathematical induction; Taking the result (276) into (272), we achieve the result (273). □

**Lemma 21.** *The optimal gap is bounded as follows,*

$$\mathbb{E}_{\mathcal{D}_0:\mathcal{D}_{T-1}} \left[ \min_{t<T} \{ J(\pi_\star) - J(\pi_t) \} \right] \leq \frac{2}{(1-\gamma)^2} \left\| \frac{d_{\pi_\star}^{\rho_0}}{\rho_0} \right\|_\infty \sqrt{\frac{|\mathcal{S}| \log |\mathcal{A}| D_1'}{T}},$$

*where the positive scalar $D_1'$ will be special later.*

*Proof.* Due to the unbiasedness of $\widehat{Q}_{\pi_\theta}(s,a)$, $\widehat{V}_{\pi_\theta}(s)$ according to Algorithm 4 and Algorithm 5, we achieve a similar result as (133) in Lemma 16, but we need to consider the over the trajectory $\{\mathcal{D}_t\}_{t=0}^{T-1}$.

Concretely, we replace the terms with respect to expectation by the corresponding estimators, then we have

$$\mathbb{E}_{\mathcal{D}_0:\mathcal{D}_{T-1}} \left[ J(\pi_\star) - \frac{1}{T} \sum_{t=0}^{T-1} J(\pi_t) - \frac{1}{T} \sum_{t=0}^{T-1} \boldsymbol{\lambda}_t^\top (\mathbf{c}(\pi_\star) - \mathbf{c}(\pi_t)) \right]$$

$$\leq \frac{1}{T} \left( \frac{\chi|\mathcal{S}| \log |\mathcal{A}|}{\eta} + \frac{2\chi}{\rho_{\min}(1-\gamma)^2} \right) + 2\eta\chi C_1. \tag{277}$$

Furthermore, taking Eq.(273) (we have also presented it in Lemma 20) in to Eq.(277), we obtain

$$\mathbb{E}_{\mathcal{D}_0:\mathcal{D}_{T-1}} \left[ J(\pi_\star) - \frac{1}{T} \sum_{t=0}^{T-1} J(\pi_t) \right] \leq \frac{\chi|\mathcal{S}| \log |\mathcal{A}|}{\eta T} + \frac{2\chi}{\rho_{\min}(1-\gamma)^2 T} + \frac{\eta}{2} \left( 4\chi C_1 + b_{\max}^2 + \frac{m|\mathcal{S}|}{1-\gamma} \right). \tag{278}$$

The above result (278) implies

$$\mathbb{E}_{\mathcal{D}_0:\mathcal{D}_{T-1}} \left[ \min_{t<T} \{J(\pi_\star) - J(\pi_t)\} \right] \leq \frac{\chi|\mathcal{S}|\log|\mathcal{A}|}{\eta T} + \frac{2\chi}{\rho_{\min}(1-\gamma)^2 T} + \frac{\eta}{2} \left( 4\chi C_1 + b_{\max}^2 + \frac{m|\mathcal{S}|}{1-\gamma} \right).$$

(279)

It is similar to the analysis of result (167), if the next condition (280) holds, then above Eq.(279) obtains the optimal gap (that is shown in Eq.(283)) for the difference over $\{J(\pi_\star) - J(\pi_t)\}_{t=0}^{T-1}$.

$$\frac{\chi|\mathcal{S}|\log|\mathcal{A}|}{\eta T} = \frac{\eta}{2} \left( 4\chi C_1 + b_{\max}^2 + \frac{m|\mathcal{S}|}{1-\gamma} \right),$$

(280)

which implies

$$\eta = \sqrt{\frac{\chi|\mathcal{S}|\log|\mathcal{A}|}{\left( 2\chi C_1 + \frac{1}{2}b_{\max}^2 + \frac{mC_{\max}^2}{2(1-\gamma)^2} \right)T}} = \sqrt{\left\| \frac{d_{\pi_\star}^{\rho_0}}{\rho_0} \right\|_\infty \frac{|\mathcal{S}|\log|\mathcal{A}|}{(1-\gamma)C'} \frac{1}{T}},$$

(281)

where the positive scalar $C'$ is defined as follows,

$$C' = c_\star \left( 2\chi C_1 + \frac{1}{2}b_{\max}^2 + \frac{m|\mathcal{S}|}{1-\gamma} \right) < +\infty.$$

(282)

Taking the step-size (281) into (279), we obtain the optimal gap as follows,

$$\mathbb{E}_{\mathcal{D}_0:\mathcal{D}_{T-1}} \left[ \min_{t<T} \{J(\pi_\star) - J(\pi_t)\} \right]$$

(283)

$$\leq \sqrt{\frac{\chi|\mathcal{S}|\log|\mathcal{A}|}{T} \left( 2\chi C_1 + \frac{1}{2}b_{\max}^2 + \frac{m|\mathcal{S}|}{1-\gamma} \right)} + \frac{2\chi}{\rho_{\min}(1-\gamma)^2 T}$$

(284)

$$= \frac{1}{(1-\gamma)c_\star} \left\| \frac{d_{\pi_\star}^{\rho_0}}{\rho_0} \right\|_\infty \sqrt{\frac{|\mathcal{S}|\log|\mathcal{A}|}{T} 2C_1 \left( M_1' + 1 \right)} + \frac{2\chi}{\rho_{\min}(1-\gamma)^2 T}$$

(285)

$$\overset{(157)}{=} \frac{1}{(1-\gamma)c_\star} \left\| \frac{d_{\pi_\star}^{\rho_0}}{\rho_0} \right\|_\infty \sqrt{\frac{2|\mathcal{S}|\log|\mathcal{A}|}{T} \frac{m}{(1-\gamma)^2} \left( b_{\max} + \frac{1}{1-\gamma} \right) \left( M_1' + 1 \right)}$$

$$+ \frac{1}{(1-\gamma)c_\star} \left\| \frac{d_{\pi_\star}^{\rho_0}}{\rho_0} \right\|_\infty \frac{2}{\rho_{\min}(1-\gamma)^2 T}$$

(286)

$$= \frac{1}{(1-\gamma)c_\star} \left\| \frac{d_{\pi_\star}^{\rho_0}}{\rho_0} \right\|_\infty \sqrt{\frac{|\mathcal{S}|\log|\mathcal{A}|}{T} D_1'} + \frac{D_2'}{\rho_{\min}(1-\gamma)^3 T} \left\| \frac{d_{\pi_\star}^{\rho_0}}{\rho_0} \right\|_\infty,$$

(287)

where Eq.(285) holds since we choose the constant $M_1'$ satisfies $2\chi C_1 M_1' = \frac{1}{2}b_{\max}^2 + \frac{m|\mathcal{S}|}{1-\gamma}$, i.e.,

$$M_1' \overset{(157)}{=} (1-\gamma)c_\star^3 \left\| \frac{d_{\pi_\star}^{\rho_0}}{\rho_0} \right\|_\infty^{-1} \left( b_{\max} + \frac{1}{1-\gamma} \right)^{-1} \left( \frac{b_{\max}^2}{4m} + \frac{|\mathcal{S}|}{4(1-\gamma)^2} \right);$$

(288)

the constants $D_1'$ and $D_2'$ in (287) are defined as follows,

$$D_1' := \frac{2m}{(1-\gamma)^2} \left( b_{\max} + \frac{1}{1-\gamma} \right) \left( M_1' + 1 \right) < +\infty, \quad D_2' := \frac{2}{c_\star} < +\infty.$$

(289)

Finally, it is similar to the same analysis of (175), we conclude that if

$$T \geq \frac{(D_2')^2}{(1-\gamma)^2 |\mathcal{S}|\log|\mathcal{A}|\rho_{\min}^2 D_1'},$$

(290)

which implies

$$\mathbb{E}_{\mathcal{D}_0:\mathcal{D}_{T-1}} \left[ \min_{t<T} \{J(\pi_\star) - J(\pi_t)\} \right] \leq \frac{2}{(1-\gamma)^2} \left\| \frac{d_{\pi_\star}^{\rho_0}}{\rho_0} \right\|_\infty \sqrt{\frac{|\mathcal{S}|\log|\mathcal{A}|D_1'}{T}}.$$

(291)

$\square$

**Bounding the Constraint Violation.**

**Lemma 22.** *For any given $\|\tilde{\boldsymbol{\lambda}}\|_2 \in [0, \tilde{\beta}]$, where the positive scalar $\tilde{\beta}$ will be special later, the term $\frac{1}{T}\sum_{t=0}^{T-1}(\boldsymbol{\lambda}_t - \tilde{\boldsymbol{\lambda}})^\top(\mathbf{b} - \mathbf{c}(\pi_t))$ is bounded as follows,*

$$\mathbb{E}_{\mathcal{D}_0:\mathcal{D}_{T-1}}\left[\frac{1}{T}\sum_{t=0}^{T-1}(\boldsymbol{\lambda}_t - \tilde{\boldsymbol{\lambda}})^\top(\mathbf{b} - \mathbf{c}(\pi_t))\right] \leq \frac{1}{2\eta T}\left\|\tilde{\boldsymbol{\lambda}}\right\|_2^2 + \frac{\eta}{2}\left(b_{\max}^2 + \frac{m|\mathcal{S}|}{1-\gamma}\right). \tag{292}$$

*Proof.* We consider the parameter $\|\tilde{\boldsymbol{\lambda}}\|_2 \in [0, \tilde{\beta}]$, where the positive scalar $\tilde{\beta}$ will be special later. According to the update rule (259), we have

$$\begin{aligned}
\left\|\boldsymbol{\lambda}_{t+1} - \tilde{\boldsymbol{\lambda}}\right\|_2^2 &= \left\|\{\boldsymbol{\lambda}_t - \eta(\mathbf{b} - \hat{\mathbf{c}}(\pi_t))\}_+ - \tilde{\boldsymbol{\lambda}}\right\|_2^2 \\
&\leq \left\|\boldsymbol{\lambda}_t - \eta(\mathbf{b} - \hat{\mathbf{c}}(\pi_t)) - \tilde{\boldsymbol{\lambda}}\right\|_2^2 \\
&= \left\|\boldsymbol{\lambda}_t - \tilde{\boldsymbol{\lambda}}\right\|_2^2 - 2\eta(\boldsymbol{\lambda}_t - \tilde{\boldsymbol{\lambda}})^\top(\mathbf{b} - \hat{\mathbf{c}}(\pi_t)) + \eta^2\|\mathbf{b} - \hat{\mathbf{c}}(\pi_t)\|_2^2 \\
&\leq \left\|\boldsymbol{\lambda}_t - \tilde{\boldsymbol{\lambda}}\right\|_2^2 - 2\eta(\boldsymbol{\lambda}_t - \tilde{\boldsymbol{\lambda}})^\top(\mathbf{b} - \hat{\mathbf{c}}(\pi_t)) + \eta^2\left(b_{\max}^2 + \frac{m|\mathcal{S}|}{1-\gamma}\right),
\end{aligned} \tag{293}$$

where the last Eq.(293) holds the next result (294) holds, which is contained in the previous Eq.(273),

$$\|\mathbf{b} - \hat{\mathbf{c}}(\pi_t)\|_2^2 \leq \|\mathbf{b}\|_2^2 + \|\hat{\mathbf{c}}(\pi_t)\|_2^2 \leq b_{\max}^2 + \frac{m|\mathcal{S}|}{1-\gamma}. \tag{294}$$

We rewrite Eq.(293) as follows,

$$\left\|\boldsymbol{\lambda}_{t+1} - \tilde{\boldsymbol{\lambda}}\right\|_2^2 - \left\|\boldsymbol{\lambda}_t - \tilde{\boldsymbol{\lambda}}\right\|_2^2 \leq -2\eta(\boldsymbol{\lambda}_t - \tilde{\boldsymbol{\lambda}})^\top(\mathbf{b} - \hat{\mathbf{c}}(\pi_t)) + \eta^2\left(b_{\max}^2 + \frac{m|\mathcal{S}|}{1-\gamma}\right). \tag{295}$$

Summing Eq.(295) from $t = 0$ to $T - 1$, we achieve the following equation

$$0 \leq \left\|\boldsymbol{\lambda}_T - \tilde{\boldsymbol{\lambda}}\right\|_2^2 \leq \left\|\boldsymbol{\lambda}_0 - \tilde{\boldsymbol{\lambda}}\right\|_2^2 - 2\eta\sum_{t=0}^{T-1}(\boldsymbol{\lambda}_t - \tilde{\boldsymbol{\lambda}})^\top(\mathbf{b} - \hat{\mathbf{c}}(\pi_t)) + T\eta^2\left(b_{\max}^2 + \frac{m|\mathcal{S}|}{1-\gamma}\right),$$

which implies

$$\begin{aligned}
\frac{1}{T}\sum_{t=0}^{T-1}(\boldsymbol{\lambda}_t - \tilde{\boldsymbol{\lambda}})^\top(\mathbf{b} - \hat{\mathbf{c}}(\pi_t)) &\leq \frac{1}{2\eta T}\left\|\boldsymbol{\lambda}_0 - \tilde{\boldsymbol{\lambda}}\right\|_2^2 + \frac{\eta}{2}\left(b_{\max}^2 + \frac{m|\mathcal{S}|}{1-\gamma}\right) \\
&= \frac{1}{2\eta T}\left\|\tilde{\boldsymbol{\lambda}}\right\|_2^2 + \frac{\eta}{2}\left(b_{\max}^2 + \frac{m|\mathcal{S}|}{2(1-\gamma)}\right).
\end{aligned} \tag{296}$$

According to (242), taking expectation on Eq.(296), we obtain

$$\mathbb{E}_{\mathcal{D}_0:\mathcal{D}_{T-1}}\left[\frac{1}{T}\sum_{t=0}^{T-1}(\boldsymbol{\lambda}_t - \tilde{\boldsymbol{\lambda}})^\top(\mathbf{b} - \mathbf{c}(\pi_t))\right] \leq \frac{1}{2\eta T}\left\|\tilde{\boldsymbol{\lambda}}\right\|_2^2 + \frac{\eta}{2}\left(b_{\max}^2 + \frac{m|\mathcal{S}|}{1-\gamma}\right), \tag{297}$$

where we use the fact $\mathbb{E}[\mathbf{c}_t] = \mathbf{c}(\pi_t)$ (242), and $\boldsymbol{\lambda}_t$ is independent of $\hat{\mathbf{c}}(\pi_t)$ for a given $\boldsymbol{\theta}_{t-1}$. $\qquad\square$

**Lemma 23.** *The constraint violation is bounded as follows,*

$$\begin{aligned}
\mathbb{E}_{\mathcal{D}_0:\mathcal{D}_{T-1}}\left[\min_{t<T}\{C_i(\pi_t) - b_i\}_+\right] &\leq \frac{2}{\beta - \|\boldsymbol{\lambda}_\star\|_\infty}\left(1 + \frac{2m}{(1-\gamma)^2\iota^2}\right)\left\|\frac{d_{\pi_\star}^{\rho_0}}{\rho_0}\right\|_\infty\sqrt{\frac{|\mathcal{S}|\log|\mathcal{A}|D_1'}{(1-\gamma)^4 T}} \\
&= \frac{2\left(1 + \frac{2m}{(1-\gamma)^2\iota^2}\right)}{\tilde{\beta} - \|\boldsymbol{\lambda}_\star\|_\infty}\left\|\frac{d_{\pi_\star}^{\rho_0}}{\rho_0}\right\|_\infty\sqrt{\frac{|\mathcal{S}|\log|\mathcal{A}|D_1'}{(1-\gamma)^4 T}}.
\end{aligned}$$

Taking (297) into (277), we obtain

$$\mathbb{E}_{\mathcal{D}_0:\mathcal{D}_{T-1}}\left[J(\pi_\star) - \frac{1}{T}\sum_{t=0}^{T-1}J(\pi_t) + \frac{1}{T}\sum_{t=0}^{T-1}\boldsymbol{\lambda}_t^\top(\mathbf{b}-\mathbf{c}(\pi_\star)) - \frac{1}{T}\sum_{t=0}^{T-1}\tilde{\boldsymbol{\lambda}}^\top(\mathbf{b}-\mathbf{c}(\pi_t))\right]$$

$$\leq \frac{1}{T}\left(\frac{\chi|\mathcal{S}|\log|\mathcal{A}|}{\eta} + \frac{1}{2\eta}\left\|\tilde{\boldsymbol{\lambda}}\right\|_2^2 + \frac{2\chi}{\rho_{\min}(1-\gamma)^2}\right) + \eta\left(2\chi C_1 + \frac{b_{\max}^2}{2} + \frac{m|\mathcal{S}|}{2(1-\gamma)}\right). \quad (298)$$

Due to $\mathbf{c}(\pi_\star) \succeq \mathbf{b}$, and $\boldsymbol{\lambda}_t \succeq \mathbf{0}$, then we have

$$\mathbb{E}_{\mathcal{D}_0:\mathcal{D}_{T-1}}\left[J(\pi_\star) - \frac{1}{T}\sum_{t=0}^{T-1}J(\pi_t) + \frac{1}{T}\sum_{t=0}^{T-1}\boldsymbol{\lambda}_t^\top(\mathbf{b}-\mathbf{c}(\pi_\star)) - \frac{1}{T}\sum_{t=0}^{T-1}\tilde{\boldsymbol{\lambda}}^\top(\mathbf{b}-\mathbf{c}(\pi_t))\right]$$

$$\geq \mathbb{E}_{\mathcal{D}_0:\mathcal{D}_{T-1}}\left[J(\pi_\star) - \frac{1}{T}\sum_{t=0}^{T-1}J(\pi_t) - \frac{1}{T}\sum_{t=0}^{T-1}\tilde{\boldsymbol{\lambda}}^\top(\mathbf{b}-\mathbf{c}(\pi_t))\right]. \quad (299)$$

It is similar to the previous proof from Eq.(186) to Eq.(191), we need to show the boundedness of the expectation $\mathbb{E}_{\mathcal{D}_0:\mathcal{D}_{T-1}}\left[J(\pi_\star) - J(\tilde{\pi}_t) + \sum_{i=1}^m \tilde{\lambda}_i\left(C_i(\tilde{\pi}_t) - b_i\right)\right]$ defined in (301), which is a fundamental result for us to show the boundedness of the constraint violation.

Recall $\mathbf{c}(\pi_t) = \left(C_1(\pi_t), C_2(\pi_t), \cdots, C_m(\pi_t)\right)^\top$, and $\tilde{\boldsymbol{\lambda}} = \left(\tilde{\lambda}_1, \tilde{\lambda}_2, \cdots, \tilde{\lambda}_m\right)^\top$, combining the result (298) and (299), and taking consideration to the step-size $\eta$ defined in (281), we have

$$\mathbb{E}_{\mathcal{D}_0:\mathcal{D}_{T-1}}\left[J(\pi_\star) - \frac{1}{T}\sum_{t=0}^{T-1}J(\pi_t) + \frac{1}{T}\sum_{t=0}^{T-1}\tilde{\boldsymbol{\lambda}}^\top(\mathbf{c}(\pi_t)-\mathbf{b})\right]$$

$$= \mathbb{E}_{\mathcal{D}_0:\mathcal{D}_{T-1}}\left[J(\pi_\star) - \frac{1}{T}\sum_{t=0}^{T-1}J(\pi_t) + \sum_{i=1}^m \tilde{\lambda}_i\left(\frac{1}{T}\sum_{t=0}^{T-1}(C_i(\pi_t)-b_i)\right)\right] \quad (300)$$

$$= \mathbb{E}_{\mathcal{D}_0:\mathcal{D}_{T-1}}\left[J(\pi_\star) - J(\tilde{\pi}_t) + \sum_{i=1}^m \tilde{\lambda}_i\left(C_i(\tilde{\pi}_t) - b_i\right)\right] \quad (301)$$

$$= \mathbb{E}_{\mathcal{D}_0:\mathcal{D}_{T-1}}\left[J(\pi_\star) - J(\tilde{\pi}_t) + \tilde{\beta}\mathbf{1}_m^\top\{\mathbf{c}(\tilde{\pi}_t)-\mathbf{b}\}_+\right] \quad (302)$$

$$\stackrel{(298)}{\leq} \frac{1}{T}\left(\frac{\chi|\mathcal{S}|\log|\mathcal{A}|}{\eta} + \frac{1}{2\eta}\left\|\tilde{\boldsymbol{\lambda}}\right\|_2^2 + \frac{2\chi}{\rho_{\min}(1-\gamma)^2}\right) + \eta\left(2\chi C_1 + \frac{b_{\max}^2}{2} + \frac{m|\mathcal{S}|}{2(1-\gamma)}\right) \quad (303)$$

$$\stackrel{(307)}{\leq} \frac{1}{T}\left(\frac{\chi|\mathcal{S}|\log|\mathcal{A}|}{\eta} + \frac{m}{2\eta}\tilde{\beta}^2 + \frac{2\chi}{\rho_{\min}(1-\gamma)^2}\right) + \eta\left(2\chi C_1 + \frac{b_{\max}^2}{2} + \frac{m|\mathcal{S}|}{2(1-\gamma)}\right) \quad (304)$$

$$\stackrel{(281)}{=} \sqrt{\frac{\chi|\mathcal{S}|\log|\mathcal{A}|}{T}\left(2\chi C_1 + \frac{1}{2}b_{\max}^2 + \frac{m|\mathcal{S}|}{2(1-\gamma)}\right)} + \frac{m}{2\eta T}\tilde{\beta}^2 + \frac{2\chi}{\rho_{\min}(1-\gamma)^2 T} \quad (305)$$

$$\stackrel{(307),(287)}{=} \left(\frac{1}{(1-\gamma)^2} + \frac{2m}{(1-\gamma)^4\iota^2}\right)\left\|\frac{d_{\pi_\star}^{\rho_0}}{\rho_0}\right\|_\infty\sqrt{\frac{|\mathcal{S}|\log|\mathcal{A}|D_1'}{T}} + \frac{D_2'}{\rho_{\min}(1-\gamma)^3 T}\left\|\frac{d_{\pi_\star}^{\rho_0}}{\rho_0}\right\|_\infty, \quad (306)$$

where Eq.(301) holds due to the same reason as (184); Eq.(302) holds since we set the parameter $\tilde{\lambda}_i$ as follows

$$\tilde{\lambda}_i = \begin{cases} \tilde{\beta} = \sqrt{\chi|\mathcal{S}|\log|\mathcal{A}|}\dfrac{2}{(1-\gamma)\iota}, & \text{if } \sum_{t=0}^{T-1}(C_i(\pi_t) - b_i) \geq 0, \\[2ex] 0, & \text{if } \sum_{t=0}^{T-1}(C_i(\pi_t) - b_i) < 0. \end{cases} \quad (307)$$

Furthermore, let

$$\delta := \left(\frac{1}{(1-\gamma)^2} + \frac{2m}{(1-\gamma)^4\iota^2}\right)\left\|\frac{d_{\pi_\star}^{\rho_0}}{\rho_0}\right\|_\infty\sqrt{\frac{|\mathcal{S}|\log|\mathcal{A}|D_1'}{T}} + \frac{D_2'}{\rho_{\min}(1-\gamma)^3 T}\left\|\frac{d_{\pi_\star}^{\rho_0}}{\rho_0}\right\|_\infty,$$

then the results (306) implies,

$$\mathbb{E}_{\mathcal{D}_0:\mathcal{D}_{T-1}} \left[ J(\pi_\star) - J(\tilde{\pi}_t) + \beta \mathbf{1}_m^\top \{ \mathbf{c}(\tilde{\pi}_t) - \mathbf{b} \}_+ \right] \leq \delta. \tag{308}$$

Finally, it is similar to the proof of (196), we have

$$\mathbb{E}_{\mathcal{D}_0:\mathcal{D}_{T-1}} \left[ \min_{t<T} \{ C_i(\pi_t) - b_i \}_+ \right] \tag{309}$$

$$\leq \frac{1}{\tilde{\beta} - \|\boldsymbol{\lambda}_\star\|_\infty} \left( \left( \frac{1}{(1-\gamma)^2} + \frac{2m}{(1-\gamma)^4 \iota^2} \right) \left\| \frac{d_{\pi_\star}^{\rho_0}}{\rho_0} \right\|_\infty \sqrt{\frac{|\mathcal{S}| \log |\mathcal{A}| D_1'}{T}} + \frac{D_2'}{\rho_{\min} (1-\gamma)^3 T} \left\| \frac{d_{\pi_\star}^{\rho_0}}{\rho_0} \right\|_\infty \right).$$

Furthermore, let

$$T \geq \frac{(D_2')^2}{\left( (1-\gamma)^2 + 2m/\iota^2 \right)^2 \rho_{\min}^2 D_1' |\mathcal{S}| \log |\mathcal{A}|}, \tag{310}$$

then we obtain

$$\mathbb{E}_{\mathcal{D}_0:\mathcal{D}_{T-1}} \left[ \min_{t<T} \{ C_i(\pi_t) - b_i \}_+ \right] \leq \frac{2}{\beta - \|\boldsymbol{\lambda}_\star\|_\infty} \left( 1 + \frac{2m}{(1-\gamma)^2 \iota^2} \right) \left\| \frac{d_{\pi_\star}^{\rho_0}}{\rho_0} \right\|_\infty \sqrt{\frac{|\mathcal{S}| \log |\mathcal{A}| D_1'}{(1-\gamma)^4 T}}$$

$$= \frac{2 \left( 1 + \dfrac{2m}{(1-\gamma)^2 \iota^2} \right)}{\tilde{\beta} - \|\boldsymbol{\lambda}_\star\|_\infty} \left\| \frac{d_{\pi_\star}^{\rho_0}}{\rho_0} \right\|_\infty \sqrt{\frac{|\mathcal{S}| \log |\mathcal{A}| D_1'}{(1-\gamma)^4 T}}. \tag{311}$$

**Summarizing the Conclusion under Special Hyper-Parameter Setting**.

Finally, recall the condition for the term $T$ in (290), (310), we conclude if the time-step $T$ satisfies

$$T \geq \max \left\{ \frac{1}{(1-\gamma)^2}, \frac{1}{\left( (1-\gamma)^2 + 2m/\iota^2 \right)^2} \right\} \cdot \frac{(D_2')^2}{|\mathcal{S}| \log |\mathcal{A}| \rho_{\min}^2 D_1'},$$

the step-size $\eta$ defined in (281) satisfies

$$\sqrt{\left\| \frac{d_{\pi_\star}^{\rho_0}}{\rho_0} \right\|_\infty} \frac{|\mathcal{S}| \log |\mathcal{A}|}{(1-\gamma) C'} \frac{1}{T},$$

and the constant term $\tilde{\beta}$ satisfies

$$\tilde{\beta} \overset{(307)}{=} \sqrt{\chi |\mathcal{S}| \log |\mathcal{A}|} \frac{2}{(1-\gamma) \iota} \tag{312}$$

$$\overset{(157)}{=} \sqrt{\frac{1}{(1-\gamma) c_\star} \left\| \frac{d_{\pi_\star}^{\rho_0}}{\rho_0} \right\|_\infty |\mathcal{S}| \log |\mathcal{A}|} \frac{2}{(1-\gamma) \iota} \tag{313}$$

$$:= \sqrt{\left\| \frac{d_{\pi_\star}^{\rho_0}}{\rho_0} \right\|_\infty \frac{D' |\mathcal{S}| \log |\mathcal{A}|}{(1-\gamma)^3 \iota^2}}, \tag{314}$$

where we define the constant $D'$ as follows

$$D' := \frac{4}{c_\star}.$$

Then, according to (175) and (198), the following holds

$$\mathbb{E}_{\mathcal{D}_0:\mathcal{D}_{T-1}} \left[ \min_{t<T} \{ J(\pi_\star) - J(\pi_t) \} \right] \leq 2 \left\| \frac{d_{\pi_\star}^{\rho_0}}{\rho_0} \right\|_\infty \sqrt{\frac{|\mathcal{S}| \log |\mathcal{A}| D'}{(1-\gamma)^4 T}},$$

$$\mathbb{E}_{\mathcal{D}_0:\mathcal{D}_{T-1}} \left[ \min_{t<T} \{ C_i(\pi_t) - b_i \}_+ \right] \leq \frac{2}{\beta - \|\boldsymbol{\lambda}_\star\|_\infty} \left( 1 + \frac{2m}{(1-\gamma)^2 \iota^2} \right) \left\| \frac{d_{\pi_\star}^{\rho_0}}{\rho_0} \right\|_\infty \sqrt{\frac{|\mathcal{S}| \log |\mathcal{A}| D'}{(1-\gamma)^4 T}},$$

where each $i \in \{1, 2, \cdots, m\}$. This concludes the proof of Theorem 4.

