# OpenReview forum: "Convergence Rate of Primal-Dual Approach to Constrained Reinforcement Learning with Softmax Policy"
_ICLR.cc/2023/Conference — Submitted to ICLR 2023_

### Official Review · Reviewer_akKw · 2022-10-22

**Confidence:** 5
**Correctness:** 3
**Technical Novelty And Significance:** 3
**Empirical Novelty And Significance:** 3
**Recommendation:** 6

**Clarity, Quality, Novelty And Reproducibility:**

(1) For the sample-based result, if my understanding is correct, this paper only studies iteration-complexity not total sample complexity. In fact total sample complexity is more interested in RL community as every sample is usually very expensive to obtained in practice. When comparing with SOTA for example CSPDA, the author should note that CSPDA only requires a single sample/state-action pair at each iteration, while this paper requires a trajectory with random width at each iteration. After doing some simple math, the total sample complexity of the algorithm proposed in this paper should at least be $S^2A/(1-\gamma)^5 \epsilon^2$ (in expectation), which only matches ConRL. I hope the author can make it clear how the sample complexity in this paper compare with SOTA results in the revision.

(2) Can the author clarify why the variance of policy gradient (in sample-based setting) does not introduce additional dependence on $1/(1-\gamma)$ in the final complexity result? If it comes from the advanced proof technique proposed by the author then it definitely deserve a highlight.

I will increase my score if the author can answer the above questions.

**Strength And Weaknesses:**

Strength:
(1) This paper is clearly written and easy-to-follow
(2) The technical proof is solid and comparison with SOTA is complete

Weakness:
(1) The comparison with SOTA is not fair (I will include in the next section for detail). Thus, it is difficult to justify the contribution of this paper.

**Summary Of The Paper:**

This paper studies the iteration complexity of primal-dual gradient descent-ascent algorithm in safe RL problem. In the soft-max tabular policy setting, the author is able to provide the best iteration complexity rate for this type of algorithm.

**Summary Of The Review:**

Overall this paper provide some solid results for a fundamental problem However, as I mentioned in previous sections, the comparison with SOTA is somehow misleading and not fair thus I hope the author can clarify that part so that we can justify the contribution of this paper.

---

### Official Review · Reviewer_YnNr · 2022-10-24

**Confidence:** 5
**Correctness:** 4
**Technical Novelty And Significance:** 2
**Empirical Novelty And Significance:** 1
**Recommendation:** 3

**Clarity, Quality, Novelty And Reproducibility:**

The paper is clearly written and easy to follow. The novelty is incremental and not a significant contribution.

**Strength And Weaknesses:**

*Strengths*

- The paper is well written and easy to follow.
- The theoretical results are derived in detail.


*Weakness*

- The contribution is incremental and limited.
- As mentioned by the authors, the problem is already studied with the natural policy gradient. So what are the additional challenges to consider vanilla policy gradient? Wouldn't that follow directly?
- The results in this work follows directly from NPG-PG and Paternain's work, what are the specific challenges being addressed in this work. The challenge of designing an unbiased gradient estimator via utilizing geometric distribution rollouts is already addressed in Paternain 2018.
- The authors have discussed the strong duality and slater's condition with policy pi. Does these holds for the softmax policy parameterization as well? If not, then how to utilize the similar concepts?
- Sec 4.2 in the paper is not novel.
- There is no empirical evidence of comparison between PG-PD and NPG-PD. This is also a major downside of the results.
- Given the analysis in Ding et al, it is required to mention the additional mathematical challenges  exist to do the analysis in this paper.


**Summary Of The Paper:**

The authors have considered the problem of constrained RL which is formulated via constrained Markov decisions process. The problem is then solved via first formulating the Lagrangian, and writing down the primal dual method. The authors have used the vanilla policy gradient in the primal domain to solve the problem. Theoretical results are derived. The paper is nicely written.

**Summary Of The Review:**

The most of the mathematical tools/analysis used in the paper already exists. The major concern is regarding the contribution/novelty which is not significant enough.

---

### Official Review · Reviewer_zsRP · 2022-10-25

**Confidence:** 4
**Clarity, Quality, Novelty And Reproducibility:** The paper is in general clear, and th…
**Correctness:** 1
**Technical Novelty And Significance:** 1
**Empirical Novelty And Significance:** Not applicable
**Recommendation:** 1

**Strength And Weaknesses:**

Strength: The paper is in general well-written, and the literature review is thorough.


Weakness: Some of the theoretical arguments are not accurate/correct.

1. The motivation of studying policy gradient primal-dual method might need more justification: given the existing results on natural PG primal-dual method (with potentially better rates, since the $c_*$ in the result in this paper might be very small), why should we still focus the vanilla PG-PD algorithm, with tabular softmax parameterization (except out of theoretical interest)?
2. Some of the claims regarding technical contributions might need improvement: for example, it was claimed with quite some space in the abstract and intro that, designing unbiased value function estimators with finite-horizon trajectories for the infinite-horizon case is one of the contributions. However, it is known that such a technique has been widely used in the literature, e.g., Zhang et al., 2020, Agarwal et el., 2020, Ding et al., 2020, etc.
3. Some of the comparisons with the literature might need to be improved. For example, the results in Table 1 might not be completely comparable: some of them are for finite-horizon settings, and more importantly, some of them are the online exploration setting, while some of them are the generative model setting; some allow constraint violation and some do not; are the sample/iteration complexity in the table referring to suboptimality guarantee or constraint violation guarantee? I am not sure if it is accurate to compare them this way.
4. As it is a theory-oriented paper, it would be helpful to summarize the "novelty of the Techniques" (and the intuition) upfront, in introduction, so that people can better compare and understand the results.
5. Some parts of the writing might need more care.
Typos:
1) what is the definition of $\Pi_S$ in (3)? if it means "stationary policy" (which misses "Markov"), why it is "assumed" to be convex in Theorem 1? Is it a subset of it?
2) would it be more accurate to write $\in\argmax$ instead of $=\argmax$ in (4)? Similarly for the definition of $L_D(\lambda)$: does the maximizer have to be unique? So is the optimal dual variable $\lambda_*$.
3) Sentence before (12): the problem (8) should also be "constrained", not "unconstrained".
4) Right after Algorithm 1: "adapted" -> "adopted"?
6. I think there is a fundamental issue with the technical part of the paper: it is known that the optimal policy of CMDP might not be deterministic. See http://readingsml.blogspot.com/2020/03/constrained-mdps-and-reward-hypothesis.html for example. However, the analysis in the paper strongly depends on the “optimal policy” being deterministic: see argument around (14) and page 33.

Given 6, I might not be able to recommend acceptance of the paper given its current form.




**Summary Of The Paper:**

This paper studies primal-dual approach to solve constrained reinforcement learning (RL) problems under the framework of constrained MDPs. It studies the global convergence of policy gradient primal-dual method under softmax parameterization, for both model-based and model-free settings. The paper is in general well-written, and easy to follow, with some solid theoretical results.

**Summary Of The Review:**

The paper studied a fundamental setting for safe RL: solving constrained MDP using primal-dual policy gradient methods, under tabular softmax parameterization. The writing is clear and easy to follow. However, there is some technical issue with the current paper.

---

### Official Review · Reviewer_5KTD · 2022-10-29

**Confidence:** 4
**Clarity, Quality, Novelty And Reproducibility:** The results are clear, while the appr…
**Correctness:** 3
**Technical Novelty And Significance:** 3
**Empirical Novelty And Significance:** 2
**Recommendation:** 3

**Strength And Weaknesses:**

+: The analysis seems correct.

-: The related work is misrepresented. The generative model should give additive samples, and not multiplicative.

-: The approach is standard, and novelty seems limited.

-: The constraint violations in the literature have been improved to zero violations, which should be incorporated.

-: Based on the table presented, the results do not seem the state of the art.

-: The paper is giving iteration complexity, and not sample complexity.

**Summary Of The Paper:**

This paper considers primal-dual approach to solve constrained reinforcement learning (RL) problems, where we formulate constrained reinforcement learning under constrained Markov decision process (CMDP).

**Summary Of The Review:**

This paper considers primal-dual approach to solve constrained reinforcement learning (RL) problems, where we formulate constrained reinforcement learning under constrained Markov decision process (CMDP). The paper needs significant modifications to be at the level of ICLR paper as mentioned in weaknesses.

---

### Decision · Program_Chairs · 2023-01-20

**Decision:**

Reject

**Justification For Why Not Higher Score:**

incremental contribution, inaccurate claims, lack of proper empirical evaluation.

**Justification For Why Not Lower Score:**

N/A

**Metareview: Summary, Strengths And Weaknesses:**

The reviewers are concerned about the limited novelty of the work. There are also issues with the experimental results, their validity, whether the comparison is fair, and lack of comparison with existing relevant methods. There are claims in the paper that the reviewers are concerned about their accuracy. Finally, there is an important question about the motivation behind this work: given the results on natural PG primal-dual method, why would it be important to focus on vanilla PG primal-dual? All in all, the paper does not seem to be ready for publication and requires more work and major improvement. I would also recommend that the authors think about the motivation behind their work and the possible contributions of this line of work.